# Groundwater influence on soil moisture memory and land-atmosphere fluxes in the Iberian Peninsula

Alberto Martínez-de la Torre[1,a] and Gonzalo Miguez-Macho[1]

[1]Nonlinear Physics Group, Faculty of Physics, Universidade de Santiago de Compostela, Spain
[a]now at: Centre for Ecology and Hydrology, Wallingford, United Kingdom

*Correspondence to:* Alberto Martínez-de la Torre (albmar@ceh.ac.uk)

**Abstract.** Groundwater plays an important role in the terrestrial water cycle, interacting with the land surface via vertical fluxes through the water table and distributing water resources spatially via gravity-driven lateral transport. It is therefore essential to have a correct representation of groundwater processes in land surface models, as land-atmosphere coupling is a key factor in climate research. Here we use the Land Surface and Groundwater Model LEAFHYDRO to study the groundwater influence on

soil moisture distribution and memory, and evapotranspiration (ET) fluxes in the Iberian Peninsula over a 10-year period. We validate our results with time series of observed water table depth from 623 stations covering different regions of the Iberian Peninsula, showing that the model produces a realistic water table, shallower in valleys and deeper under hilltops. We find patterns of shallow water table and strong groundwater-land surface coupling over extended interior semi-arid regions and river valleys. We show a strong seasonal and interannual persistence of the water table, which induces bimodal memory in the soil

moisture fields; soil moisture "remembers" past wet conditions, buffering drought effects, and also past dry conditions, causing a delay in drought recovery. The effects on land-atmosphere fluxes are found to be significant, on average over the region, ET is 17.4 % higher when compared with a baseline simulation with LEAFHYDRO's groundwater scheme deactivated. The maximum ET increase occurs in summer (34.9 %; 0.54 mm day$^{-1}$). The ET enhancement is larger over the drier southern basins, where ET is water limited (e.g., the Guadalquivir basin and the Mediterranean Segura basin), than in the northern

Miño/Minho basin, where ET is more energy limited than water limited. In terms of river flow, we show how dry season baseflow is sustained by groundwater originating from accumulated recharge during the wet season, improving significantly on a free-drain approach, where baseflow comes from water draining through the top soil, resulting in rivers drying out in summer. Convective precipitation enhancement through local moisture recycling over the semiarid interior regions and summer cooling are potential implications of these groundwater effects on climate over the Iberian Peninsula. Fully coupled land surface and

climate model simulations are needed to elucidate this question.

## 1  Introduction

Groundwater dynamics and its interactions with the land-atmosphere system play a key role in the terrestrial water cycle. Groundwater exchanges with the land surface occur via vertical fluxes through the water table surface, and horizontal water redistribution, via gravity-driven lateral transport within the saturated zone. A shallow water table slows down drainage and

affects soil moisture and evapotranspiration (ET), particularly in water limited environments. The Iberian Peninsula, with a typical Mediterranean climate of dry growing season, is one such region where ET is largely constrained by water availability.

Soil moisture memory refers to the persistence of wet or dry anomalies in the soil, after the atmospheric conditions that originated them have passed. In turn, if there is high land-atmosphere coupling, that is, if the conditions of the soil can have a significant impact on atmospheric dynamics, then soil moisture memory can influence weather conditions, with major implications for seasonal and long-term forecasting (e.g., Koster et al., 2010). The Mediterranean region, a transitional zone between year-long wet and dry climates, presents high soil moisture memory and high land-atmosphere coupling (Seneviratne et al., 2006). This is mostly on account of the high seasonality of precipitation, with a pronounced dry and warm summer and a wetter and colder winter. ET is highly water limited and hence dependent on soil moisture availability and precipitation from previous seasons. At the subsurface, soil moisture is linked to the water table when the latter is relatively shallow, hence the weak time variability of groundwater (Fan et al., 2007) might enhance greatly this high soil moisture memory.

The water table depth is the main indicator of the intensity of groundwater-soil moisture coupling, and consequently of how much memory the long timescales of variation of groundwater can induce in soil moisture. The water table is linked to the unsaturated zone above by two-way fluxes: the downward gravitational flux and the capillary flux. The net flux is downward in the wet season and for some time afterwards, when groundwater continues to be recharged, but upward capillary fluxes can dominate in the dry season and, if the water table is sufficiently shallow, groundwater will reach the root zone to meet surface ET demands. There is observational evidence from field experiments showing that groundwater can be one of the main sources for ecosystem ET in water limited environments (e.g., Lubczynski, 2008; Liu et al., 2016), and that the groundwater table depth determines strong sensitivities of local rooting depths (Fan et al., 2017). In the Iberian Peninsula in particular, David et al. (2007) found that during the summer drought in a plot in southern Portugal, daily soil moisture fluctuations in the top-1 m related to transpiration could be attributed to groundwater via isotopic analysis. These authors estimated that up to 70 % of the evapotranspired water had its origin in groundwater over that area. Beyond experimental plots, observational evidence of the connection between groundwater and soil moisture over a larger area is reported by Sutanudjaja et al. (2013), using remote sensing soil moisture products to predict groundwater heads in time and space over Germany, and reproducing groundwater head fluctuations reasonably well, particularly in shallow water table areas, where soil moisture dynamics are tightly connected to groundwater head positions.

Many modelling reports concerning soil moisture memory lack the interaction of the top-soil crust with the water table. However, groundwater dynamics are increasingly being taking into consideration in climate and ecosystem modelling studies. There are several studies that do explicitly include groundwater processes (e.g., Maxwell and Miller, 2005; Fan et al., 2007; Miguez-Macho et al., 2007; Niu et al., 2007; Yuan et al., 2008; Lo and Famiglietti, 2010; Leung et al., 2011; Maxwell et al., 2011; Vergnes et al., 2014; Decker, 2015). In general they all conclude that the interaction with a shallow water table drastically changes soil moisture dynamics and affects ET fluxes in water limited conditions. Notwithstanding, most modelling schemes fail to produce a realistic water table spatial distribution, which compromises the generality of their results. One important reason why this happens is that most Land Surface Models (LSMs) treat the evolution of the water table as a process dominated by vertical fluxes, as they do with soil moisture, ignoring or misrepresenting the lateral gravitational groundwater flow, which

is the main driver of the water table distribution across the landscape (Fan et al., 2013; de Graaf et al., 2015; Maxwell and Condon, 2016). The main modelling challenge thus remains to couple groundwater to soil moisture with a realistic water table; only then the importance of their mutual interaction for climate can be reliably assessed in the large scale.

Groundwater impacts directly surface water. Miguez-Macho and Fan (2012b) explicitly evaluated the influence of ground-water on the Amazon's surface water dynamics and showed that the water table buffers the impact of the seasonal drought on surface waters due to its longer timescale of evolution, and supports wetlands in lowlands and valley floors, where a persistently shallow water table is found because of lateral flow convergence or slow drainage associated with the flatness of the terrain and low elevation. Decharme et al. (2010) also pointed out the potential significance of the groundwater store as an uncertainty in simulating continental hydrological systems. Vergnes et al. (2012) improved the spatio-temporal variability of streamflow and, particularly over France's main rivers, summer baseflow, when using a river routing model that included groundwater-river exchanges.

The Iberian Peninsula is a region of high precipitation seasonality and land-atmosphere coupling (Rios-Entenza and Miguez-Macho, 2014), where the importance of an accurate representation of soil moisture is well known (e.g., Sánchez et al., 2010; Jiménez et al., 2011). Sánchez et al. (2010) used in situ soil moisture measurements to validate a water balance model over a shallow water table region in the Duero/Douro basin during 2002, and found that their model underestimated soil moisture. We speculate that the role of groundwater in these kind of modelling studies should be taken into account and may change soil moisture behaviour in shallow water table regions. Groundwater memory of long past surface episodes has also been recorded in Doñana National Park, Spain, by Serrano and Zunzunegui (2008), finding higher observed wet phase duration correlations with the previous two years' rainfall than with the previous one year's rainfall. Moreover, over the upper Guadiana basin, Mejías-Moreno et al. (2012) showed how during the hydrological years 2009-2010 and 2010-2011 with rainfall 50 % above climatology, the water table depth recovered 4 m and 8 m, respectively, and during the 2011-2012 hydrologically dry year, the water table still recovered 2.5 m up to spring level, in a way not observed since 1983 at the location (Martínez-Cortina et al., 2011). Additionally, the recovery of several ponds in La Mancha Húmeda (Biosphere Reserve in the upper Guadiana basin) during the dry year 2012 was reported in the Spanish press, reflecting the importance of groundwater influence on surface hydrology. A modelling study (Hassan et al., 2014) in the Sardon basin (a small shallow water table basin in the central Iberian Peninsula; $\sim$80 km$^2$) incorporated groundwater interactions with the soil and surface water, finding significant figures for groundwater recharge (16 % of precipitation), exfiltration ($\sim$11 % of precipitation) and groundwater evapotranspiration ($\sim$5 % of precipitation).

Understanding the relevant processes within the water cycle becomes of major importance for the integrated management of water resources provided the high irrigation withdrawal from wells or directly from surface waters in the Iberian Peninsula (e.g., Garrido et al., 2006; Daccache et al., 2014; Hunink et al., 2015). There has been a spectacular increase over the last decades in intensive groundwater use for irrigation in most arid and semi-arid regions of Spain, carried out mainly by individual farmers, often with little planning and control on the part of governmental water authorities (Garrido et al., 2006).

In this paper, we present a modelling study linking groundwater to soil moisture, land-atmosphere interactions and surface water at the regional scale in the Iberian Peninsula. We investigate the role of groundwater in the hydrology of the region,

focusing first, on its impact on soil moisture spatial variability, dynamics and long-term memory, second, on its effects on land-atmosphere ET fluxes, and third, on its direct impact on river flow.

Our work uses the LEAFHYDRO model, which includes water table dynamics considering explicitly lateral flow (Fan et al., 2007; Miguez-Macho et al., 2007; Miguez-Macho and Fan, 2012a, b; Martinez et al., 2016). The model formulation and parametrization of groundwater relies on a high resolution steady state simulation of the equilibrium position of the water table. In the lower resolution, time-evolving run with the full model, the water table pattern stems from the high resolution simulation, where local drainage is better resolved, and is therefore realistic, reflecting topography with deeper water table under hilltops and shallower in valleys. Preceding our discussions on groundwater-soil moisture interactions over the Iberian Peninsula, in this study we validate the modelled water table with available time series of observations in Spain and Portugal.

## 2 Model description and settings

### 2.1 Groundwater and Land Surface Model LEAFHYDRO

LEAF (Land-Ecosystem-Atmosphere-Feedback) is the LSM included in the Regional Atmosphere Modeling System (RAMS) (http://rams.atmos.colostate.edu/). It calculates heat and water fluxes and storages in the land surface, resolving several vertical soil layers of variable depth. The vertical flux $F$ between adjacent unsaturated soil layers is given by the Richards' Equation:

$$F = -\rho_w K_\eta \frac{\partial(\Psi + z)}{\partial z} \tag{1}$$

where $\rho_w$ (kg m$^{-3}$) is the density of liquid water, $K_\eta$ (m s$^{-1}$) is the hydraulic conductivity at a given volumetric water content $\eta$, $\Psi$ (m) is the soil capillary potential and $z$ (m) is height. Parameters $K_\eta$ and $\Psi$ depend on the water content and the pore-size index of the soil. To compute such parameters, the model follows the Clapp and Hornberger (1978) formulation:

$$K_\eta = K_f \left(\frac{\eta}{\eta_f}\right)^{2b+3}, \qquad \Psi = \Psi_f \left(\frac{\eta_f}{\eta}\right)^b \tag{2}$$

where b is the soil pore-size index and subscript $f$ denotes quantity at saturation. A canopy layer including vegetation and surface air interacts with the soil/surface water below and the atmosphere above. Derived from the version 2 of LEAF (Walko et al., 2000), LEAFHYDRO incorporates a groundwater dynamics scheme, based on the formulation presented by Miguez-Macho et al. (2007).

LEAFHYDRO introduces a prognostic water table depth that fluctuates in the model as a result of three main interactions: 1) two-way water flux between the saturated and the unsaturated zones, 2) two-way water flux between the groundwater reservoir and rivers, and 3) lateral groundwater flow within the saturated zone. Hence, the mass balance of the dynamic groundwater reservoir in a LEAFHYDRO cell is given by

$$\frac{dS_G}{dt} = \Delta x \Delta y R + \sum_{n=1}^{8} Q_n - Q_r \tag{3}$$

where $S_G$ (m$^3$) is the groundwater storage in a model column, $\Delta x \Delta y$ (m$^2$) is the horizontal resolution of the model, $R$ (m s$^{-1}$) is the flux through the water table, $Q_n$ (m$^3$ s$^{-1}$) is the lateral flow from or to the n$^{th}$ neighbouring model cell, and $Q_r$

($m^3 \, s^{-1}$) is the groundwater-rivers exchange. Fluxes $R$ and $Q_n$ in Equation 3 are assumed to be positive when going into the groundwater reservoir and negative when going out of it, whereas $Q_r$ is positive when going into the river and negative when going from the river into the groundwater reservoir. Fig. 1 (left) represents the groundwater balance in a model cell (cell 1).

The water flux through the water table or net recharge $R$ is the sum of gravitational downward groundwater recharge and capillary flux, and depending on soil wetness and atmospheric demand, it can be downwards, causing the water table to rise, or upwards, causing the water table to deepen. LEAFHYDRO calculates $R$ under the 2 possible scenarios in Fig. 1 (right).

In scenario $a$, the water table appears within the soil layers resolved by the model (4 m) and its position is diagnosed at a given time step as that yielding the equilibrium soil water content ($\eta_{eq1}$) in the unsaturated portion of layer 1. Hence, there is no vertical water flux between layers 1 and 2, and from Eq. 1:

$$\frac{\partial(\Psi + z)}{\partial z} = 0, \quad or \quad \Psi_1 - \Psi_2 = z_2 - z_1 \tag{4}$$

where $z_1$ and $z_2$ are the depths of midlayers 1 and 2, respectively. Applying the relationship between $\Psi$ and $\eta$ in Eq. 2, the equilibrium soil water content in the unsaturated portion of layer 1 is obtained as

$$\eta_{eq1} = \eta_{f1} \left( \frac{\Psi_{f1}}{\Psi_{f2} + z_2 - z_1} \right)^{1/b_1} \tag{5}$$

Then, assuming even distribution of total soil water in layer 1, the $\eta_1$ that the model calculated in the soil fluxes routine following Richards' Equations can also be calculated as

$$\eta_1 = \eta_{eq1} \left( \frac{h_1 - wtd}{h_1 - h_2} \right) + \eta_{f1} \left( \frac{wtd - h_2}{h_1 - h_2} \right) \tag{6}$$

where $wtd$ (m) is the water table depth, $h_1$ (m) is the depth of the top of layer 1 and $h_2$ (m) is the depth of the top of layer 2. Now, from Eq. 6, the water table depth is diagnosed as

$$wtd = \frac{\eta_{f1} h_2 - \eta_{eq1} h_1 + \eta_1 (h_1 - h_2)}{\eta_{f1} - \eta_{eq1}} \tag{7}$$

And finally $R$ is the amount of water flowing from or to the unsaturated portion of layer 1 necessary to cause the rise or fall of the water table from the position in the previous time step to the position calculated in Eq. 7 ($\Delta wtd$):

$$R = \Delta wtd(\eta_{f1} - \eta_{eq1}) \tag{8}$$

In scenario $b$, the water table lies below the resolved soil layers. A bottom layer is added that extends from the resolved soil layers depth to the water table position, centred in point C. This is a virtual layer, of variable thickness in space and time, and since it can be much thicker than the layer above and therefore cause instability issues for finite difference schemes, an auxiliary layer of the same thickness as the deepest resolved layer is added, centred at point B. The water content of point B is initially obtained by linear interpolation between A and C (water content in the virtual layer containing C is part of the model initialization). Then, given the water content at A and B, the flux between the two can be calculated. Similarly, an auxiliary layer of equal thickness as the virtual layer and centered in point D is added below the water table. The water content gradient

between C and D (layer containing D is saturated) determines the flux between the two, which is the net recharge $R$. Knowing the fluxes above and below, the new water content $\eta_C$ of the layer containing C can be determined by mass balance. The change in water content in the virtual layer is finally added to or taken away from the groundwater reservoir, calculated similarly to Eq. 8 as

$$\Delta wtd = \frac{R}{\eta_{fdeep} - \eta_C} \tag{9}$$

where $\eta_{fdeep}$ is the saturation soil water content for the soil at the water table position depth.

Groundwater-rivers exchange $Q_r$ follows Darcy's law and it is proportional to the elevation difference between the water table and the river water surface in the cell, as

$$Q_r = \frac{\bar{K_{rb}}}{\bar{b_{rb}}} \left( \bar{w}_r \sum L_r \right) (wth - \bar{z}_r) \tag{10}$$

where $\bar{K_{rb}}$ (m s$_{-1}$)is the mean river bed hydraulic conductivity in the cell, $\bar{b_{rb}}$ (m) is the mean thickness of river bed sediments in the cell, $\bar{w}_r$ (m) is the mean river width within the cell, $L_r$ (m) is the length of individual channels in the cell (the river depth is neglected for the calculation of contact area), $wth$ (m) is the water table head in the cell (as $wth = z + wtd$, where $z$ (m) is the cell elevation), and $\bar{z}_r$ (m) is the mean river elevation in the cell. This flux can occur as groundwater discharge (subsurface runoff) into gaining streams when the water table is above the river, sustaining stream baseflow, or as river infiltration into the groundwater reservoir in losing streams when the water table is below river bed. For gaining streams, LEAFHYDRO approach combines the physically based parameters of Darcy's law into a parameter called river conductance, commonly used in groundwater modeling literature, like the MODFLOW model (Harbaugh et al., 2000). Even though the river conductance is physically based and observable, detailed data on river geometry and bed sediments are lacking for the region studied, hence it needs to be parametrized. Such parametrization consists in a representation of the river conductance that includes two contributions; an equilibrium part, and a dynamic part that depends on the water table deviation from equilibrium at the time. Further details on this dynamic river conductance parametrization and discussion on its choice are found in Miguez-Macho et al. (2007). For losing streams, the distance of flow or river bed thickness in Eq. 10 is the same as the water table minus riverbed elevation difference (third parenthesis in Eq. 10, only with negative sign provided that $wth < \bar{z}_r$), and hence these factors cancel out one another, leaving the flux calculation to be given by

$$Q_r = -\bar{K_{rb}}\bar{w}_r \sum L_r \tag{11}$$

Therefore, the losing stream flux $Q_r$ in the model is not dependant on the water table position, once the latter is below riverbed, but on the groundwater-rivers hydraulic connection.

Lateral groundwater flow $Q_n$ is determined by the slope of the water table surface, applying Darcy's law the water flux from the n$^{th}$ neighbour into a model cell is given by

$$Q_n = cT\frac{wtd_n - wtd}{l} \tag{12}$$

where $c$ (m) is the flow cross-section connecting the cells, $T$ (m$^2$ s$^{-1}$) is the flow transmissivity between the cells, $wtd$ and $wtd_n$ (m) are the water table depths for the centre cell and the n$^{th}$ neighbour cell, respectively, and $l$ (m) is the distance

between cells. $T$ is calculated as a vertical integration down from the water table depth of the lateral hydraulic conductivity at saturation $K_L$ (Fan et al., 2007), which is derived from the vertical conductivity $K_V$ using the anisotropy ratio parameter $\alpha$ relating both parameters as $\alpha = K_L/K_V$. We apply values of $\alpha$ dependent on the clay content of the soil and within the range of observations in nature, as detailed in Fan et al. (2007). For vertical conductivity, we assume an exponential decay with

depth, as

$$K_{V_f} = K_0 exp\left(-\frac{z'}{f}\right) \tag{13}$$

where $K_0$ (m s$^{-1}$) is the known value at 1.5 m deep, $z'$ (m) is the depth below 1.5 m and $f$ (m) is the e-folding depth, calculated as a function of terrain slope $\beta$ as $f = 75/(1+150\beta)$, with a lower limit of 4 m in steep terrain where $\beta \geq 0.118$. Further details on this formulation of $Q_n$ and parametrization choices are found in Fan et al. (2007) and Gestal-Souto et al. (2010). All water

fluxes represented by arrows in Fig. 1 (left) are referred to cell 1, thus the groundwater lateral flux $Q_2$ is an incoming flux from the neighbouring cell 2 with a higher water table head ($wth2$), and $Q_3$ is an outgoing flux towards the neighbouring cell 3, which presents a lower water table head ($wth3$).

When there is vegetation on the surface, the water and heat exchanges between vegetation and the surrounding canopy air parameterization is based on Avissar et al. (1985). This methodology uses PFTs (Plant Functional Types) constant through

the simulation period, assigning a type to each cell that will determine parameters like the root depth, the minimal stomatal conductance (that will be increased by atmospheric factors) and the LAI (Leaf Area Index), that will affect the calculation of canopy resistance, transpiration and evaporation from the canopy surface. The transpiration is taken from the moistest level in the root zone.

## 2.2   Initial land surface and river parameters

The 11 soil textural classes used in LEAFHYDRO, necessary to derive soil parameters in Eq. 2 controlling the vertical water fluxes, are defined by the United States Department of Agriculture (USDA) from fractions of silt, clay and sand. The data for top (0-0.30 m depth) and bottom (0.30-4 m depth) soil layers comes originally from the Food and Agricultural Organization of the United Nations (FAO) world database (http://fao.org/soils-portal/soil-survey). Other processes in the model, such as evapotranspiration, need parameters dependent on the vegetation type (PFTs) at the land surface. For vegetation type we use

the COordination of INformation on the Environment (CORINE) Land Cover Project database (EEA, 1994).

The river flow scheme included in LEAFHYDRO (Miguez-Macho et al., 2007; Miguez-Macho and Fan, 2012b) uses the Manning's Equation. For the river flow scheme and in order to calculate the equilibrium river conductance and the groundwater-streams flux in gaining streams detailed in section 2.1, the model requires the following initial parameters: flow direction, river width, river length and river slope. To calculate such parameters in the domain, we used the United States Geological Survey

(USGS) HydroSHEDS 15 arc second resolution data (Lehner et al., 2008). The variables extracted from the HydroSHEDS database were: $fd$ (flow direction), $acc$ (accumulated drainage area) and $dem$ (void filled elevation). The methodology to calculate the requested parameters (Fig. 2) follows the following steps: 1) First, the high resolution (15 arc second) cell with the largest $acc$ within a low resolution cell (model grid is 2.5 km) is spotted; 2) The $fd$ of this cell (black arrows in Fig. 2),

together with the location of the low resolution cell containing the high resolution cell where it flows to, determine the flow direction of the low resolution cell (blue arrows in Fig. 2); 3) The flow of the main high resolution stream within every low resolution cell is then followed, highlighting the stream (red streams in Fig. 2); 4) The distance made by this high resolution main stream is taken as the low resolution river length $L$; 5) The low resolution river slope $s_r$ is taken as the average slope for all high resolution cells that take part in the main high resolution stream, where the high resolution slopes have been previously calculated from the flow direction $fd$ and the elevation $dem$; 6) The low resolution drainage area $A_d$ is calculated aggregating the area of all high resolution cells within a low resolution cell, and then accumulating it from all cells addressed to a given cell with the use of the low resolution $fd$; 7) Finally, the river width $w_r$ is calculated using an estimation of the net recharge $R$ (a 1º resolution global climatic recharge from the Mosaic LSM) and the drainage area $A_d$ in each low resolution cell, as discussed by Lucas-Picher et al. (2003): $w_r = (0.00013Q_m + 6.0)Q_m^{1/2}$, where $Q_m$ is the annual mean discharge passing through a river section, approximated for this calculation by the accumulation of flow $Q = RA_d$ for the cells along the low resolution stream.

## 2.3 Atmospheric forcing data

The atmospheric forcing data for the LEAFHYDRO simulations were extracted from the ECMWF ERA-Interim reanalysis database (Berrisford et al., 2011). Surface pressure, 2 m temperature and surface wind speed data are reanalysis fields at 6-hourly time resolution. The incoming surface radiation (shortwave and longwave) and precipitation (convective and large-scale) fields are forecasts from reanalysis datasets and are available at 3-hour time resolution. ERA-Interim is presented in a reduced Gaussian grid with approximately uniform 79 km spacing for surface grid cells.

The precipitation data to drive our simulations must account for the orographic heterogeneity of the Iberian Peninsula as much as possible. We use a regional high resolution analysis dataset of daily precipitation over Spain and Portugal (IB02: Herrera et al., 2012; Belo-Pereira et al., 2011). The IB02 dataset was built using all stations from the climatic monitoring network of both the Spanish Meteorological Agency (AEMET) and the Portuguese Meteorological Institute (IPMA), and presents a horizontal resolution of 0.2º. Once the daily precipitation is read and interpolated into the model grid, the model temporally disaggregates the daily values throughout the day using 3-hourly ERA-Interim precipitation distribution. Hence, the model uses the IB02 daily analysis data for bias-correction of daily totals and ERA-Interim data for precipitation distribution throughout the day.

## 2.4 Equilibrium Water Table Depth and initial soil moisture

In order to initialize the model, we used a climatic or Equilibrium Water Table Depth (EWTD) for the Iberian Peninsula. It was calculated using a simple two-dimensional groundwater model described by Fan et al. (2007); Fan and Miguez-Macho (2010), which finds EWTD as the long-term balance between the atmospheric influence in the form of climatic groundwater recharge ($R = P - ET - Q_{sr}$; recharge equals precipitation minus evapotranspiration minus surface runoff) and the topographic influence given by gravity-driven lateral convergence. This two-dimensional groundwater model has been recently applied to New Zealand by Westerhoff et al. (2018), providing improved water table estimations for data-sparse regions.

We used topography data at high spatial resolution (9 arc seconds) in the EWTD calculation to properly capture topographic variability and local hillslope gradients (Gestal-Souto et al., 2010). A three-step process was followed, where first, a low resolution (1º) global climatic recharge from the Mosaic LSM was used to calculate a first estimate of EWTD by ingesting it to the 2D model using the high resolution topography; second, the resulting first high-resolution estimate of EWTD is simply

aggregated to a grid of 2.5 km to serve as initial water table condition for LEAFHYDRO full LSM 10-year test run (1989-1998), and third, a new high resolution EWTD was recalculated forcing the 2D model with the groundwater net recharge obtained with the LEAFHYDRO test run at 2.5 km and the high resolution topography. The test run uses precipitation analysis and other forcings (see section 2.3) at higher resolution than the 1º climatic recharge from MOSAIC initially feeding the EWTD model, and produces a much more realistic recharge, totally compatible with our simulation settings. The resulting EWTD is the basis

of the initial water table condition for the final LEAFHYDRO simulation and is shown in Fig. 3. The water table is relatively close to the surface in many areas, such as in the Inner Plateau (northern and southern subregions), where in spite of the semi-arid climate, the water table is shallow due to the slow drainage and lateral groundwater convergence from the surrounding mountains. Low elevation coastal plains and river valleys also present a shallow water table. Topography dominates the water table depth spatial heterogeneity; however, the climatic pattern, in general wetter and with higher recharge toward the Atlantic

than in the Mediterranean, also has an influence, with shallower water table depths in the west and deeper in the east of the Iberian Peninsula.

The initial soil moisture profiles are of major importance in LSM studies (e.g., Betts, 2004; Beljaars et al., 1996). Here, we initialized the soil solving numerically the Richards equation, prescribing the climatic net recharge as top and saturation at the EWTD as lower boundary condition. Thus, the initial soil moisture content in our simulations is in equilibrium with the water

table below.

## 2.5   Simulations set-up

We performed a 10-year period simulation (referred to hereafter as WT, Water-Table) using LEAFHYDRO to investigate the role of groundwater dynamics in the Iberian Peninsula soil moisture fields, land-atmosphere fluxes and surface water. In addition, to help isolate the role of the groundwater, another simulation was performed with the groundwater scheme deactivated

(referred to hereafter as FD, Free-Drain). The FD simulation uses the commonly adopted free-drain approach, where soil water is allowed to drain out of the soil column and into the local rivers at a rate set by the hydraulic conductivity at the water content of the bottom soil layer. The potential drawback of this approach is that the escaping water is no longer available to sustain subsequent dry period ET. It should work very well where the water table is deep and the soil is sandy, but where the water table is shallow and the soil is clay rich, it may underestimate soil water storage and overlook persistence.

The simulation domain is a Lambert-Conformal grid centered in the Iberian Peninsula (Fig. 4) with a spatial resolution of 2.5 km. The simulated period starts in January 1989 and finishes in December 1998. This timeframe was chosen long enough to include wet and dry years in order to better isolate the groundwater influence on soil moisture memory. It includes the 1991-1995 drought, reported as the most severe in the Iberian Peninsula during the last 60 years (Libro Blanco del Agua en España, 2000; published by the Spanish Department of Natural Environment), as well as other dry and wet spells over different

pluviometric Iberian regions, hence allowing for a study of groundwater effects under different climatic conditions. The length of the time period of simulation is a significant improvement with respect to the prior LEAFHYDRO seasonal study over North America (Miguez-Macho et al., 2007).

The time resolution for resolving heat and water fluxes in the soil and at the land surface is 60 s. The time step for groundwater-streams exchange, groundwater mass balance and water table adjustment in the WT run is 900 s.

## 3 Validation

### 3.1 Water table depth and time evolution validation

A realistic water table depth ($wtd$) estimation is essential to couple groundwater and soil moisture in modelling studies. A modelled dynamic water table should oscillate around its equilibrium position (EWTD) at different timescales in response to rainfall events, unsaturated soil demands and multi-year dry or wet spells, as it does in nature (Fan et al., 2007; David et al., 2007; Vincke and Thiry, 2008). Thus, a validation of the time evolution of the simulated $wtd$ across the studied region is necessary to support the findings of this work. We use $wtd$ observations in this section to validate the model performance in terms of water table depth and time evolution across the Iberian Peninsula. The observational $wtd$ data were provided by the Institute of Geology and Mining of Spain (IGME), several Confederaciones Hidrográficas (Spanish agencies managing the main basins within the country) and the National Information System for Hydrological Resources of Portugal (SNIRH). The time and space coverage of these datasets are irregular. For validation, we eliminated stations with a water table deeper than 100 m in order to rule out measurements in confined aquifers as much as possible, since they are not hydrologically connected to the land surface. We also discarded stations with a sustained declining trend steeper than 0.05 m month$^{-1}$, very likely caused by pumping. After these eliminations, we only use stations with at least 3 years of data within the 10-year simulation period, leaving 623 stations suitable for $wtd$ validation (Fig. 4).

Some studies that incorporate explicitly groundwater dynamics in land surface modelling find groundwater impacts on the top soil and land-atmosphere fluxes to be negligible when the wtd is below 5 m (Leung et al., 2011; Maxwell and Kollet, 2008). However, the contribution of water tables below 5 m deep to ET by upward capillary flux has been reported to be significant at sites over Amazonia (Fan and Miguez-Macho, 2010), where groundwater sustains significant fractions of the observed ET even when the water table is at depths of around 10 m. As a compromise, we consider in our analysis water tables above 8 m deep to be shallow in this Iberian Peninsula study. A total of 31.4 % of the Iberian Peninsula territory is found in the WT simulation to have shallow mean water table (Fig. 4), which gives an estimate of the high potential for groundwater influence on top soil hydrology and land-surface fluxes in the region. With regard to the observations, 203 of the studied stations present a shallow water table (mean $wtd \leq 8$ m) during the simulation period.

The water table evolution at a given grid cell in the model must be understood as an approximation to the different possible behaviours of the natural water table within the cell. This situation is a handicap for $wtd$ validation, since the 2.5 km resolution of the WT simulation is coarse in comparison with the scale of the observed variability in topography and $wtd$. Also, the vertical design of the model detailed in section 2.1 only allows for one water table to be found per grid cell. Out of the 623

stations analysed, 136 do not correspond uniquely to one model cell and are contained in only 60 cells (2 or 3 stations per cell, orange points in Fig. 4). These different observation sites contained in one model cell do not always present the same mean $wtd$ or $wtd$ time evolution (Points 15 and 16 in Fig. 5). Inside the Point 15 grid cell (Inner Plateau, northern subregion), there are 3 different observation sites that present very different (up to 20 m difference) $wtd$ values along the simulation period (red,

green and purple series), making it very difficult to assess the accuracy of the model result (blue series). Inside the Point 16 grid cell in the southeast, there are 2 observation sites, and the model underestimates the depth of both but reflects correctly the annual cycle and the long-term trends, deepening from 1992 to 1996 and reaching shallower depths from 1996 to 1998.

Approximately one third of the stations present a shallow mean $wtd$ ($\leq 8$ m), and 66.0 % of them are also found to have shallow mean water table by the model.

In terms of mean $wtd$ error, 14.0 % of stations present less than 2 m difference between simulated and observed mean $wtd$ at the available observation times (red points in Fig. 4). If we only consider shallow water table observations (mean $wtd \leq 8$ m), 33.0 % of them present less than 2 m difference with the mean simulated $wtd$. Fig. 5 shows examples of time series of the model performance at points where the model captures the mean water table depth (Points 1 to 12). Focusing not on the mean $wtd$ values but on their time evolution, we find that 32.3 % of the station time series present a correlation coefficient over 0.5

with the simulated time series (green points in Fig. 4; the correlation is calculated using the full time series available in the observed data). Points in different shallow water table areas over the Iberian Peninsula show the model accuracy at representing the seasonal fluctuations and the long-term deepening and rising trends (Points 1 to 14 in Fig. 5; note that Points 1 to 12 fall into both red and green categories). Point 12 near the Duero/Douro river mouth in Portugal is an example on capturing the seasonal cycle and a slightly upward $wtd$ trend throughout the simulation. At some of the very shallow water table points in 5,

the amplitudes of the wtd variations are larger in the model than in the observations.

A total of 94 stations present a steep $wtd$ long-term trend (slope $\geq 0.035$ m month$^{-1}$) within the prescribed limits to avoid spurious trends due to pumping (slope $\leq 0.05$ m month$^{-1}$), and 26.6 % of them are captured in the simulation (purple points in Fig. 4) by a mean slope difference between the observation and simulation series lower than 0.02 m month$^{-1}$.

In spite of the aforementioned challenges in validating the simulated water table with point observations, we can conclude

that the model's performance is reasonably good at shallow water table points, but significantly worse where the water table is deeper. The spatial pattern of deep water table under hilltops and shallow in valleys is thus realistic in the model, however inaccurate water table levels might be where groundwater is deep. Seasonal cycles and long-term trends in groundwater are in general better captured. Notwithstanding, for the purpose of this work, LEAFHYDRO's skill in representing the shallow water table regions in the Iberian Peninsula is the key factor, since when it is deeper, the two way linkage with the top soil, which is

the focus of our study, weakens considerably.

### 3.2    River flow comparison

The groundwater-surface water link in LEAFHYDRO has been presented and validated over North America using river flow observations (Miguez-Macho et al., 2007). Similarly, for this work over the Iberian Peninsula, all calculations and parametrizations are physically based and no model calibration has been carried out for any basin. For validation, we use monthly river flow

observational data at 6 gauge stations along the main rivers of the region (Fig. 6), provided by the Centre for Hydrographics Studies (CEH, Spanish Department of Natural Environment).

There is a clear underestimation of the winter river flow by the model. Two factors contribute to this bias. First, a lack of precipitation in the forcing data, since the IB02 analysis dataset original resolution (0.2°) is coarser than our model simulations and the station density (7 km in Spain and 11.7 km in Portugal) is not sufficient to capture precipitation peaks due to orographic enhancement over the mountains, which is very pronounced in the northern cordilleras. In addition, the model does not incorporate a parametrization for subgrid saturation excess runoff (Clark and Gedney, 2008), likely associated to heavy precipitation (Martínez-de la Torre et al., 2019). Secondly, there are also model deficiencies in the representation of the river-groundwater linkage when the water table is deep. It is often the case, especially in complex terrain, that the mean water table of the cell is too high above the river, hence resulting in a fairly constant baseflow throughout the year, with very smooth rainy season peaks. This produces summer baseflow that is realistic, matching observations in some cases (right graphs in Fig. 6), but more often yields a bias in the modelled streamflow.

Another important issue affecting river flow validation is the existing high anthropogenic intervention in river regimes, both direct, through regulation reservoirs, power generation plants or irrigation withdrawals, and indirect, from groundwater extraction in wells. According to Libro Blanco del Agua en España (2000), the fraction of natural flow under this affected regime is high in northern rivers (Duero/Douro, Miño/Minho, Llobregat), but low in southern rivers: 52 % in the Guadiana (Badajoz station, Extremadura, Spain), 44 % in the Guadalquivir (Alcalá del Río station, Andalucía, Spain) and only 4 % in the Segura at the river mouth (Guardamar station, Valencia, Spain). In consequence, a comparison with observations might not be very meaningful for some rivers, such as the Gualdaquivir, where the observed river flow at station 5 is much lower than the model result, producing the poorest correlation index.

## 4   Results

### 4.1   Long-term net recharge and seasonal variability

The study of the net recharge variable, defined as the flux across the water table, gives us an understanding of the connection between groundwater and soil. This connection is bimodal, and depends on soil wetness conditions; 1) negative recharge occurs when the net flux is downward and the groundwater reservoir acts as a sink for precipitation infiltration; 2) positive recharge results when upward capillary fluxes dominate and the groundwater reservoir takes the role of water source for soil moisture, feeding ET demands.

An accurate estimation of the net recharge is of major importance for water management systems, mainly over high irrigation areas such as the semi-arid regions of the Iberian Peninsula, as it will help to understand where unconfined aquifers are at risk of being overexploited. Here, as a first result, we present the net recharge estimation produced by our long-term LSM simulation with a fully dynamic water table (WT), where upward capillary fluxes are accounted for (Fig. 7a). Long-term negative (downward) recharge patterns resemble precipitation patterns (Fig. 7b), although with the amount diminished by surface runoff and ET. Streamflow seepage when the water table is below riverbed results also in a net negative recharge in

some locations. Long-term positive recharges occur where a net upward capillary flux to satisfy ET demands is sustained by groundwater lateral convergence from surrounding cells of higher water table head. These lateral fluxes represent a more remote water source than vertical drainage through the soil above and are particularly relevant in water limited regions. The long-term upward flux in Fig. 7a is small over wide flat areas, since regional lateral groundwater convergence from the distant surrounding mountains is slow. This is the case of the Inner Plateau (northern and southern subregions), which has a dry climate with insufficient rainfall to sustain high ET for long. However, in river valleys where steep slopes in the water table head drive strong local lateral groundwater flow convergence, groundwater-fed ET can exceed precipitation by large amounts, resulting in higher values for the positive recharge. This is apparent in Fig. 7a along the main river valleys crisscrossing the dry Mediterranean areas of the Iberian Peninsula.

The net recharge presents strong seasonal variability (Fig. 7c-f). It undergoes a clear seasonal cycle following precipitation and ET cycles, which, in Mediterranean climates, are typically in opposite phases. The seasonal character of ET in the Iberian Peninsula (Sobrino et al., 2007) is induced by water availability and incoming radiation; maximum values and higher spatial variability are found in spring and summer, whereas minimum values and variability appear in autumn and winter, when the incoming radiation is lower and the leaf area index decreases. Downward fluxes (negative recharge) are strong during winter and spring in the humid areas in the west and north, responding to wet season infiltration, which furthermore is not diminished by any significant ET. Late spring precipitation and the little summer rainfall are mostly consumed by the high ET demands in the growing season, thereby substantially weakening any negative recharge during summer and autumn. Where the water table is deeper (Fig. 3), the wet season peak in recharge is delayed and variations are buffered, until it becomes a rather constant and diminished flux with much less variability throughout the year. The later is also true for shallower water tables in drier areas. Fluxes reverse and become upward mainly over shallow water table regions in spring, once the high ET consumes top soil moisture, and reach the maximum in summer when the balance between precipitation and evapotranspiration is at its seasonal minimum ($P - ET = -0.80$ mm day$^{-1}$). Any upward flux decreases significantly during autumn because of the lower ET, and only in a few locations groundwater still feeds the reduced winter ET demands. The net annual flux might be upward, as discussed above, in areas where significant groundwater convergence compensates the lack of precipitation to sustain ET.

## 4.2 Water table control on soil moisture and ET

The large scale soil moisture pattern over the Iberian Peninsula is dominated by seasonal climatic variations and a non-seasonal dependence on soil texture. The influence of groundwater is however very relevant at shorter spatial scales, as shown by the difference between the soil moisture fields (in terms of volumetric water content) from the WT and FD runs (Fig. 8a). The relation with the water table depth distribution (Fig. 8b) is very apparent. Soil moisture differences reach higher values where the water table is shallower, and are minimum or negligible in regions with deeper water table. The similarity between the patterns of soil moisture differences (WT-FD runs) and $wtd$ (WT run) illustrates the controlling role of groundwater on soil moisture spatial variability, by wetting the soil from below in regions of shallow water table: 1) low elevation flatlands, such as coastal plains and the low Guadalquivir basin, where sea level limits drainage 2) narrow river valleys where lateral groundwater flow convergence is strong, 3) wider plains surrounded by mountains (Inner Plateau), due to a combination of poor drainage,

streamflow infiltration losses and lateral groundwater convergence, albeit slow, from the high terrain around, and 4) the humid areas with high recharge rates in the northwest of the Iberian Peninsula (Galicia and northern Portugal).

The seasonal cycle of soil moisture differences between the WT and FD runs, averaged over shallow water table regions (as defined in Section 3.1: $wtd \leq 8$ m, about 30 % of the total area of the Iberian Peninsula), is shown in Fig. 8c. Only shallow water table points are considered because the effect of groundwater on soil moisture where the water table is deep is very small. Soils are always wetter in general when the water table is close to the surface, as the positive value of the differences at all times indicate. In absolute terms (purple curve), the differences in soil moisture are maximum in spring, similarly strong in winter and summer, and weaker in autumn. In relative terms (blue bars), however, groundwater reveals its stronger influence at water scarcity times, reducing soil moisture seasonality: 24.4 % soil moisture increase in spring and 23.9 % in summer. In the wet season, during autumn and winter, when soils are in general wetter, the impact of upward capillary fluxes from the water table is to slow down drainage, therefore increasing somewhat top soil moisture. In the dry season, which in the region coincides with the spring and summer growing period, root zone soil moisture and drainage can be drastically reduced. Upward capillary fluxes from a shallow water table thus dominate and may reach the root zone, sustaining, at least partially, ET demands. It is then that the effect of these upward fluxes from groundwater is more relevant, resulting in soil moisture that is significantly higher than otherwise would be if the water table were deeper.

The difference in summer ET between the WT and the FD simulations (Fig. 9) reveals an important enhancement over shallow water table regions, where there is more soil moisture availability (Fig. 8). In summer, the mean daily ET, averaged over the whole Iberian Peninsula, in increased by 34.9 % (0.54 mm day$^{-1}$) as a result of the connection between the soil and the water table. This ET enhancement is maximum in summer, as discussed above, but considering the whole year is still as high as 17.4 % (0.24 mm day$^{-1}$).

## 4.3    Water table persistence and soil moisture memory at pluri-annual timescale

The choice of a 10-year simulation period allows us to analyze groundwater persistence and influence on soil moisture at a timescale of several years. For this purpose, we first calculate the time correlation indexes between the annual anomalies (differences with respect to the annual means) of soil moisture and two key players affecting its time evolution: precipitation and water table depth, for the full 9 hydrological years simulated (September 1989 to August 1998; hy1 to hy9). In the FD simulation without groundwater, annual anomalies of soil moisture and precipitation are positively correlated at every point of the Iberian Peninsula (Fig. 10, left), with lower indexes over the northern mountains, where freezing conditions and snow cover during part of the year prevent infiltration and make soil moisture insensitive to precipitation. However, when the same relationship is evaluated using the WT simulation with groundwater, the correlation values between precipitation and soil moisture anomalies decrease in all shallow water table regions (Fig. 10, centre), indicating that soil moisture reliance on precipitation is diminished there. The correlation index between precipitation and soil moisture annual anomalies averaged over the whole Iberian Peninsula decreases from 0.81 in the FD run to 0.72 in the WT run, and from 0.82 to 0.60 when averaging only over shallow water table regions. In the WT experiment with groundwater, soil moisture anomalies are highly and positively correlated (values over 0.5) with $wtd$ anomalies (Fig 10, right) over many shallow water table regions in the

southern half of the Iberian Peninsula, the Southern and Northern Sub-Plateaus or Galicia in the northwest, precisely where the correlation between soil moisture and precipitation anomalies is reduced. In this WT simulation, the averaged correlation index between $wtd$ and soil moisture anomalies is 0.43 over the whole area, and 0.93 when averaging only over shallow water table regions. In shallow water table areas, where soil and groundwater are connected, soil moisture anomalies are thus more

linked to groundwater anomalies (0.93 correlation index) than to precipitation anomalies (0.6 correlation index), suggesting that, by wetting the soil from below, groundwater buffers soil moisture reliance on precipitation, decoupling somewhat soil and atmospheric conditions.

To better describe the connection between precipitation and groundwater timescales of variation, Fig. 11 shows a collection of paired $wtd$ and precipitation anomaly plots over the Iberian Peninsula, chronologically ordered for the 9 hydrological years

considered. These include the 1992-1995 drought (hy2 to hy6), one of the worst in the last century. Following an initial overall wet year - wetter than the mean in the centre and south (positive anomalies in the top row) but drier in the north (negative anomalies) -, precipitation anomalies in the Iberian Peninsula are clearly negative from hy2 to hy6 during the drought period, and then become clearly positive during the last 3 years. This precipitation regime is transfered with a 1-2 year delay to the groundwater. The water table (anomalies shown in the bottom row), deepens slowly during the drought, up to hy6 and then

starts to rise from the very wet hy7, but it never reaches the initial position with a positive anomaly, since recovering from the severe drought episode would likely take longer. The water table delayed response is clearly reflected in the area averaged anomalies, which do not become negative until two years into the drought. Intense climate events at the surface are buffered in the groundwater, where they may be "remembered" for some years. For instance, on the eastern coast, the hy1 and hy2 positive precipitation anomalies cause the water table to be shallower than the mean up to the end of hy4, even though precipitation

anomalies in the region are negative during hy3 and hy4. Furthermore, over the northern Cantabrian coast, the very high precipitation anomaly during hy4 translates into shallow $wtd$ anomalies in hy4 and hy5, in spite of negative precipitation anomalies during hy5 over most of the area. Another example is the high precipitation anomaly during hy5 in the northwestern part of the Iberian Peninsula, which is not sufficient to produce a shallow anomaly in $wtd$, since the region comes from 2 consecutive very dry years (hy3 and hy4), and the water table stays deeper than the mean over most of the region even after

hy5.

Groundwater's long timescales of variation result in water table persistence through atmospheric wet and dry periods. Since the water table is connected to the top soil via capillary fluxes where it is relatively shallow, groundwater's delayed and extended response to climatic events affects soil moisture evolution (Fig. 10). To further evaluate the influence of water table persistence on soil moisture memory at pluri-annual timescale, we study a 250 $x$ 225 km$^2$ region containing La Mancha Húmeda (Fig. 12;

region highlighted in Fig. 11, first plot of bottom row), a well know wetland area within the otherwise dry Southern Plateau inland Spain. The water table is very close to the surface over significant portions of this region (Fig. 8b), and therefore it is a marked wet spot that helps understanding the water table's influence on soil moisture memory at a fine scale. Soil moisture anomalies in the FD run where groundwater is not considered (second row) are a direct response to precipitation anomalies (top row). However, the soil moisture evolution patterns in the WT run with groundwater (third row) reflect a combination of

precipitation and $wtd$ patterns (bottom row), which in turn are also affected by earlier precipitation anomalies. Indeed, focusing

on the series of $wtd$ anomalies (bottom row), it is apparent that for the large shallow water table area in the center of the figures, wetter than normal conditions from hy1 extend one year into the drought period (hy3), and dry anomalies only develop after the third dry year (hy5). Due to the severity of the drought, depressed water tables persist even three years after the starting of the wet period in hy7. The regional pattern of soil moisture anomalies (third row) reflects primarily direct climatic influence over deeper water table areas, but also groundwater connections over shallow water table regions, and it's therefore a mosaic of areas of direct and delayed response to climate anomalies. These figures illustrate how depending of the extent of shallow water table regions, regionally averaged soil moisture anomalies can be decoupled from present precipitation anomalies, reflecting instead past climatic events.

## 4.4 Analysis by basin

River basins can be considered as independent, topography-driven regions integrating the hydrological system behaviour. Considering that the Iberian Peninsula presents very different precipitation regimes, we analyse in this section the WT and FD run results averaged over the main Iberian basins (Fig. 13). Since soil moisture dynamics are changed by the interaction with a shallow water table, land-atmosphere fluxes, and in particular ET, are thus also expected to be altered (as shown in Section 4.2). The focus is to understand the effects of groundwater-soil interactions on ET over different climatic regions and periods.

As mentioned earlier, the most significant impact of groundwater on soil moisture and hence land-atmosphere fluxes takes place over shallow water table regions ($wtd \leq 8$ m), where groundwater is hydraulically connected to the upper soil through upward capillary fluxes. The fraction of shallow water table cells is approximately one third of the total in the Atlantic basins, ranging from 31.6 % in the Guadiana basin to 34.7 % in the Miño/Minho basin, and one fourth in Mediterranean basins: 24.3 % in the Ebro, 27.5 % in the Júcar and 28.1 % in the Segura basin, since the drier eastern half of the Iberian Peninsula presents an overall deeper water table.

Fig. 14 shows the evolution of precipitation (seasonally accumulated, blue bars), water table depth in the WT run (red lines) and the differences between WT and FD runs in soil moisture (orange lines) and ET (green bars), averaged for the shallow water areas of the main Iberian basins for the 10 years of simulation. During the wet season (autumn-winter) the water table rises due to precipitation infiltration, but since drainage is slowed down as compared with the free-draining FD run, the soil moisture difference between both experiments also follows an upward trend. This soil moisture difference is maximal at the start of the growing period in spring because of accumulation during the wet season, meaning that there is more soil water availability to meet ET demands in the WT run, which results in a marked peak in ET difference. During late summer, the higher soil water availability in the WT run continues due to capillary rise and there is ET enhancement until the next wet season, when the cycle starts again. The increase in ET is more significant in the drier southern basins, where ET is more water limited (an overall 21.4 % ET enhancement in the Atlantic Guadalquivir and 28.4 % in the Mediterranean Segura basins). In the northern Miño/Minho basin, where ET is not so much water limited as energy limited, the ET enhancement is less significant (13.3 %).

Focusing on longer time-scale patterns, in terms of climatic conditions, the 10-year simulation period presents a long drought from 1990 to 1995/1996 in all basins except the Miño/Minho (see precipitation, blue bars in Fig. 14). The water table follows

the long-term precipitation trend, with a slow $wtd$ decline during the drought, and then a gradual recovery (or at least a change in tendency from deepening to stabilizing or slightly rising in the cases of Ebro and Segura) in the last 3 years of simulation, when precipitation is high. This long-term $wtd$ tendencies are passed on to soil moisture: differences between WT and FD values are smaller when the water table is depressed and increase as it becomes shallower. Therefore, soil moisture availability
"remembers" past dry and wet years due to the strong connection with groundwater, and in turn, this soil moisture memory induces ET memory, more clearly so in the southern drier basin, where the intensity of land-atmosphere fluxes depends not only on precipitation from the previous wet season but also on the long-term $wtd$ evolution. For example, in the Guadiana basin, ET differences during 1996 are clearly lower than during 1990, even though there was much higher precipitation in the previous wet season in 1996 than in 1990. This is explained by the soil moisture memory induced by groundwater; in 1996 the
water table is depressed after several years of drought; hence the soil and ET fluxes behave more like in a free-drain approach, without connection to the water table. The high infiltration from a very wet winter is thus rapidly lost and unavailable to rise back up by capillarity in the growing season. In contrast, in 1990, the water table is shallower and thus infiltration slower, soils wetter and as they dry, upward capillary fluxes can reach the top soil to feed ET demands.

Fig. 15 shows power spectrum analyses of soil moisture and $wtd$ time series for the same basins as in Fig. 14. The power
spectrum of a given time series was simply based on performing the Fourier transform. Again, only shallow water table areas are considered. The figure illustrates more clearly the coupling between groundwater and soil in shallow water table regions and the long-term memory induced by groundwater in the combined system. Water table power spectra (insets in Fig. 15) peak at one year frequencies. The annual cycle, linked to that of the surface water balance (shown in Fig. 14 on ET and soil moisture), is very marked in the humid Miño basin, in the north, with an oceanic climate with abundant precipitation and
rare water scarcity. Longer timescales of evolution dominate as the climate gets drier towards the south and east of Iberia, where multi-year droughts alternating with wetter periods are the norm. Soil moisture spectra show evidence of a pronounced annual cycle in both the WT experiment with groundwater (blue lines) and the FD run with a free-drain approach (red lines). This is explained by the seasonality of precipitation and ET, which in a Mediterranean climate are in opposite phases, as mentioned earlier. There is an increased relevance of long frequencies of variation in the southern and Mediterranean basins as
a consequence of the irregularity of precipitation in these regions. Very little difference exists between the spectra of WT and FD simulations in the humid Miño basin; they however diverge in drier climates, with soil moisture in the WT run presenting significantly higher amplitudes at long timescales than in the FD experiment without groundwater. The higher weight of longer timescales of variation in the WT soil moisture series reflects those in the water table series (insets) and is an indication of the strength of the coupling between soil and a shallow water table in semiarid climates. This is particularly evident in the Segura
basin, in the southeast, the driest area of the Iberian Peninsula.

## 4.5 Groundwater influence on river flow

Finally, we briefly discuss groundwater's modulation of streamflow. The water table and rivers are linked in LEAFHYDRO through the groundwater-rivers flux $Q_r$, which can go in either direction. In the experiment with a free-drain (FD) approach, however, the water draining through the bottom soil layer at 4 m depth, goes directly into the rivers, without delay. Loosing

streams are not contemplated either in FD. We choose the Ebro river for this discussion because this is where the model exhibits the best performance (i.e. less winter underestimation and best matching of seasonal cycle) of all major Iberian rivers. Besides, it has the largest draining basin in the domain. Fig. 16 shows results from observations and from both experiments for the Ebro river, close to its mouth. The winter river flow underestimation general to all rivers in the WT run (Section 3.2) is not as pronounced in the FD run. In contrast, the summer baseflow in the WT run is higher and closer to observations than that of the FD simulation; without groundwater, rivers dry out in summer practically every year in the FD run. In the WT run, the groundwater reservoir feeds rivers during the dry season with accumulated wet season infiltration, sustaining summer flows. After summer, the WT river flow rises with autumn precipitation from September-October, while in the FD simulation it takes longer to recover from the dry summer, since soils are too dry and infiltration is delayed. The better representation of the seasonal river flow by the WT run is reflected in the improvement in the monthly mean time series correlation index with observations (Fig. 16, left).

## 5 Discussion

The strong groundwater-soil coupling has a noticeable impact on soil moisture patterns across the Iberian Peninsula. Where the water table is close to the surface, soil moisture availability increases; thus soil moisture fields have the signature of the presence of shallow groundwater, a pattern superimposed on those due to soil physical properties or climatic conditions. The interaction with groundwater reduces soil moisture seasonality. Upward fluxes from the water table have a larger impact on soil moisture in water scarce seasons (spring and summer). This effect was also found by other studies with LEAFHYDRO and other models accounting for groundwater influence (e.g., Miguez-Macho et al., 2007; Decker, 2015).

The water table depth shows generally strong seasonal and interannual persistence, responding to long-term climatic conditions but not immediately to seasonal or annual highs and lows in precipitation. Over shallow water table regions, this water table persistence modulates soil moisture long-term evolution. Soil moisture memory is bimodal; soil moisture "remembers" past wet conditions through interaction with a shallow water table, buffering drought effects, and on the other hand, past dry conditions reflected in a depressed water table are passed on to soil moisture, delaying drought recovery.

The wetter soil induced by the proximity of a shallow water table results in higher ET during the dry growing season, due to higher water availability for vegetation transpiration. The spatial patterns of this ET enhancement over the studied region resemble those of shallow water table, where there is also strong groundwater-soil coupling. On average over the Iberian Peninsula, our model experiments estimate a 17.4 % (0.24 mm day$^{-1}$) increase in ET attributable to groundwater. ET maximum enhancement occurs in summer (34.9 %; 0.54 mm day$^{-1}$). We find the largest impact over the drier southern basins, where ET is water limited: 21.4 % yearly ET enhancement in the Guadalquivir basin and 28.4 % in the Segura basin. The northern Miño/Minho basin, where ET is more energy-limited than water-limited, presents the lowest groundwater impact on ET, of 13.3 %. In terms of time evolution, the influence of the water table on ET follows the trends of groundwater and soil moisture, which, as discussed previously, show significant persistence from year to year. Therefore, ET enhancement from groundwater "remembers" past dry and wet years and is decoupled from current climate conditions. This result of transpiration

enhancement from the groundwater reservoir and its lateral convergence has been reported by the use of other groundwater-land surface coupled models (e.g., Maxwell and Condon, 2016).

Groundwater sustains dry season streamflow. Wet season infiltration recharges groundwater, which is later on drained out by the river network during the dry season where the water table is above riverbed. When comparing with river flow observations, this seasonal behaviour improves on the results from the experiment without groundwater, in which rivers dry out in summer. Notwithstanding, the model produces in general an overly constant river flow and peak events are smoothed out. This problem is related to deficiencies in formulating the connection between groundwater and rivers where the water table in the cell is deep, since subgrid variability to represent riparian and valley zones where rivers and groundwater are in contact, is not considered.

In shallow water table areas, a declining trend in the water table, as the one found over the Ebro basin (see Fig. 14), would be partially sustaining ET; however where the water table is deeper, the lowering groundwater store is sustaining streamflow, explaining why there is more water in the annual mean total river flow in the Ebro in the WT run (Fig. 16). This issue is common to the other Mediterranean basins, more affected by the drought.

Our results show significantly wetter soil and enhanced ET over shallow water table regions. Here, we have supported this results with the validation of water table depth positions across the Iberian Peninsula. These results suggest that groundwater might have a sizable impact on climate over the Iberian Peninsula. For instance, it can enhance convective precipitation through local moisture recycling or lead to summer cooling lowering sensible fluxes and producing more cloudiness. Coupled land hydrology-climate models are needed to elucidate this question. We stress that this line of research might find similar results if a dynamic groundwater model is applied in other semi-arid regions of the world, which cover around 15 % of the global land area (EMG, 2011).

## 6   Summary and conclusions

In this paper, we have studied the influence of groundwater on soil moisture distribution and memory, ET fluxes and surface waters over the Iberian Peninsula. We used the LEAFHYDRO Land Surface Model, which represents the water table inter-actions with the unsaturated soil above and rivers, and lateral groundwater flow. We performed 10-year simulations with the groundwater scheme activated (WT run), and without groundwater, using a free-drain lower boundary condition for the soil column (FD run). The initial state of water table depth in the WT run is an equilibrium value calculated using a groundwater model (Fan et al., 2007, 2013) that finds a balance between the vertical groundwater recharge, driven by climate, and the lateral groundwater divergence, driven by topography.

We have shown that LEAFHYDRO is a solid tool to assess the groundwater-land surface link. Validation with observational water table depth data shows that the model simulates a realistic water table distribution, with shallow groundwater in valleys and deeper under hilltops. The water table seasonal evolution and longer-term trends are also well captured, particularly over shallow water table regions ($wtd \leq 8$ m).

We estimated the annual climatology net recharge and its mean seasonal cycle, and identified areas of strong groundwater-land surface coupling as those where the net flux from the water table is upward. Annual net groundwater-soil fluxes can

be positive (upward flux) to meet ET demands where gravity-driven lateral groundwater flow and river sipping represent a groundwater source in addition to infiltration. In the Iberian Peninsula, this occurs markedly in some river valleys, especially in those with strong groundwater convergence and in drier climates. Upward capillary fluxes can also dominate, albeit not so pronouncedly, over extensive interior semi-arid regions of shallow water table, where precipitation is insufficient to meet ET demands and there is lateral groundwater flow from neighboring mountains. Seasonally, groundwater-soil coupling is strong in spring, when ET is high, maximal in summer, when ET demands are even higher and precipitation is at a minimum, moderate in autumn, when precipitation mostly covers ET, and minimal in winter, which is the season of highest precipitation and lowest ET.

We have shown direct groundwater influence on land surface hydrology; controlling soil moisture distribution, providing strong seasonal and interannual memory to the soil wetness that ultimately reflects in enhanced ET fluxes during precipitation scarcity periods, and sustaining the dry season streamflow.

*Code and data availability.* This study uses LEAFHYDRO, which is a Land Surface and Groundwater model developed from the LEAF v2 LSM (see Section 2.1). The atmospheric forcing dataset ERA-Interim is available after registration with the European Centre for Medium-Range Weather Forecast (ECMWF) here: http://ecmwf.int/research/era/. The forcing IB02 precipitation dataset and the site water table and river flow data used for validation were collected by the authors via request to references given in Sections 2.3 and 3. LEAFHYDRO output data for this work are available from the corresponding author upon request.

*Competing interests.* The authors declare that they have no conflict of interest.

*Acknowledgements.* This work has been supported by the Spanish Department of Education and Science and the European Union Seventh Framework Programme under grant agreement no. 603608, "Global Earth Observation for integrated water resource assessment": eartH2Observe. The authors would like to thank and acknowledge the data providers: European Centre for Medium-Range Weather Forecast (ECMWF) for the forcing ERA-Interim dataset; the University of Cantabria (Spain) and the Portuguese Meteorological Institute (IPMA) for the IB02 analysis precipitation dataset; the United States Geological Survey (USGS) for the river routing parameters; the Institute of Geology and Mining of Spain (IGME), several Confederaciones Hidrográficas (Spanish agencies managing the main basins within the country) and the National Information System for Hydrological Resources of Portugal (SNIRH) for the sit observations of water table depth; and the Spanish Centre for Hydrographic Studies (CEH) for the river flow gauge data.

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

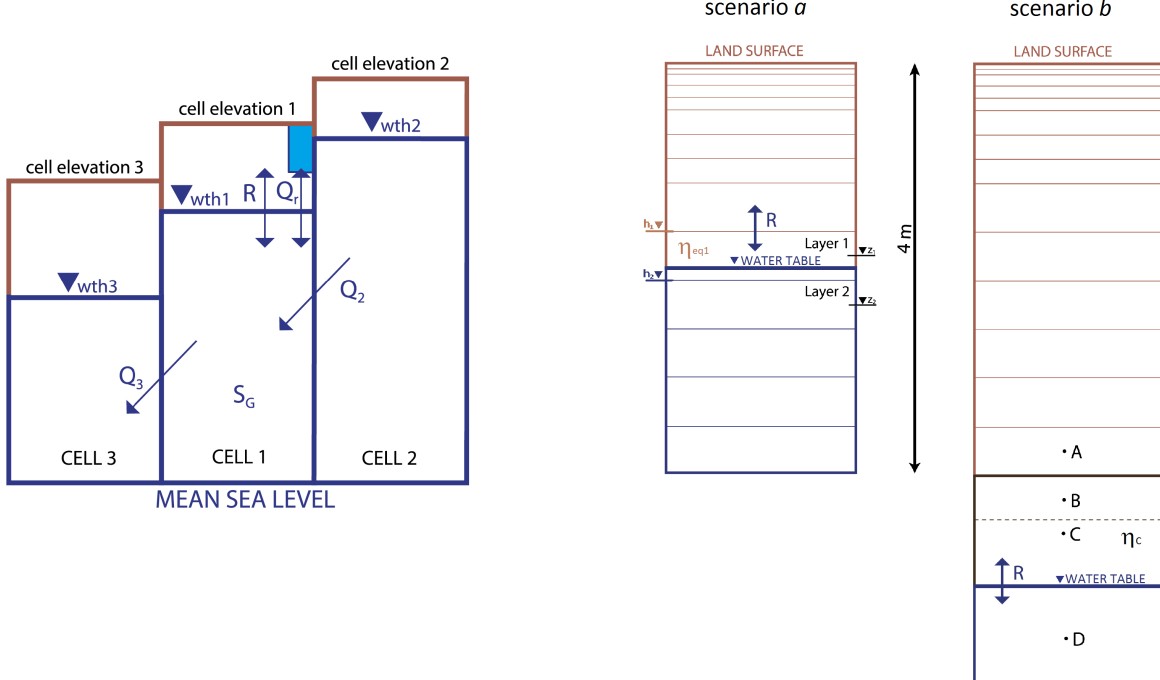

**Figure 1.** Left: LEAFHYDRO groundwater balance in a model cell (cell 1). Right: LEAFHYDRO double scenario to calculate the water flux through the water table ($R$).

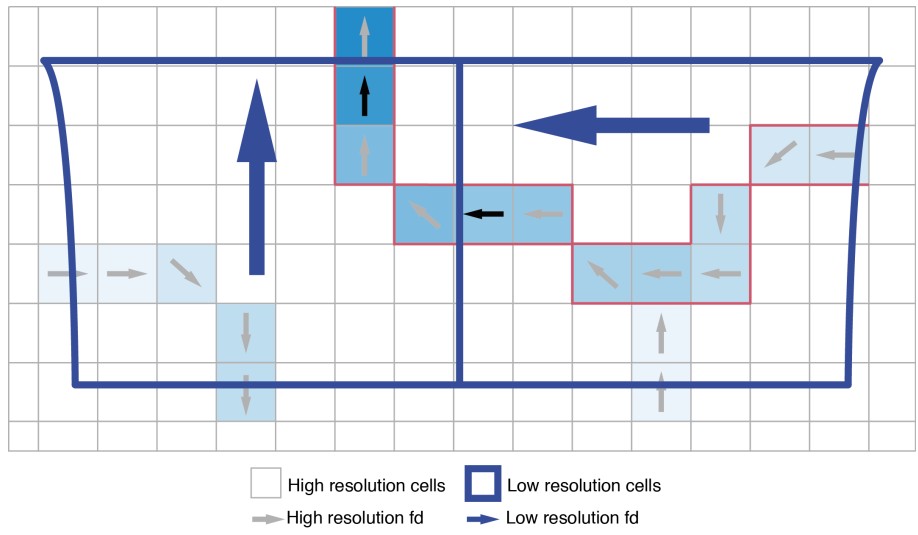

**Figure 2.** Sketch for the methodology to calculate river parameters from the HydroSHEDS high resolution database to the 2.5 km grid domain in LEAFHYDRO.

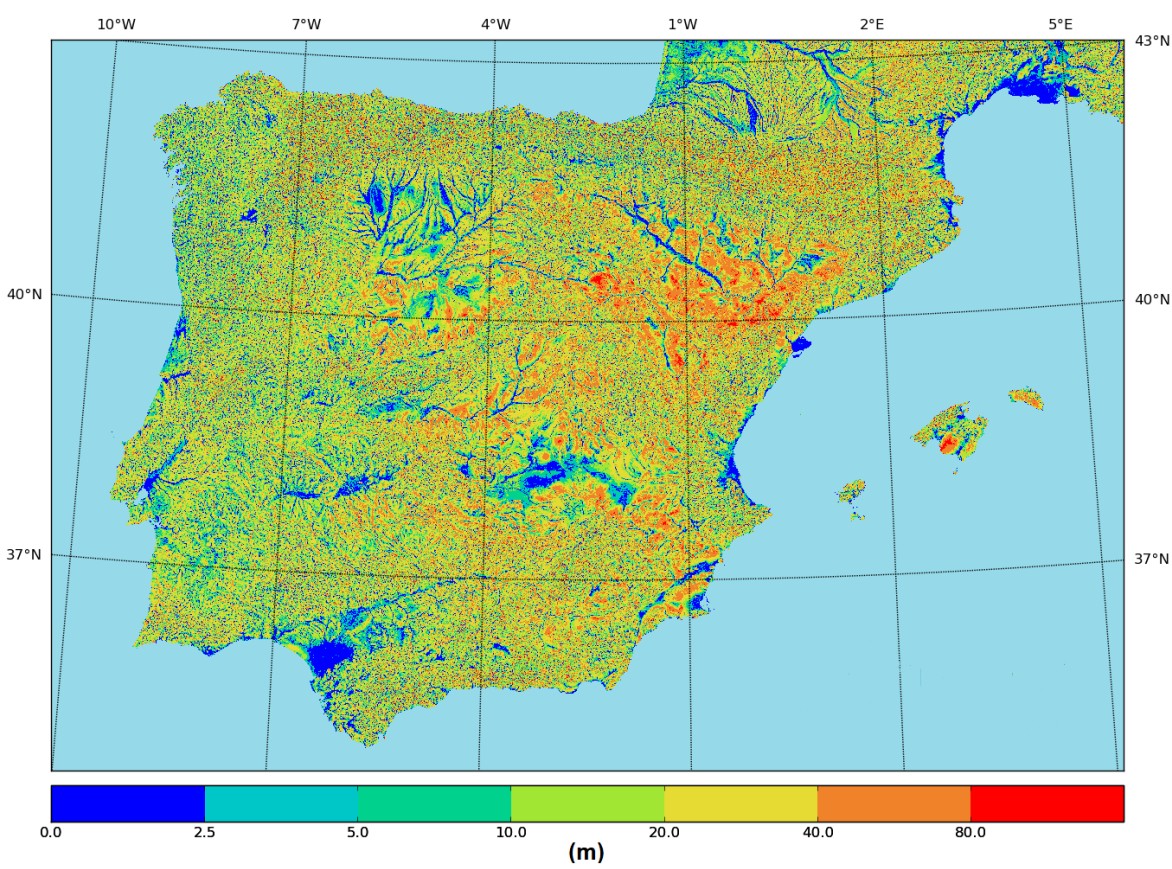

**Figure 3.** Iberian Peninsula Equilibrium Water Table Depth (m). The spatial resolution comes from topography data used for the Iberian Peninsula EWTD calculation (9 arc second; ∼213x278 m at 40°N). This EWTD was validated with 2601 observation points (Gestal-Souto et al., 2010).

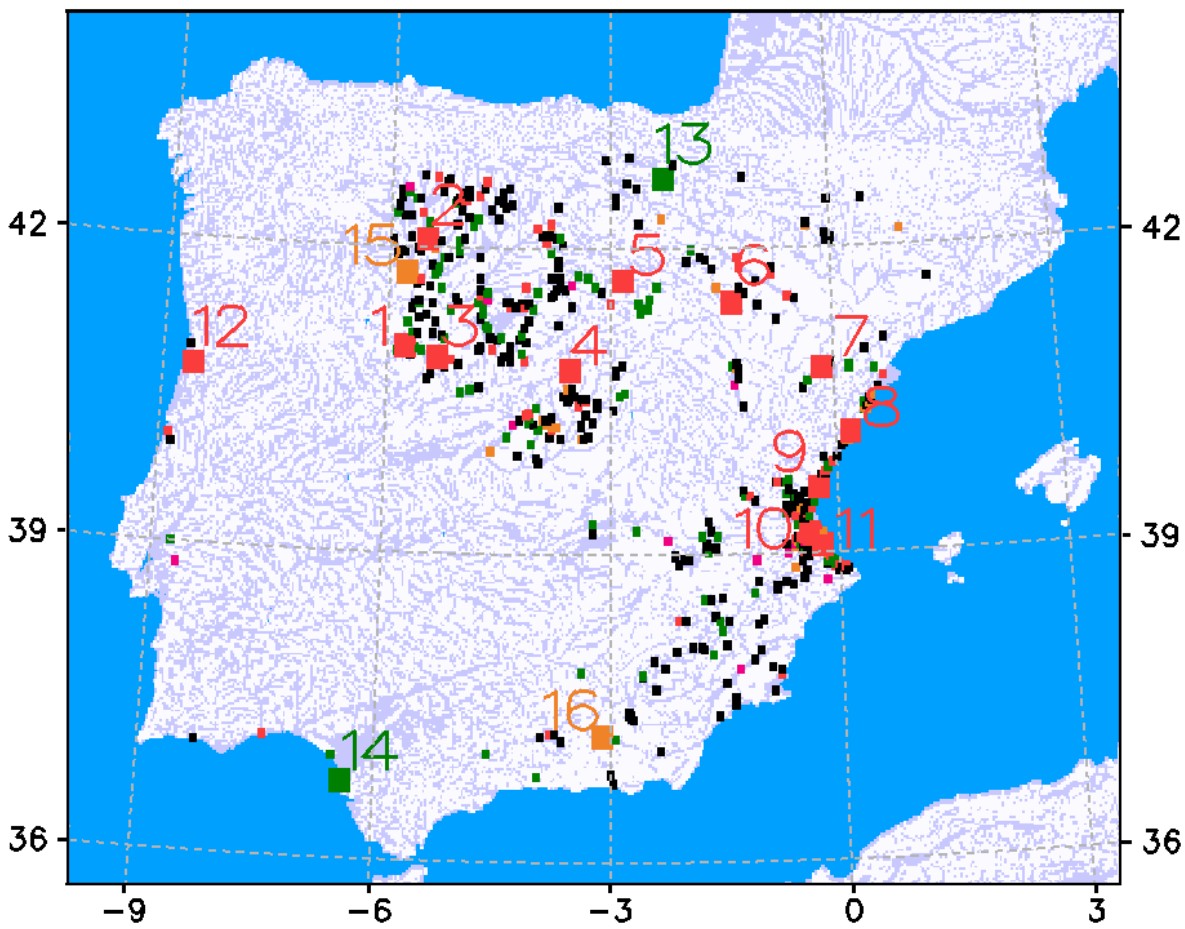

**Figure 4.** Shallow water table zones (light blue shades) and Iberian Peninsula $wtd$ observation stations (dots). Red dots are locations where observed and simulated $wtd$ differences are within 2 m; green dots are stations with correlation over 0.5 between observed and simulated $wtd$ series (full time series available in the observed data); purple dots are stations with steep $wtd$ slope ($\geq$0.035 m month$^{-1}$), well captured by the model; orange dots are cells containing more than one observation station; black dots are cells where none of the above criteria is met by the model. Over cells where more than one validation criteria is reached the point adopts the colour of the first criterium met (in the order presented here); for instance, cells with mean $wtd$ differences lower than 2 m and also correlations above 0.5, are shown as red on the map.

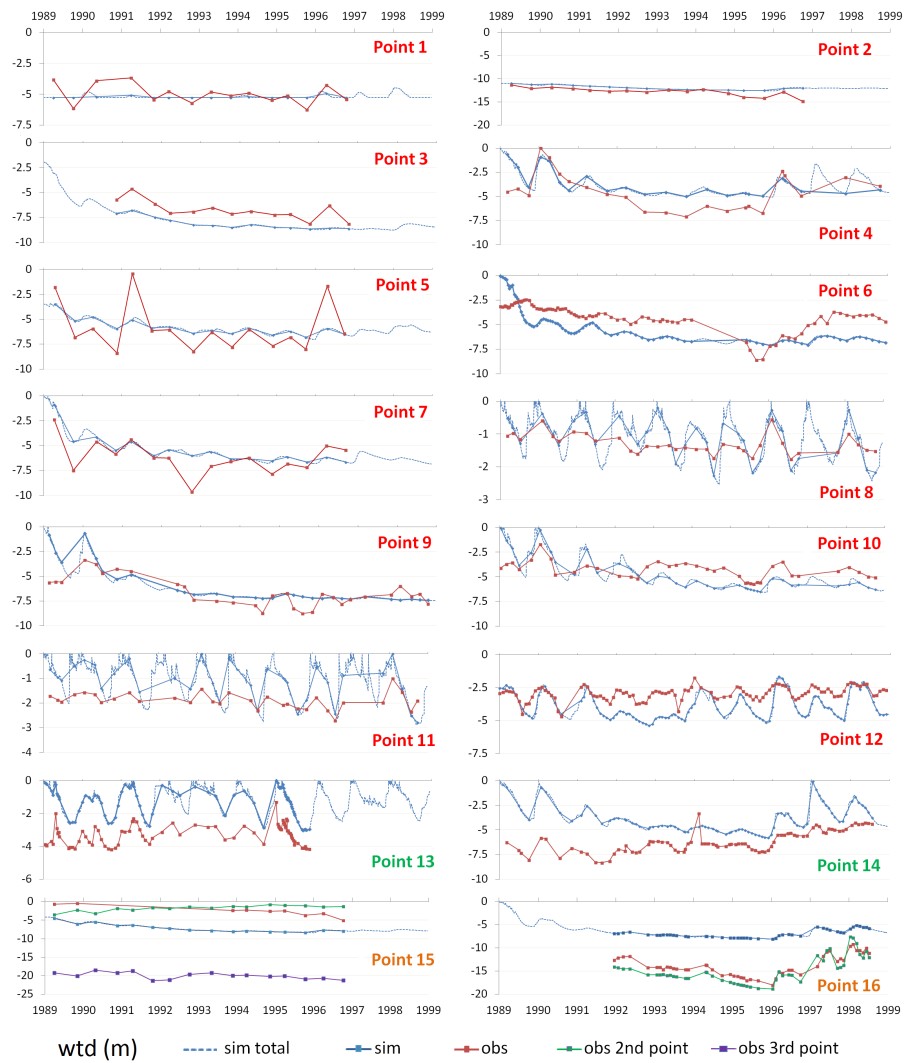

**Figure 5.** Water table depth (m) time series along the 10-year period at the stations numbered in Fig. 4 (1 to 16); observed (connected red dots), simulated at observation times (connected blue dots), simulated daily (dashed blue line), and observed at the second and third observation points within one model cell (connected green and purple dots, respectively).

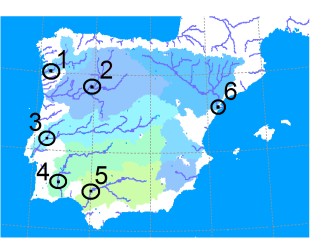

**RIVER FLOW STATIONS**

1 - Foz do Mouro, Miño/Minho river
2 - Puetepino, Duero/Douro river
3 - Almourol, Tajo/Tejo river
4 - Pulo do Lobo, Guadiana river
5 - Cantillana, Guadalquivir river
6 - Tortosa, Ebro river

--- observations    --- WT simulation

Monthly mean river flow (m³/s)    Monthly river flow (m³/s)

1  r=0.98

2  r=0.96

3  r=0.91

4  r=0.71

5  r=0.66

6  r=0.94

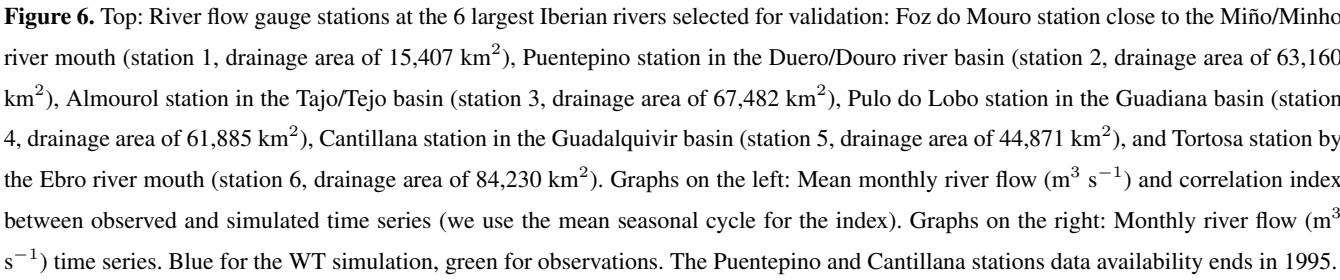

**Figure 6.** Top: River flow gauge stations at the 6 largest Iberian rivers selected for validation: Foz do Mouro station close to the Miño/Minho river mouth (station 1, drainage area of 15,407 km$^2$), Puentepino station in the Duero/Douro river basin (station 2, drainage area of 63,160 km$^2$), Almourol station in the Tajo/Tejo basin (station 3, drainage area of 67,482 km$^2$), Pulo do Lobo station in the Guadiana basin (station 4, drainage area of 61,885 km$^2$), Cantillana station in the Guadalquivir basin (station 5, drainage area of 44,871 km$^2$), and Tortosa station by the Ebro river mouth (station 6, drainage area of 84,230 km$^2$). Graphs on the left: Mean monthly river flow (m$^3$ s$^{-1}$) and correlation index between observed and simulated time series (we use the mean seasonal cycle for the index). Graphs on the right: Monthly river flow (m$^3$ s$^{-1}$) time series. Blue for the WT simulation, green for observations. The Puentepino and Cantillana stations data availability ends in 1995.

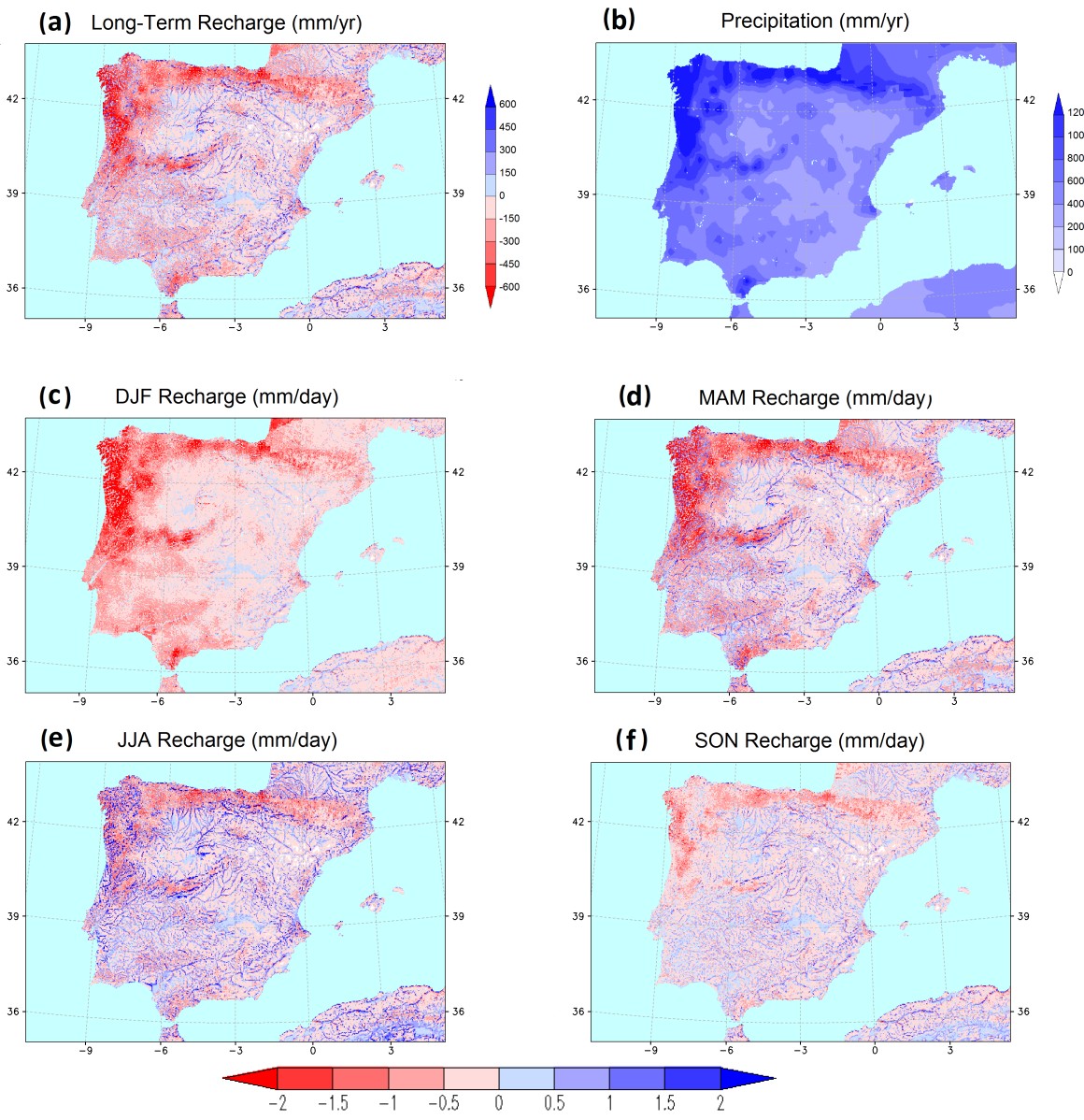

**Figure 7.** (a) Long-term recharge (mm yr$^{-1}$), defined as net moisture flux at the water table. (b) Mean precipitation (mm yr$^{-1}$). From (c) to (f): Mean seasonal recharge (mm day$^{-1}$) for winter (DJF), spring (MAM), summer (JJA) and autumn (SON). In the recharge plots, red colours indicate negative (downward) recharge and blue colours correspond to positive (upward) flux. All values are calculated for the 10-year simulation period

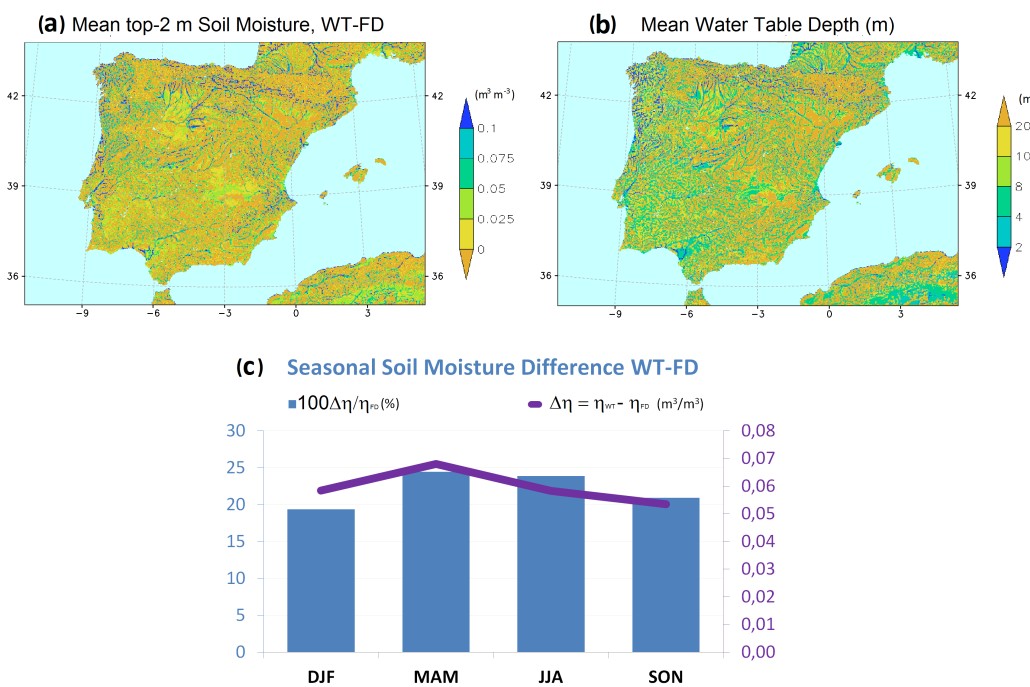

**Figure 8.** (a) Mean top-2 m soil moisture difference (WT-FD; volumetric water content, m$^3$ m$^{-3}$) . (b) Mean $wtd$ (m). (c) Seasonal top-2m soil moisture differences between the experiments with and without groundwater (WT-FD), averaged over the Iberian Peninsula shallow water table regions ($wtd \leq 8$ m): percent of soil moisture increase (%; blue columns) and soil moisture absolute difference (volumetric water content, m$^3$ m$^{-3}$; purple line).

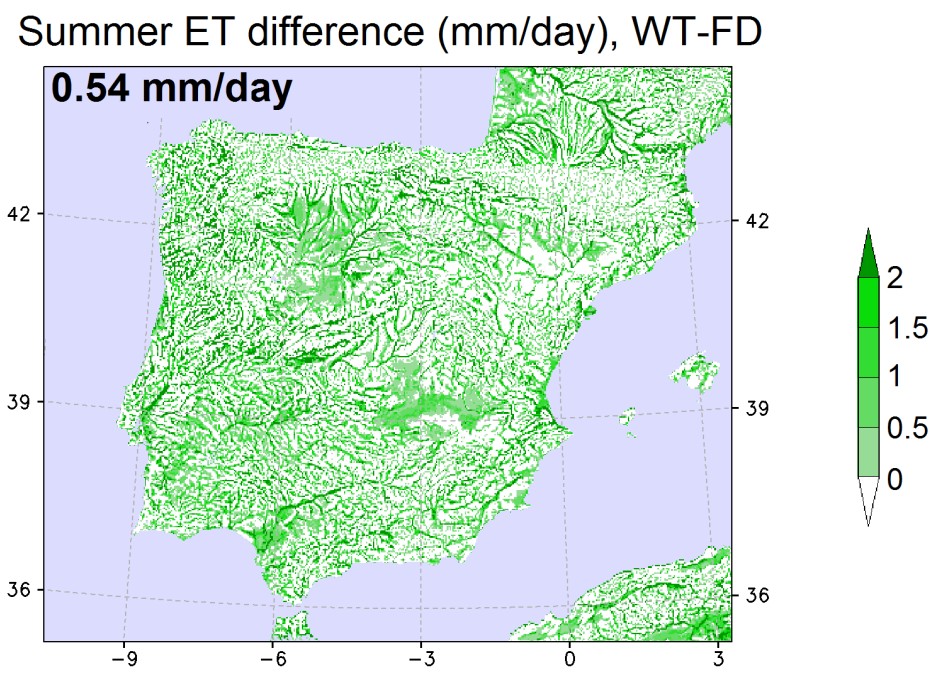

**Figure 9.** Mean summer (JJA) ET difference (mm day$^{-1}$) between the experiments with and without groundwate (WT - FD ) for the 10-year simulation period, and averaged value over the Iberian Peninsula (black text in top-left corner).

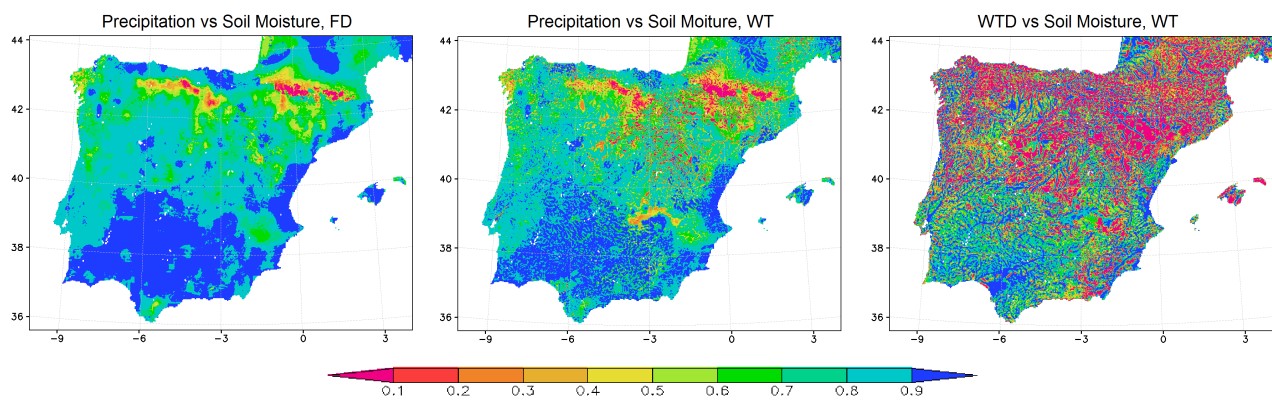

**Figure 10.** Correlation maps of the yearly anomaly time series for the Iberian Peninsula along the 9 complete hydrological years simulated. Left: between precipitation and soil moisture in the free-drain (FD) run. Centre: between precipitation and soil moisture in the groundwater (WT) run. Right: between $wtd$ and soil moisture in the grounwater (WT) run.

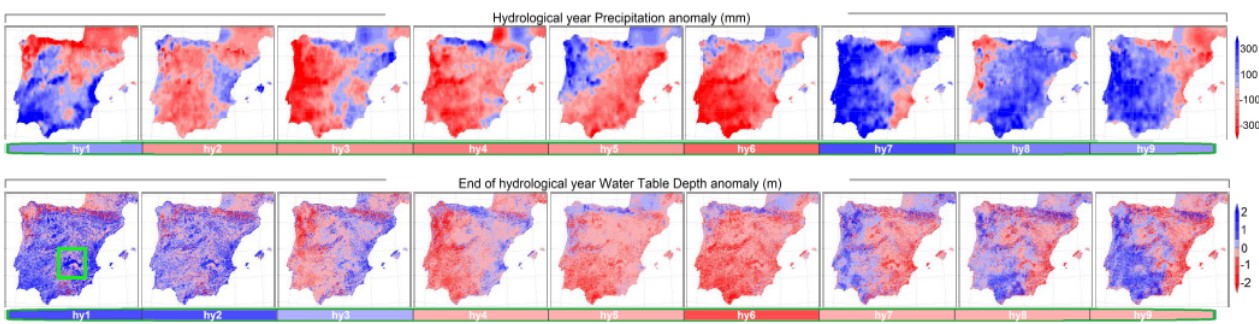

**Figure 11.** Hydrological year anomaly plots. Each column corresponds to a complete hydrological year (hy1 to hy9). Top row: total yearly precipitation anomalies (mm). Bottom row: end of hydrological year (September 1st) $wtd$ anomalies (m). Colour bars below each plot represent the averaged anomaly value for the Iberian Peninsula.

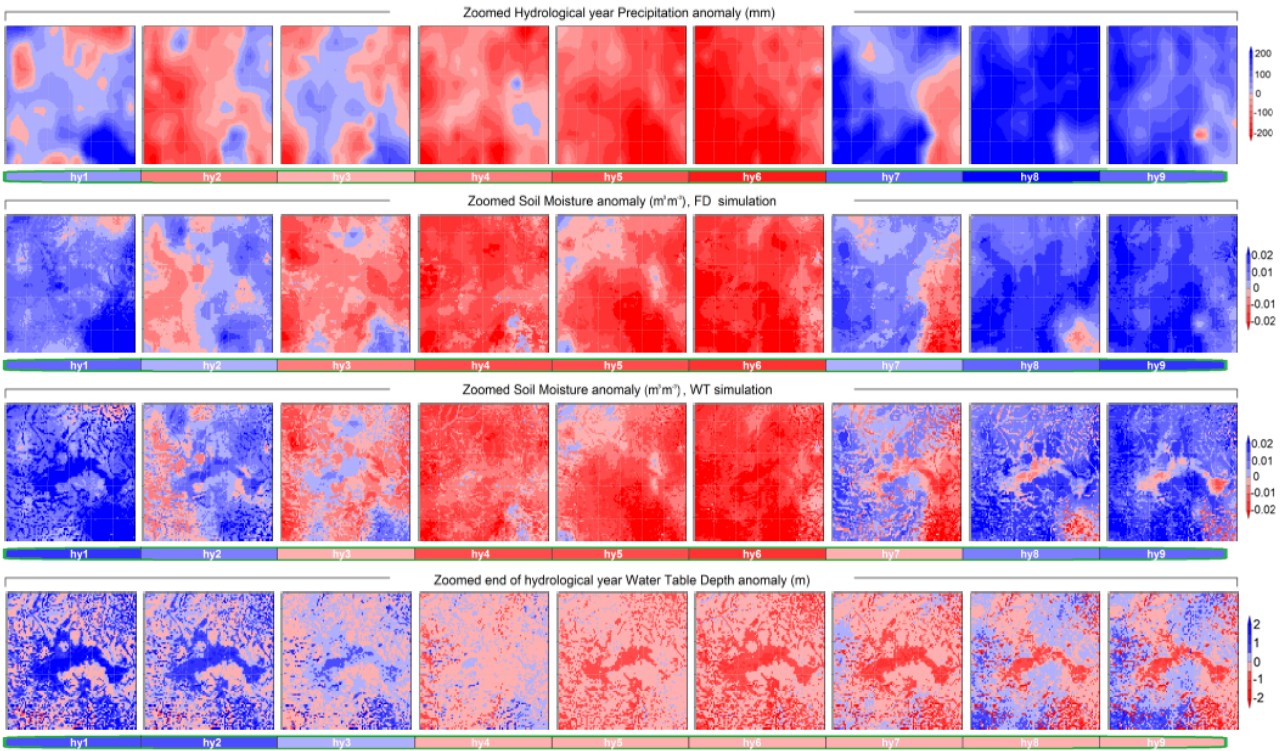

**Figure 12.** Zoomed hydrological year anomaly plots in the 250x225 km$^2$ region highlighted in light green in Fig. 11 (bottom row, first plot), containing La Mancha Húmeda, approximately between 38º and 40.2º latitude and -4.5º and -1.5º longitude. Each column corresponds to a complete hydrological year (hy1 to hy9). Rows from top to bottom: total yearly precipitation anomalies (mm), top-2 m soil moisture anomalies (m$^3$ m$^{-3}$) in the FD run, top-2 m year soil moisture anomalies (m$^3$ m$^{-3}$) in the WT run, and end of hydrological year (September 1st) $wtd$ anomalies (m). Colour bars below each plot represent the averaged anomaly value for the zoomed area.

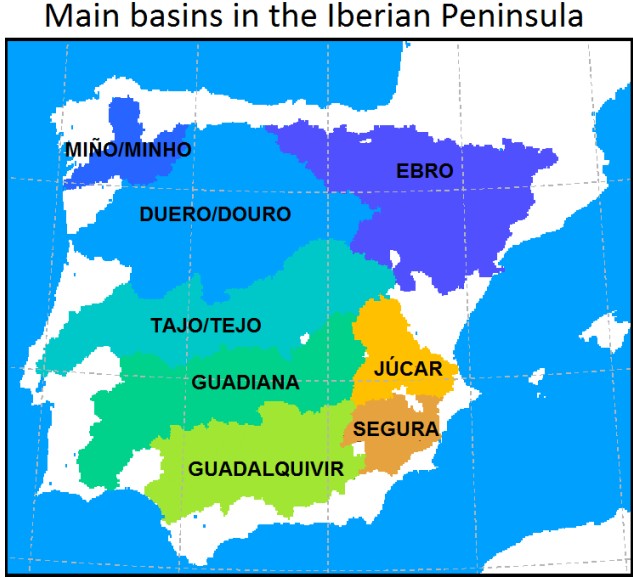

**Figure 13.** Main river basins in the Iberian Peninsula.

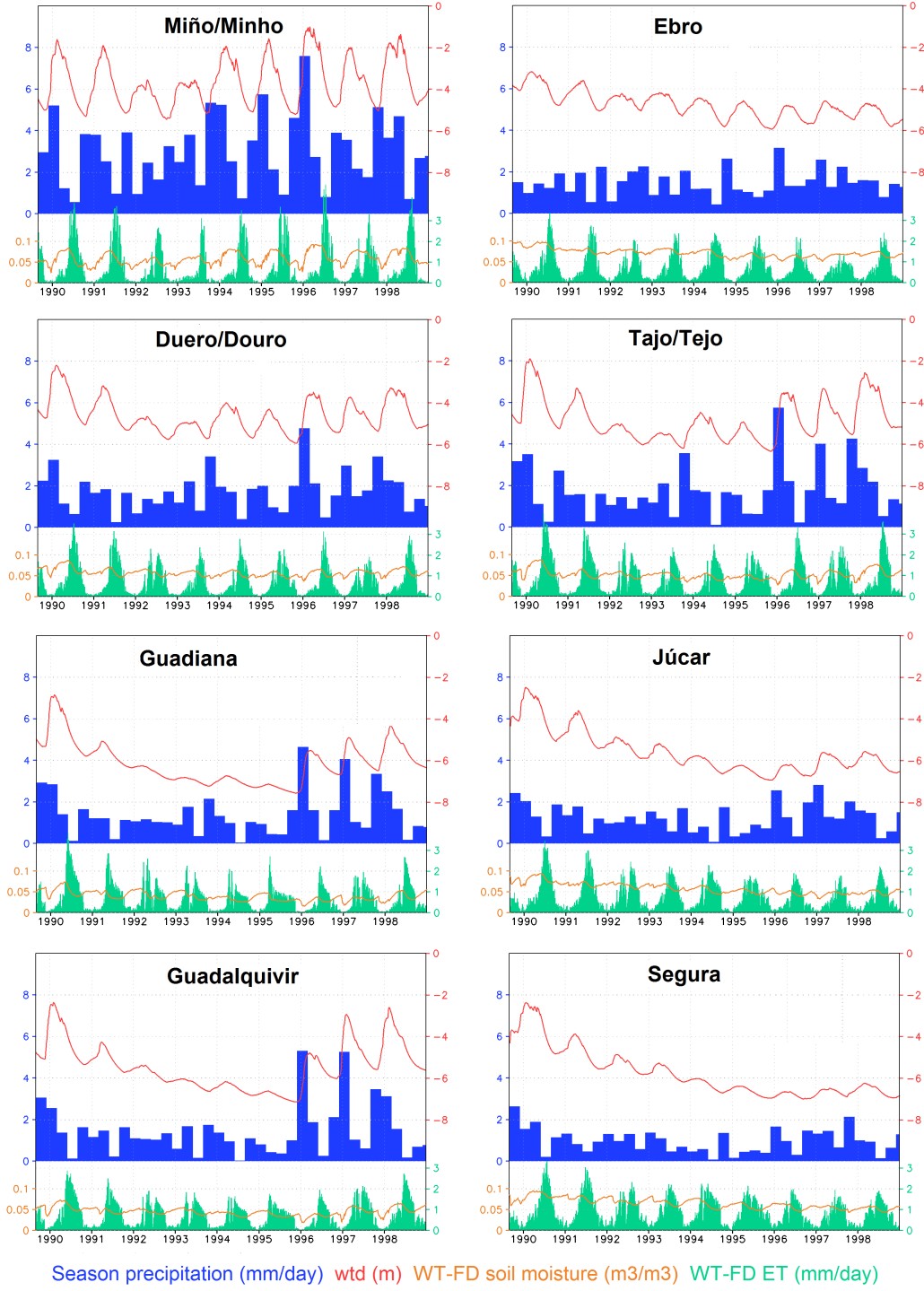

**Figure 14.** Results for the river basins in Fig. 13. Time series of seasonal precipitation averaged over the entire basin (mm day$^{-1}$; blue bars), and averages over shallow water table cells only ($wtd \leq 8$ m) of $wtd$ (m; red line), WT-FD top-2 m soil moisture difference (m$^3$ m$^{-3}$; orange line) and WT-FD ET difference (mm day$^{-1}$; light green bars).

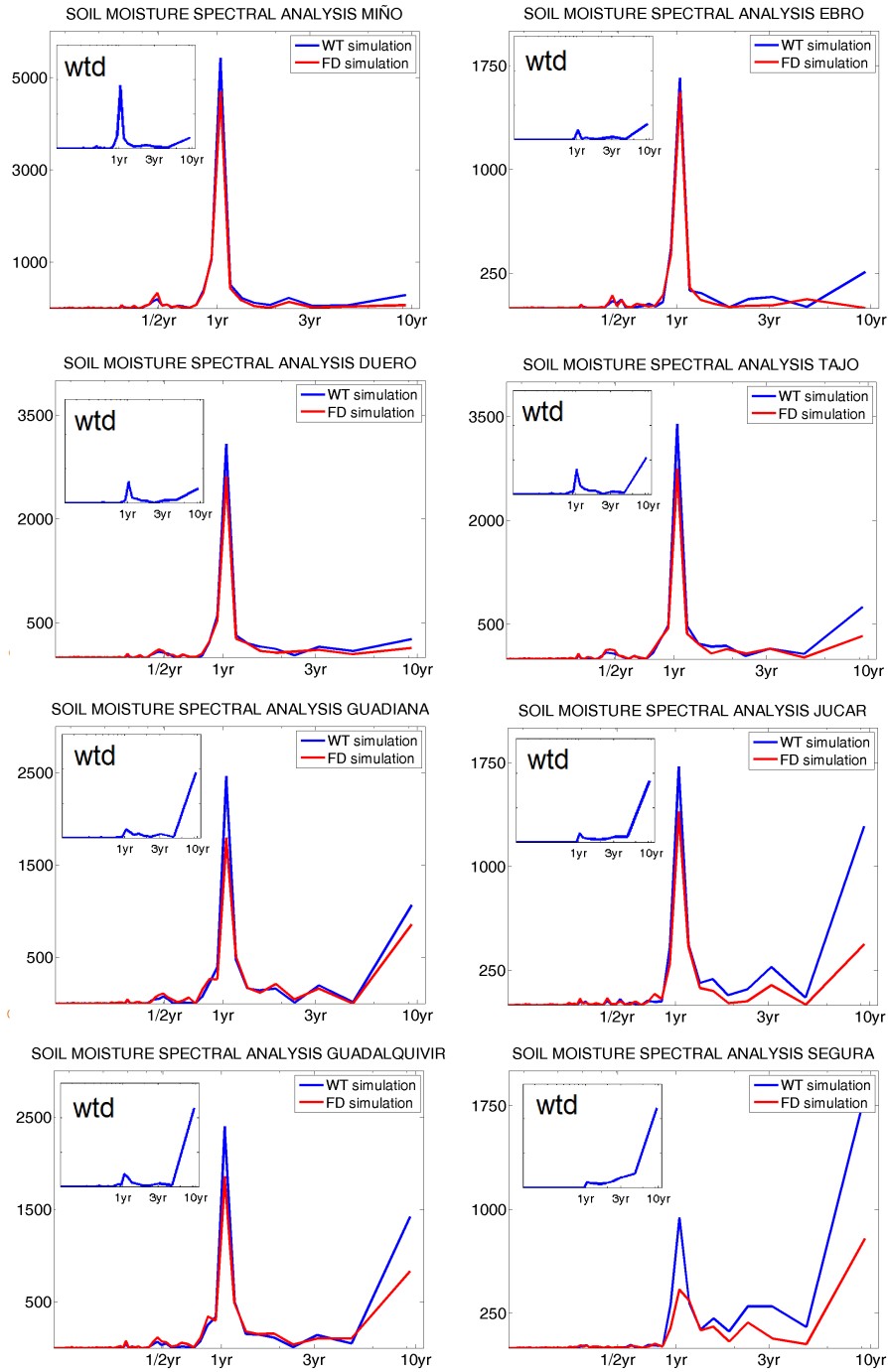

**Figure 15.** Power spectrum analyses over the main Iberian basins of top-2 m soil moisture (WT run in blue and FD run in red) and $wtd$ (insets). Only shallow water table cells ($wtd \leq 8$ m) within the basin are used. Basins are ordered, as in Fig. 14, from north (top) to south (bottom) and those on the left drain to the Atlantic and on the right to the Mediterranean, except for the Tajo, which is also an Atlantic basin.

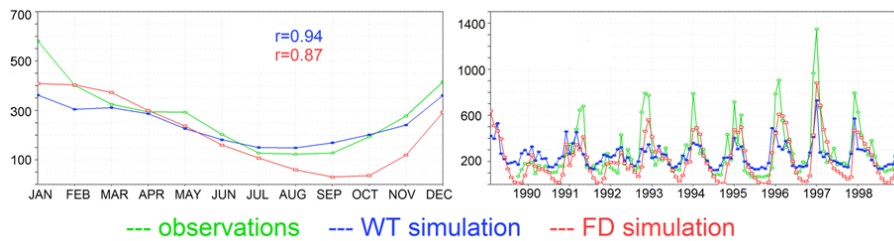

**Figure 16.** Modelled and observed river flow for the Ebro station 6 in Fig. 6. Left: Monthly mean river flow (m$^3$ s$^{-1}$) and correlation indexes between the observed and simulated time series (we used the mean seasonal cycle for the index). Right: Monthly river flow (m$^3$ s$^{-1}$). Blue for the WT simulation, red for the FD simulation and green for observations.