# Peer review of "Groundwater influence on soil moisture memory and land-atmosphere fluxes in the Iberian Peninsula"

_Hydrology and Earth System Sciences, 2018_

## Referee Comment (RC1) · Anonymous Referee #1 · 19 Feb 2019

General comments

This manuscript assesses the groundwater influence on soil moisture memory and land-atmosphere interactions in the Iberian Peninsula by using the LEAFHYDRO model. The simulation was performed at 2.5-km over the Iberian Peninsula for a 10 year period. The authors found significantly wetter soil and enhanced ET over shallow water table regions suggesting that groundwater might have an impact on climate over the Iberian Peninsula.

This study follows two previous studies carried out in United Sates (Miguez-Macho et al., 2007) and over the Amazon basin (Miguez-Macho and Fan, 2012) where the

same model was used to depict the influence of groundwater on soil moisture and atmospheric variables. The methodology and science questions of the present paper are very similar to these two previous papers but applied for the Iberian Peninsula. Regarding the main conclusion obtained from the present paper, most of them are consistent and confirm the findings in numerous previous studies, including the two previous studies using the LEAFHYDRO model. However, no significant new findings can be drawn from this modeling, hence the novelty cannot be said to be high. In my opinion, this is the first major issue of the paper. The authors should consider to better highlight what is the interest of using such high-resolution model over Spain with respect to the previous study using the same model (in United Sates and Amazonia). A reorganization of the introduction may help to better define the novelty introduced by the use of LEAFHYDRO over the Iberian Peninsula.

The paper is articulated in five sections: introduction, methodology, validation, results, and conclusion. Regarding this structure, I identified two general remarks that need to be solved. First, the methodology section do not give enough details on the model description and the data used. In my opinion, while the main purpose of the paper is related to groundwater-surface land relationships, this part is not enough detailed in the paper. Secondly, the results section introduces some elements of discussion that are not at all linked with the literature. No references are cited, neither in this results section, nor in the conclusion. Regarding the bunch of paper related to this subject (i.e. groundwater-soil moisture influence), the paper lacks of references. This is the second major issue.

Besides these general comments, I identified specific comments and technical errors in the text and in the figures that I put in comment in the subsequent sections. In particular, I wish to see all the figures wider, regarding the size of the simulated area.

Based on the above statement, I think major revisions are needed to solve the two previous major issues and the below specific comments and errors before the paper can be eventually published in HESS.

Specific comments

Generally speaking, and regarding the bunch of papers on the subject, the Introduction part lacks of references on land surface-atmosphere coupling and soil moisture memory influences on groundwater and atmosphere. As an example the following papers should be considered:

Maxwell, R. M., Lundquist, J. K., Mirocha, J. D., Smith, S. G., Woodward, C. S. and Tompson, A. F. B.: Development of a Coupled Groundwater–Atmosphere Model, Mon. Wea. Rev., 139(1), 96–116, doi:10.1175/2010MWR3392.1, 2010.

Vergnes, J.-P., Decharme, B. and Habets, F.: Introduction of groundwater capillary rises using subgrid spatial variability of topography into the ISBA land surface model, J. Geophys. Res. Atmos., 119(19), 2014JD021573, doi:10.1002/2014JD021573, 2014.

Page 2, line 4 to 5: This is the purpose of the paper. I suggest to move this part near the end of the introduction, after the definition of the science questions.

Page 3 line 21 to the end : "Here, we present a modelling study linking groundwater to soil moisture, land-atmosphere interactions and surface water" : You introduce the purpose of the paper in the first sentence and then explain why you chose your case study. To better highlight the subject of the paper and enhance the problematics that occurred in Spain and the opportunity to simulate groundwater and soil moisture at this scale, you should consider to move all this part before introducing the purpose of the paper.

Page 4 line 9: "Model description and settings". This part describes the model and data used in the study. Most of the formulation of the model's equations are described in (Miguez-Macho et al., 2007) and (Fan et al., 2007), so only the mass balance of the dynamic groundwater reservoir is given here. However, it could have been useful to have a description of how the water table head is calculated since it the main variable that is evaluated in the following sections. Information on how hydrodynamic parameters
(transmissivity and porosity) are taken into account in the model could be added.

Page 4 Line 27 – Page 5 line 7: The coupling of the water table and the soil layers is unclear. Why the layer B is added? This part needs more details on how the water content is computed.

Page 5 line 7-14 : This part lacks of details about the calculation of the river-groundwater exchanges. It is the so-called river conductance model used in MOD-FLOW? Are river heights variables (using Manning's Formula) or prescribed? How are the river conductances determined?

Page 5, line 15 "2.2 Initial land and river parameters" Regarding the title, should it be "Land-surface and river parameters"? Generally speaking, this part lacks of many details about the parameters used for developing the model over the Iberian Peninsula. The authors should consider the following remarks and maybe add a Figure depicting the case study.

Page 5 line 16-20: Why the soil textural classes are needed in LEAFHYDRO? What is the dominant soil type/vegetation type?

Page 5 line 21-24: How does the river flow scheme work ? Does it used Manning's Formula? How are the river widths determined? This part lacks of details for the Iberian Peninsula.

Page 6 line 4: Could you add details about the method used to disaggregate the IB02 data using the ERA-Interim precipitation data? This is not clear how the link between the two of them is described.

Page 6 line 3: You speak about the model grid without having define his resolution before (0.2° ?).

Page 6 line 12-17: What is the resolution of the model grid? How was the global climatic recharge at low resolution used? Was it disaggregated at a higher resolution? Is it an annual mean average over a period? How was the test run aggregated to the

model grid? This part is not clear.

Page 6 line 25-28: Much details are needed on how soil moisture is calculated in LEAFHYDRO. This remark is linked to the model description in section 2.1. Some details could be added to illustrate soil moisture.

Page 6, line 29: 10 year is a rather short period to validate the model. I know some water table characterized by multi-year annual cycles of 20 years. Could you explained why you choose this time period? What is the time step?

Page 7 line 24: "in order to rule out measurements in confined aquifers as much as possible": does it means that you used some observations of confined aquifers to validate unconfined aquifers? It should be clarified.

Page 8 line 3 : "With regard to the observations, 203 of the studied stations present a shallow water table (wtd < 8 m) during the simulation period": does it mean that the mean water table depth is lower than 8 m?

Page 8 line 10 : Do these 3 different observation sites in Point 15 grid cell belong to the same aquifer or maybe to different layers? Coarse spatial resolution is a factor that could explain these differences, but the different piezometers can also monitored different aquifers. It should be verified. The same remarks applied for the other points with several observations.

Page 8 line 14-line 17: The presentation of these percentage need to be clarified.

Page 8 line 17: "capturing the mean water table depth" : is it rather "capturing the water table depth time evolution"?

Page 8, line 26-27: this statement should be better connected with the results.

Page 10, line 14-16: Recharge mean annual cycles is linked to ET and precipitation mean annual cycles, but Figure 6 only shows the climatology of the recharge variable. Results for ET and precipitation should be mentioned here, maybe in the Figure, or

with some details in the text.

Page 10, line 19-23: "As the water table gets deeper": does it correspond to the EWTD of Figure 2 ? Or a time evolution ? It must be clarified.

Page 10, line 22: ET evolution is mentioned but no Figure show it.

Page 11, line 24: Anomalies are computed with respect to the annual mean or the mean annual cycle ? It must be clarified in this subsection.

Page 12, line 29: The authors should add a sentence on the location of this region (reference to Figure 10).

Page 13 line 24: "but drainage is slowed down". This result need to be reinforced with further results, maybe with a water budget or a time evolution of the recharge.

Page 14 line 15: "one year frequencies and at decadal timescales". Decadal timescale appear on these power spectrum analysis, but I wonder the pertinence of finding decadal timescale with a 10-year time series. A period of at least 20 years would have been more appropriate.

Page 14, line 15-16: "The annual cycle, linked to that of the surface water balance": Could you better explain this statement ? Maybe by linking it with previous results ?

Page 14, line 24-25 : "The higher weight of longer timescales of variation in the WT soil moisture series": same remark as above. A 10-year simulation appear rather short to establish this result.

Page 14, line 28: For this section, Figure15 is not necessary since it is the same as Figure 5. You should consider to had the FD simulation in Figure 5 directly. Figure 15 could be replaced by a Figure showing the stream-groundwater exchanges in order to discuss this flux.

Page 23, Figure 2: The legend refers to EWTD and topographic data, but only one map is shown that corresponds to EWTD. Why describing topographic data here ?

[Figure]

This Figure could be wider and extend to the full wide of the page. Add a unit to the colorbar and a title.

Page 24, Figure 3: The authors describe a grid centered in the Iberian Peninsula. Figure 3 shows this peninsula, but also parts of France and North Africa. Could you add the limits of the simulated domain ? Are the France and North Africa part also simulated ? Generally speaking, Figure 3 should be reorganized to better highlight the results. A wider map centered on Spain could improve the reading. Using different color points for different information on the same map is confusing. I suggest to use different maps for the different informations (wtd, correlation, steep, number of station par cells) and grouping them into a single figure.

Page 25, Figure 4: Point 8 and Point 11: the model seems to overestimate the amplitude of the piezometric head evolution. It should be mentioned and explained.

Page 26, Figure 5: only correlation scores are given. The Nash-Sutcliffe (Nash and Sutcliffe, 1970) score could be used and commented in the text to quantify the quality of the simulation.

Page 29, Figure 7: add units for the maps and the y-axis of the bars. The (a), (b) and (c) letters need to be added to the titles.

Page 30, Figure 8: the authors should think to show the seasonality of ET in Figure 8, or elsewhere, as said earlier.

Page 34, Figure 12: add a title.

Page 37, Figure 15: suppress this Figure and add the FD simulation to Figure 5.

Technical corrections

Page 1, line 2 : "a key role" not "an key role"

Page 8, line 19 : "simulated time series" instead of "simulated series"

Page 13, line 1 : "for the large" instead of "for the the large"

References

Fan, Y., Miguez-Macho, G., Weaver, C. P., Walko, R. and Robock, A.: Incorporating water table dynamics in climate modeling: 1. Water table observations and equilibrium water table simulations, J. Geophys. Res., 112(D10), D10125, doi:10.1029/2006JD008111, 2007.

Miguez-Macho, G. and Fan, Y.: The role of groundwater in the Amazon water cycle: 2. Influence on seasonal soil moisture and evapotranspiration, J. Geophys. Res., 117(D15), D15114, doi:10.1029/2012JD017540, 2012.

Miguez-Macho, G., Fan, Y., Weaver, C. P., Walko, R. and Robock, A.: Incorporating water table dynamics in climate modeling: 2. Formulation, validation, and soil moisture simulation, J. Geophys. Res., 112(D13), D13108, doi:10.1029/2006JD008112, 2007.

Nash, J. E. and Sutcliffe, J. V.: River flow forecasting through conceptual models part I — A discussion of principles, J. Hydrol., 10(3), 282–290, doi:10.1016/0022-1694(70)90255-6, 1970.

---

## Referee Comment (RC2) · Anonymous Referee #2 · 26 Mar 2019

Manuscript hess-2018-626 by Martinez-de la Torre & Miguez-Macho: "**Groundwater influence on soil moisture memory and land–atmosphere interactions in the Iberian Peninsula**"

The LEAFHYDRO model was among the first ones to couple a physically-based 2D groundwater (GW) flow model with a land surface model (LEAF2). Transient applications so far focused on the USA (2007 papers), the Amazon (2012 papers), while the groundwater model has been coupled to Noah over South-America (Martinez et al 2016), and over New-Zealand (Westerhoff et al., 2018, not cited). The present paper reports the application of the LEAFHYDRO over the Iberian Peninsula (IB), with novel insights regarding the propagation of precipitation anomalies to water table depth (WTD) and soil moisture anomalies, at a pluri-annual timescale. This effect is all the more pronounced as the climate is more arid, as are the more "classical" impacts of GW on ET, soil moisture, and river discharge at seasonal or shorter timescales.

This makes this paper very commendable for HESS, but on the other hand, the paper lacks (i) good quality figures, (ii) sufficient information on the model and forcing datasets, (iii) solid quantification of the reported impacts and model validation, (iv) a real discussion of the results, including the limitations of the approach. Other problems include a structure that is not always very logic, and a tendency to overstatement. Apart from a few spelling errors, the language is very clear. **Eventually, I advise to substantially revise the paper before publication in HESS**. None of the suggested revisions is very complicated, but many of them are advisable, as detailed below.

1. **Figures:**
    (a) Most IB maps are too small, and it's almost impossible to see something. Please remove Southern France, Northern Africa and oceans, and magnify the remaining IB. When possible, please use the same color scale for comparable variables (e.g. equilibrium and non-equilibrium WTD). It would also be very informative to add the mean and std of each map over the IB.
    (b) Fig. 3 is too small as well, and the color code of the points locating the points with observed does not seem well adapted, since a point can meet several conditions: for instance, what is the color of a point where bias is less than 2 m (red), and correlation with observed time series is more than 0.5 (green), which must be possible? There also seems to be some black points, in which case their meaning should be explained. But maybe they are purple… It would also be useful to report the classification used for Fig. 3 on the various panels of Fig 4 (insert for each point the bias, the correlation coefficient, and the wtd slope).
    (c) For Fig3, Fig 5, Fig 15 (and potentially Fig4), it must be clarified if the reported correlation coefficient is calculated on the full time series (120 monthly values), or on the mean seasonal cycle (12 monthly mean values).
    (d) Fig. 6: R seems negative if downward, which seems odd for the flux which recharges the GW.
    (e) Fig. 7c: why not show a real mean seasonal cycle, with 12 monthly mean values, instead of 4 seasonal mean values? And couldn't you plot the seasonal cycle of the shallow WTD as well?
    (f) Fig. 8: the color scale is not clear, we cannot distinguish the values that are not zero. Besides, could you add a scatter plot of summer ET difference against summer WTD, to show if there is a kind a threshold WTD inducing a marked ET difference?
    (g) Fig. 9 is very noisy: could you add the difference between center and left panels?
    (h) Fig. 11: please add the lon/lat of the mapped area, either on the maps, or on the caption.
    (i) Consider merging top panel of Fig 5 and Fig12; same for Fig 5 and Fig 15. By the way, why not show the FD simulations at all stations? And correct the statement that Ebro at Tortosa is where the model exhibits the best scores (L32-33 p 14): based on correlation coefficient, this station is only the third best for simulation WT based on Fig 5; besides, the ms discusses two models, so clarification is needed. Finally, the correlation coefficient is far from enough to support a performance analysis, and I strongly recommend that other classical criteria are

documented (bias, important since ET changes between the two simulations; RMSE; Nash and/or KGE, which are classical skill criteria in river hydrology).

(j) Fig. 13: a full paragraph is devoted to analyzing the seasonal variations of the different variables (L21-31 p 13), but we cannot see them. Please add the mean seasonal cycle next to the 10-yr time series to support this discussion. It would also help to magnify the scale of SM differences. The caption says the precipitation anomalies are calculated over the entire basins, while the other anomalies are calculated in the fraction with shallow WT (ca 1/3): why not calculate them over the same domain, to avoid any doubts. Finally, the text says the WTD of Ebro and Segura recover after the pluri-annual central drought, but it is not discernible in the panels.

**2. Methods:**

The LEAFHYDRO model has already been published, but a paper needs to be self-consistent, and more info is needed on the parts that are relevant to the conclusions.

The recharge calculation, in particular, is far from being clear, at least to me, although I have looked for more information in Miguez-Macho et al. 2007, Part 2. This should be clarified in the article, and the following questions might help the authors:

(a) The calculation is different depending if the WTD is larger or not than the soil depth, but I couldn't understand scenario b with larger WTD. In this case, how are the water content of points B and C estimated? It's written the one of point C "is determined by mass balance from the fluxes above and below" (L5-6 p 5) but these fluxes also need to be estimated, and there seem to be too many unknowns: please clarify the system, including flux equations, boundary conditions, or any assumption regarding water content profiles, etc.

(b) In both cases, the water content of the unsaturated zone and WTD must be coupled, so what is the effective sequence of calculations over time? I struggle with "R is the amount of water from or to the unsaturated portion of layer 1 necessary to cause the rise or fall of the water table from its former position", knowing that WTD is updated based on equation 1 which depends on R.

(c) Since R is calculated differently if the WTD is larger or smaller than 4m, can we see a discontinuity of net recharge values at 4m (plotting R as a function of WTD)?

(d) This flux R is defined as the result of downward gravitational flux and capillary flux, which can be either up or downward. The resulting flux is called recharge in the results, but "flux through the water table in section 2.1 (L22 p 14): I invite the authors to harmonize throughout the ms, and use recharge, but as mentioned in my comment 1d, this "net" recharge, which can be positive or negative, should positive when down, to match the meaning of GW recharge.

(e) As an interested reader, I would also appreciate some explanations regarding the links between R and evapotranspiration, which must be tightly coupled as well: how is transpiration described? How is rooting depth described? What is the vegetation description at the surface: PFTs, mosaic approach, constant or varying over time, which input datasets?

The persistence induced by the GW component must somehow be related to its long residence time (as written p12, L27):

(f) Is there a way to quantify it, at least at first order? How does persistence link with the transmissivity of the GW system (it would be useful to give information on it, how is Ks estimated, based on soil texture? which effective thickness?) and the GW-river flux (Qr), for which some quantitative parameter values would also be useful.

River flow scheme:

(g) It is said that river width is taken form HydroSHEDS, but this variable does not belong to the standard dataset (https://hydrosheds.org/pages/availability). Please be more specific.

The simulations are forced by an atmospheric reanalysis, ERA-Interim, without any bias correction except for precipitation:

(h) The reported horizontal resolution is about 0.7° x 1°, but the authors should check L30-31 p5, since I don't see why the resolution would be fixed in latitude and varying in longitude, it's usually the opposite which is done if seeking for constant grid-cell areas, but on the other hand, a factor of two over IB seem excessive.

(i) Precipitation is bias-corrected and downscaled to the 0.2° resolution. At 40°N (inside IB), the area of a 0.2°x0.2° grid-cell is a bit less than $20^2$ km², thus includes 64 LEAFHYDRO grid-cells ($2.5^2$ km², cf. resolution introduced L6 p7, when presenting the simulations). This resolution mismatch should be discussed, as it can have an impact on validation performances.

(j) Better meteorological forcing data sets probably exist in Spain, as the SAFRAN dataset of Quintana-Segui et al. 2017, containing all the variables required to force a LSM at the 5km resolution and 1-hourly time step, for 1979-2014. Else, WFDEI (Weedon et al., 2014) is a ready-to-use forcing data set, with bias-correction and downscaling to the 0.5° resolution, based of ERA-Interim. The submitted paper should include a justification for choosing ERA-Interim compared to other products, especially given that Gonzalo Miguez-Macho, co-author of the submitted paper, is also co-author of Quintana-Segui et al. 2017.

(k) I couldn't find the time step of LEAFHYDRO, and it is required for a modelling paper. If the model time step is shorter than the one of the forcing dataset, the downscaling should be mentioned.

Initial WTD: section 2.4 is not crystal clear for me, and some rewriting is advisable. In particular, the order of what is done is hard to follow, and the reasons to do what is done are not justified:

(l) There seems to be three successive initial WTD estimates at three different resolutions (1°; 9 arc-sec; the 2.5-km resolution of the simulations) but I don't understand at all what relates to the last two resolutions in the explanations of L12-16 p6. Can you please clarify?

(m) Why using recharge from a model without GW (Mosaic LSM) at 1°? Why not relying instead on the FD version of LEAFHYDRO model? At L10 p 6, is Qsr surface runoff?

(n) If topography is very important for the WTD patterns (L12 p 6), why using a higher resolution for the initial WTD and not for the transient simulation? Is it a problem of a computing power?

(o) The differences between the initial WTD (EWTD from Fig2) with the mean WTD over the 10 years (Fig 7b) should be discussed.

The way to obtain the power spectra of Fig. 14 is not at all explained but a few words wouldn't hurt.

(p) Shouldn't the compared curves have the same integral if they are calculated from time series of the same length?

**3. Quantification of results**

(a) The difference maps are interesting since they reveal clear sensitivity patterns related to WTD. Yet, an important part of the results is about water budgets, and means of the differences over IB would be interesting. This can be achieved either on the maps, or in a summary Table.

(b) An important question is about the significance of the reported changes in front of variability (seasonal and inter-annual), which can be assessed using inference tests. With simulations of only 10 years, non-parametric tests are probably advisable, and another solution would be to extend the simulation period, with additional advantages for persistence and long-term memory analysis.

(c) The validation of the models should involve more quantitative criteria. In particular, Fig 5 shows that WT strongly underestimates observed river flow (written p9 L7). Since ET is higher in WT than FD, one would expect that the river flow bias is smaller with FD (less ET with the same precip means more runoff and river flow): is it what is found? By how much? It doesn't seem true for the Ebro based on Fig. 15, which is weird.

(d) Eventually, what is the best simulation if we try to combine several performance criteria (correlation and bias, and also RMSE and Nash efficiency, or KGE which directly combines these scores, cf. Gupta et al., 2009)?

(e) P8 L27-28 claims that "the model's performance is reasonably good at shallow water table depth points, but significantly worse where the water table is deeper": I don't think it is supported by any figure or result.

(f) P10 L7-8: can you prove/justify/quantify how "small" is the long-term upward flux in flat areas?

(g) P12, L3: "precisely where the correlation between soil moisture and precipitation are reduced": this is not obvious from Fig. 9, and additional diagnostics would be interesting to prove this conclusion.

**4. Structure and writing**

(a) In absence of land-atmosphere coupling, the title is not well supported for the "land-atmosphere interactions", and should be modified.

(b) The introduction is long and messy, and would benefit from serious reshaping. The discussion on the need for realistic water table simulations (L34 p2 to L5 p3) is not well articulated with the rest, and is actually contestable. Besides, it raises questions since the WTD and river flow simulated by LEAFHYDRO (section 3.1) are not particularly realistic, although not very bad either. The paragraph at L13-20 p3 is very general and seems odd when the introduction starts to present the specificities of the presented work (starting at "Our work, p3, L5). The last part of the introduction (p2 L21 to P4 L8) reviews Spanish hydrology, and finishes on irrigation. Eventually, the specific research questions of the paper are not clearly stated by the end of the introduction.

(c) The paper frequently refers to a "bimodal" memory of soil moisture induced by GW persistence, but this term "bimodal" is not very clear: why isn't it just normal memory? I urge the authors to define what they really mean.

(d) Consider gathering the validation of river flow and sensitivity of river flow to WT vs FD.

(e) Consider presenting Fig. 10 before Fig. 9, which makes a nice introduction to Fig. 10, and justifies why time correlation is analyzed on time series of annual means, while many papers in the literature consider time lags of months. The paper may insist on memory at pluri-annual timescale, which is quite novel in the literature surface-GW interactions.

(f) The conclusion is mostly a summary of what was just presented in sections 3 and 4. The summary part should be strongly shortened, to better highlight the main findings instead of again comparing % changes. Another advantage would be to leave space for a real discussion of the results, which is cruelly lacking.

(g) In particular, the results and the conclusion they support are likely dependent on the model, its assumptions, and the forcing datasets (meteorology, soils, vegetation). This must be said and leads to compare the conclusions of the paper to the literature.

(h) For instance, the underestimation of riverflow (Figs 5 and 15) means that ET is too strong in the model(s) or precipitation too weak: can't it create a bias in the sensitivity of ET to WTD? This should be discussed, in relationship with the quality of meteorological and vegetation forcing datasets, or the fact that irrigation is not taken into account (cf. p9 L31-32).

(i) The perspectives could also be developed… Are "coupled land hydrology-climate models" (p16, L29) the only to move forward?

**References (for those not cited in the discussion paper):**

Gupta, Kling, Yimaz, Martinez: Decomposition of the Mean Squared Error and NSE Performance Criteria: Implications for Improving Hydrological Modelling; Journal of Hydrology, 377(1), 80-91, doi:10.1016/j.jhydrol.2009.08.003, 2009.

Quintana-Seguí, P., Turco, M., Herrera, S., & Miguez-Macho, G.: Validation of a new SAFRAN-based gridded precipitation product for Spain and comparisons to Spain02 and ERA-Interim, Hydrology and Earth System Sciences, 21(4), 2187–2201. doi: 10.5194/HESS-21-2187-2017, 2017.

Weedon, G.P., Balsamo, G., Bellouin, N., Gomes, S., Best, M.J. and Viterbo, P., 2014. The WFDEI meteorological forcing data set: WATCH Forcing Data methodology applied to ERA-Interim reanalysis data. *Water Resources Research*, 50, doi:10.1002/2014WR015638.

Westerhoff, R., White, P., and Miguez-Macho, G.: Application of an improved global-scale groundwater model for water table estimation across New Zealand, Hydrol. Earth Syst. Sci., 22, 6449-6472, https://doi.org/10.5194/hess-22-6449-2018, 2018.

---

## Referee Comment (RC3) · Anonymous Referee #3 · 4 Apr 2019

This paper discusses the role the water table plays in the terrestrial water cycle through the provision of vertical fluxes it provides for crops to evapo-transpire. The authors apply a Land Surface Model LEAFHYDRO that also simulates the dynamics of water table. They present results that show the difference between the simulated soil moisture values with and without the inclusion of the water table. I think this paper addresses relevant scientific questions within the scope of HESS, it represents interesting tools and ideas; however, the presented methodology and data fall short from supporting the reached conclusions.

It is clear that significant work has been undertaken to produce the results; however,

I think because the authors are dealing with many processes including land surface, unsaturated zone, and saturated processes, the paper as it stands lacks a lot of information that are necessary to convince the reader with the applied methodology and possibly the repeatability of the experiment. In addition, there are concerns related to the structure of the paper where introduction, results, and discussions are all mixed together. My points below make these comments clear:

At the beginning of the introduction, the authors state that "groundwater exchanges with the land surface occur via vertical fluxes through the water table surface, and horizontal water redistribution via gravity driven lateral flow". The authors must be specific regarding the type of the lateral flows. Are these flows in the saturated zones only? Are they in the main aquifers or perched aquifers? Or do they also include what is called through flows, i.e. lateral movement of infiltrated due to the existence of low permeability materials above the water table?

The scale the authors are dealing with is a national scale. It is expected that many types of hydro-geological conditions will be met at this scale. It is not expected that they will deal with all possible hydro-geological settings, however, the paper must clearly state the selected hydro-geological condition the model is applied to. A diagram showing a conceptual model of this hydrogeological setting is needed. All the results to be presented and discussed has to be put always within the context of this conceptual model.

The introduction must be more focused. The paper states the aim of the paper in the first paragraph of the paper. The introduction then tries to explain the reasons for undertaking the work afterwards. I think the argument should be built the other way round. In addition the introduction includes description of the methodology applied (Page 3 Lines 5 to 12) and site description (paragraph starting from Line 20 on Page 3). I have difficulties with some of the definitions and terminology used. For example, on Line 34 Page 3, the authors write "reflecting the importance of groundwater memory". Why do they need to call it memory? It is the groundwater storage that reduces the

impact of extreme weather events. The use of positive and negative recharge is also confusing (although clearly defined) and not intuitive.

Section 2 must be split into two sections one describing the study area including the information that are presented in the "Introduction", in addition to the conceptual model. The other section must be dedicated to the Methodology, which must include a lot more information than what is already presented. For example:

- Equation 1 shows the temporal variations of groundwater storage as a response to recharge. What about the soil moisture temporal variations?

- How does the model calculate evapotranspiration? Does it calculate runoff? Does it account for overland routing? Is overland water added to the groundwater flows emerging in the rivers to calculate total flows at the gauging station?

- How is the capillary flux calculated? Is it dependent on the position of the water table? (It is clear it is but at least it must be described in the methodology)

- How capillary forces are presented in the model? When a water table exists, the water is available to evapo-transpire wherever the water table depth is?

- It is not clear how the high resolution steady state simulation results are used in the low resolution time variant results (This is explained later, but what is mentioned in Section 2 is not enough to clarify this approach.

- It is stated that the shallow water table slows down drainage. If the soil is not fully saturated and the water does not pond on the surface, how the shallow water slows down drainage?

- It must be explained here that rivers could be influent and effluent

- Are the groundwater flows also driven using Darcy's law or is it based on hydraulic gradient only? What is the calibration procedure used to find the spatially distributed hydraulic conduct values?

Section 2.2 provides information about the source of data but no information about the data are provided. For example information about the spatial distribution of landuse is important to understand the amount of water extracted by evapo-transpiration from the soil store. Nothing is mentioned about the hydrogeological data used in the model such as the values of the hydraulic conductivity and storage coefficient of the aquifer, river bed conductance values, etc.

In Section 2.4, can you state please which groundwater model is used with the Mosaic LSM recharge model to calculate the initial EWTD? On Lines 10 to 18 (Page 6) it is unclear which model has the high resolution and which one has the low resolution. A diagram that shows the steps followed in methodology will be helpful. Text from Line 18 onward in this section are results. Why are they included in this section?

In Section 3 the authors dip into discussing the validation of a model while no information about the hydraulic parameters used in the model are provided. These include parameters controlling overland, subsurface, and unsaturated flows as well as soil and landuse data. They claim that the temporal variabilities are reproduced. However, with the lack of the parameter values and the definition of the context (assumptions and conceptual model) within which the model is built, this conclusion is easily challenged.

In Section 4.1 (Lines 25 to 30 on Page 9), the authors define positive and negative recharge in an unintuitive way since in groundwater, recharge is referred to as inflow to the groundwater reservoir and the opposite is a discharge from the water store and that could be in any direction (like the upward capillary fluxes). The sentences on Lines 10 to 14 on Page 10 are not very well formulated and together with the comment above, it is difficult to understand the point the authors are trying to make. On Line 15, the argument "this cycle is more pronounced the shallower the water table" is not very strong since Figures 6c to f all show seasonal variations across the whole peninsula.

In Section 4.3: can you please state how annual anomalies are calculated? Is it a difference from a long term average value or the difference from an average calculated

on the date the anomaly is determined?

Line 28 Page 11: are anomalies in precipitation and anomalies in soil moisture correlated or are the anomalies in soil moisture correlated with precipitation. Please clarify

Section 4.4 Line 21: "water table depth (red lines)" are observed or simulated? If simulated is it from the model with water table or with free drainage?

Figure 9: Please correct the caption for the left figure which should be related to the free drainage (FD)

In Figure 11, I expect the soil moisture anomalies calculated from the simulation with a water table to be lower in absolute value than those calculated from the simulation with free drainage. This appears to hold true for all hydrological years except Years 8 and 9 (Compare row 3 to row 2). Why?

Finally, I think the paper has to include a Discussion section where the analysis of the results has to be aligned with the assumptions listed in the conceptual model together with the hydraulic characteristics of the studied domain and the landuse controlling the amount of evapotranspiration from the soil zone. While the amount of work that has been taken and presented must be recognised and appreciated, I think the addition of a discussion section and rewriting the conclusion section to address the main findings concisely will greatly improve the presentation of this work

---

## Author Comment (AC1) · 8 Jul 2019

General comments

This manuscript assesses the groundwater influence on soil moisture memory and land atmosphere interactions in the Iberian Peninsula by using the LEAFHYDRO model. The simulation was performed at 2.5-km over the Iberian Peninsula for a 10 year period. The authors found significantly wetter soil and enhanced ET over shallow water table regions suggesting that groundwater might have an impact on climate over the Iberian Peninsula.

This study follows two previous studies carried out in United Sates (Miguez-Macho et al., 2007) and over the Amazon basin (Miguez-Macho and Fan, 2012) where the same model was used to depict the influence of groundwater on soil moisture and atmospheric variables. The methodology and science questions of the present paper are very similar to these two previous papers but applied for the Iberian Peninsula. Regarding the main conclusion obtained from the present paper, most of them are consistent and confirm the findings in numerous previous studies, including the two previous studies using the LEAFHYDRO model. However, no significant new findings can be drawn from this modeling, hence the novelty cannot be said to be high. In my opinion, this is the first major issue of the paper. The authors should consider to better highlight what is the interest of using such high-resolution model over Spain with respect to the previous study using the same model (in United Sates and Amazonia). A reorganization of the introduction may help to better define the novelty introduced by the use of LEAFHYDRO over the Iberian Peninsula.

The paper is articulated in five sections: introduction, methodology, validation, results, and conclusion. Regarding this structure, I identified two general remarks that need to be solved. First, the methodology section do not give enough details on the model description and the data used. In my opinion, while the main purpose of the paper is related to groundwater-surface land relationships, this part is not enough detailed in the paper. Secondly, the results section introduces some elements of discussion that are not at all linked with the literature. No references are cited, neither in this results section, nor in the conclusion. Regarding the bunch of paper related to this subject (i.e. groundwater-soil moisture influence), the paper lacks of references. This is the second major issue.

Besides these general comments, I identified specific comments and technical errors in the text and in the figures that I put in comment in the subsequent sections. In particular, I wish to see all the figures wider, regarding the size of the simulated area.

Based on the above statement, I think major revisions are needed to solve the two previous major issues and the below specific comments and errors before the paper can be eventually published in HESS.

Authors: Thanks for this complete assessment. We understand the issues pointed out by the reviewer and have introduced substantial editions and changes to the manuscript to address them. We discuss such changes in response to the reviewer's specific comments below.

Specific comments

Generally speaking, and regarding the bunch of papers on the subject, the Introduction part lacks of references on land surface-atmosphere coupling and soil moisture mem- ory influences on groundwater and atmosphere. As an example the following papers should be considered:

Maxwell, R. M., Lundquist, J. K., Mirocha, J. D., Smith, S. G., Woodward, C. S. and Tompson, A. F. B.: Development of a Coupled Groundwater–Atmosphere Model, Mon. Wea. Rev., 139(1), 96–116, doi:10.1175/2010MWR3392.1, 2010.

Vergnes, J.-P., Decharme, B. and Habets, F.: Introduction of groundwater capillary rises using subgrid spatial variability of topography into the ISBA land surface model, J. Geophys. Res. Atmos., 119(19), 2014JD021573, doi:10.1002/2014JD021573, 2014.

Authors: Agreed. We have included the suggested references and others in the reviewed manuscript where they were relevant. Other references included:

Ying Fan, Gonzalo Miguez-Macho, Esteban G. Jobbágy, Robert B. Jackson, and Carlos Otero-Casal: Hydrologic regulation of plant rooting depth, PNAS 114 (40) 10572-10577, 2017 https://doi.org/10.1073/pnas.1712381114

Sobrino, J., Gómez, M., Jiménez-Muñoz, J., and Olioso, A.: Application of a simple algorithm to estimate daily evapotranspiration from NOAA-AVHRR images for the Iberian Peninsula, Remote Sensing of Environment, 110, 139–148, 2007. https://doi.org/https://doi.org/10.1016/j.rse.2007.02.017

Westerhoff, R., White, P., and Miguez-Macho, G.: Application of an improved global-scale groundwater model for water table estimation across New Zealand, Hydrol. Earth Syst. Sci., 22, 6449-6472, https://doi.org/10.5194/hess-22-6449-2018, 2018. https://www.hydrol-earth-syst-sci.net/22/6449/2018/

Page 2, line 4 to 5: This is the purpose of the paper. I suggest to move this part near the end of the introduction, after the definition of the science questions.

Page 3 line 21 to the end: "Here, we present a modelling study linking groundwater to soil moisture, land-atmosphere interactions and surface water" : You introduce the purpose of the paper in the first sentence and then explain why you chose your case study. To better highlight the subject of the paper and enhance the problematics that occurred in Spain and the opportunity to simulate groundwater and soil moisture at this scale, you should consider to move all this part before introducing the purpose of the paper.

Authors: Thanks. We agree that the characteristics of the region should be introduced first and then presenting what we have done for the Iberian Peninsula at the end of the introduction. We have deleted the mentioned sentences and included them as the last paragraph of the introduction:

"*In this paper, we present a modelling study linking groundwater to soil moisture, land-atmosphere interactions and surface water at the regional scale in the Iberian Peninsula. We investigate the role of groundwater in the hydrology of the region, focusing first, on its impact on soil moisture spatial variability, dynamics and long-term memory, second, on its effects on land-atmosphere ET fluxes, and third, on its direct impact on river flow.*"

Page 4 line 9: "Model description and settings". This part describes the model and data used in the study. Most of the formulation of the model's equations are described in (Miguez-Macho et al., 2007) and (Fan et al., 2007), so only the mass balance of the dynamic groundwater reservoir is given here. However, it could have been useful to have a description of how the water table head is calculated since it the main variable that is evaluated in the following sections. Information on how hydrodynamic parameters (transmissivity and porosity) are taken into account in the model could be added.

Page 4 Line 27 – Page 5 line 7: The coupling of the water table and the soil layers is unclear. Why the layer B is added? This part needs more details on how the water content is computed.

Page 5 line 7-14 : This part lacks of details about the calculation of the river- groundwater exchanges. It is the so-called river conductance model used in MOD- FLOW? Are river heights variables (using Manning's Formula) or prescribed? How are the river conductances determined?

Authors: Thanks. Our initial approach was to not include too many details on the model formulations, but rather refer the reader to relevant literature where such formulations are described and focus on the results. But given this one and another reviewer's comment, we have realized that the manuscript needs some of these model details to be consistent, and therefore we have edited substantially the Methodology section in the reviewed manuscript, adding information that responds to the reviewer concerns.

Page 5, line 15 "2.2 Initial land and river parameters" Regarding the title, should it be "Land-surface and river parameters"? Generally speaking, this part lacks of many details about the parameters used for developing the model over the Iberian Peninsula. The authors should consider the following remarks and maybe add a Figure depicting the case study.

Page 5 line 16-20: Why the soil textural classes are needed in LEAFHYDRO? What is the dominant soil type/vegetation type?

Page 5 line 21-24: How does the river flow scheme work? Does it used Manning's Formula? How are the river widths determined? This part lacks of details for the Iberian Peninsula.

Authors: Thanks. The textural classes are needed to derive the parameters governing the vertical water flux through the soil layers. They also appear in the calculation of transmissivities for lateral groundwater flow. We have edited section 2.2 to clarify this and the rest of the reviewer's concerns, including a description of the methodology used to calculate the river parameters and a new Figure to follow this methodology.

Page 6 line 4: Could you add details about the method used to disaggregate the IB02 data using the ERA-Interim precipitation data? This is not clear how the link between the two of them is described.

Authors: Yes, we have clarified this point in the reviewed version as follows:

*"Once the daily precipitation is read and interpolated into the model grid, the model temporally disaggregates the daily values throughout the day using 3-hourly ERA-Interim precipitation distribution. Hence, the model uses the IB02 daily analysis data for bias-correction of daily totals and ERA-Interim data for precipitation distribution throughout the day."*

Page 6 line 3: You speak about the model grid without having define his resolution before (0.2° ?).

Authors: This mention refers to the IB02 data resolution. After the new edition in section 2.2, the model grid resolution has been defined before this point.

Page 6 line 12-17: What is the resolution of the model grid? How was the global climatic recharge at low resolution used? Was it disaggregated at a higher resolution? Is it an annual mean average over a period? How was the test run aggregated to the model grid? This part is not clear.

Authors: We agree that the explanation on how we calculated our initial EWTD was not clear enough. We have edited the text from lines 11 to 17. The second paragraph in section 2.4 answers the reviewer's questions now and reads as follows:

*"We used topography data at high spatial resolution (9 arc seconds) in the EWTD calculation to properly capture topographic variability and local hillslope gradients (Gestal-Souto et al., 2010) A three-step process was followed, where first, a low resolution (1º) global climatic recharge from the Mosaic LSM was used to calculate a first estimate of EWTD by ingesting it to the 2D model using the high resolution topography; second, the resulting first high-resolution estimate of EWTD is simply aggregated to a grid of 2.5km to serve as initial water table condition for LEAFHYDRO full LSM 10-year test run (1989-1998), and third, a new high resolution EWTD was recalculated forcing the 2D model with the groundwater net recharge obtained with the LEAFHYDRO test run at 2.5 km and the high resolution topography. The test*

*run uses precipitation analysis and other forcings (see section 2.3) at higher resolution than the 1º climatic recharge from MOSAIC initially feeding the EWTD model, and produces a much more realistic recharge, totally compatible with our simulation settings"*

Page 6 line 25-28: Much details are needed on how soil moisture is calculated in LEAFHYDRO. This remark is linked to the model description in section 2.1. Some details could be added to illustrate soil moisture.

Authors: We have now added details about this in the first paragraph of section 2.1.

Page 6, line 29: 10 year is a rather short period to validate the model. I know some water table characterized by multi-year annual cycles of 20 years. Could you explained why you choose this time period? What is the time step?

Authors: The 10 year choice was a compromise between the computational capabilities at our disposal and the science issue of choosing a period long enough to include wet and dry years in order to study soil moisture and water table memory. The time resolution for resolving heat and water fluxes in the soil and at the land surface was 60 s. The time step for groundwater-stream exchange, groundwater mass balance and water table adjustment in the WT run is 900 s. We have included this information in Section 2.5.

Page 7 line 24: "in order to rule out measurements in confined aquifers as much as possible": does it means that you used some observations of confined aquifers to validate unconfined aquifers? It should be clarified.

Authors: We used water table depth data from the IGME (*Institute of Geology and Mining of Spain*), several *Confederaciones Hidrológicas* (Spanish agencies managing
the main watersheds within the country) and the SNIRH (*National Information System for Hydrological Resources of Portugal*). We had no information about the confinement of the aquifers, hence we decided to neglect those stations with observations with wtd lower than 100 m.

Page 8 line 3: "With regard to the observations, 203 of the studied stations present a shallow water table (wtd < 8 m) during the simulation period": does it mean that the mean water table depth is lower than 8 m?

Authors: Yes, exactly. We have added the word "mean" in the reviewed manuscript to clarify this point.

Page 8 line 10: Do these 3 different observation sites in Point 15 grid cell belong to the same aquifer or maybe to different layers? Coarse spatial resolution is a factor that could explain these differences, but the different piezometers can also monitored different aquifers. It should be verified. The same remarks applied for the other points with several observations.

Authors: That is true, thanks for the remark. LEAFHYDRO can only resolve one water table per grid cell. If observations at one point come from different aquifers within the same column, as the reviewer point out, the "vertical design of the model" would be the difficulty we were trying to point out in this paragraph, as well as the coarse resolution. We have added this point to the manuscript.

Page 8 line 14-line 17: The presentation of these percentage need to be clarified.

Authors: Agreed. We have tried to clarify, as:

*"Approximately one third of the stations present a shallow mean wtd (< 8 m), and 66.0 % of them are also found to have shallow mean water table by the model.*

*In terms of mean wtd error, 14.0 % of stations present less than 2 m difference between simulated and observed mean wtd at the available observation times*

*(red points in Fig. X. If we only consider shallow water table observations (mean wtd < 8 m), 33.0 % of them present less than 2 m difference with the mean simulated wtd."*

Page 8 line 17: "capturing the mean water table depth": is it rather "capturing the water table depth time evolution"?

Authors: We meant "capturing the mean water table depth" here, for points 1-12. Then in the next sentences we talk about time evolution and mention points 1-14, therefore points 1-12 are in both red (capturing mean wtd) and green (capturing time evolution) categories.

Page 8, line 26-27: this statement should be better connected with the results.

Authors: The following results section focused on the shallower water table points or regions, where the groundwater is connected to the top-soil hydrology. With this statement we aimed to summarize some of the figures presented in the previous paragraphs, particularly the percentages that were discussed two responses above.

Page 10, line 14-16: Recharge mean annual cycles is linked to ET and precipitation mean annual cycles, but Figure 6 only shows the climatology of the recharge variable. Results for ET and precipitation should be mentioned here, maybe in the Figure, or with some details in the text.

Authors: We have seasonal plots of the evapotranspiration fluxes in the model:

[Figure]

However, we decided not to include them, as we did not intend to include too many figures. We have added some text about the ET cycle in the manuscript: "*The seasonal character of ET in the Iberian Peninsula (Sobrino et al, 2007) is induced by water availability and incoming radiation; maximum values and higher spatial variability are found in spring and summer, whereas minimum values and variability appear in autumn and winter, when the incoming radiation is lower and the leaf area index decreases*". The Sobrino (2007) reference presents ET seasonal patterns using NOAA satellite images for the Iberian Peninsula, in agreement with the LEAFHYDRO patterns shown here. Of course we can add the figures as supplementary material if the reviewer see it necessary.

Page 10, line 19-23: "As the water table gets deeper": does it correspond to the EWTD of Figure 2 ? Or a time evolution? It must be clarified.

Authors: Yes, thanks for spotting this. We meant "Where the water table is deeper (Fig. 2)", and have corrected it in the reviewed manuscript.

Page 10, line 22: ET evolution is mentioned but no Figure show it.

Authors: See response 2 comments above.

Page 11, line 24: Anomalies are computed with respect to the annual mean or the mean annual cycle? It must be clarified in this subsection.

Authors: With respect to the annual mean. It has been clarified in the revised manuscript.

Page 12, line 29: The authors should add a sentence on the location of this region (reference to Figure 10).

Authors: Done in revised manuscript. Thank you.

Page 13 line 24: "but drainage is slowed down". This result need to be reinforced with further results, maybe with a water budget or a time evolution of the recharge.

Authors: We meant that drainage is slowed down in comparison with the FD run, as there are no upward capillary fluxes from below the top soil column or shallow water tables in free-drain approach. Drainage should then be faster than with the presence of a water table, at least in the regions where the water table is shallow. We have edited slightly the sentence as:

*"During the wet season (autumn-winter) the water table rises due to precipitation infiltration, but since drainage is slowed down as compared with the free-draining FD run, the soil moisture difference between both experiments also follows an upward trend"*

Page 14 line 15: "one year frequencies and at decadal timescales". Decadal timescale appear on these power spectrum analysis, but I wonder the pertinence of finding decadal timescale with a 10-year time series. A period of at least 20 years would have been more appropriate.

Authors: Yes, of course. Thanks for pointing this out. Decadal timescale results are not relevant in this analysis. We have deleted the comment.

Page 14, line 15-16: "The annual cycle, linked to that of the surface water balance": Could you better explain this statement? Maybe by linking it with previous results?

Authors: We were referring to the cycle apparent in the previous Fig. 13 (on ET and soil moisture). We have added this reference in the revised manuscript.

Page 14, line 24-25: "The higher weight of longer timescales of variation in the WT soil moisture series": same remark as above. A 10-year simulation appear rather short to establish this result.

Authors: The statement is still true for most basins at lower frequencies, particularly the Mediterranean basins at around 3yr frequencies.

Page 14, line 28: For this section, Figure15 is not necessary since it is the same as Figure 5. You should consider to had the FD simulation in Figure 5 directly. Figure 15 could be replaced by a Figure showing the stream-groundwater exchanges in order to discuss this flux.

Authors: We believe after consideration that having the 2 figures makes it easier for the reader. We present in Fig 5 the streamflow comparison for WT run only as this is the flow we obtain as output of our main simulation we extract conclusions from. We then present fig. 15 (now 16) when we study the main basins, focusing on the differences between WT and FD representations of river flow.

Page 23, Figure 2: The legend refers to EWTD and topographic data, but only one map is shown that corresponds to EWTD. Why describing topographic data here? This Figure could be wider and extend to the full wide of the page. Add a unit to the colorbar and a title.

Authors: The topography data was used to calculate EWTD as a balance between the climatic recharge and the lateral flow driven by topography, using the 2-D model in Fan et al (2007), as explained in section 2.4. Therefore, the original resolution of the EWTD data shown in the figure is the resolution of the used topography, we have clarified this point in the caption in the revised manuscript. We have also widen the figure and added the units to the colourbar.

Page 24, Figure 3: The authors describe a grid centered in the Iberian Peninsula. Figure 3 shows this peninsula, but also parts of France and North Africa. Could you add the limits of the simulated domain? Are the France and North Africa part also simulated? Generally speaking, Figure 3 should be reorganized to better highlight the results. A wider map centered on Spain could improve the reading. Using different color points for different

information on the same map is confusing. I suggest to use different maps for the different informations (wtd, correlation, steep, number of station par cells) and grouping them into a single figure.

Authors: Yes, the portions of France and North Africa in the figure are also part of the domain. But the reviewer is right in that the study is not about them, and particularly for this figure, their inclusion makes the information harder to read. We have cropped the figure to focus it on the peninsula, and made it wider in the manuscript, so that the information can be clearer seen. Still, we have decided to keep the colour code to avoid presenting too many maps. We believe that it can be better understood now:

[Figure]

Page 25, Figure 4: Point 8 and Point 11: the model seems to overestimate the amplitude of the piezometric head evolution. It should be mentioned and explained.

Authors: We have included in the revised text that at these very shallow water table points the amplitude of the wtd variations are larger in the model than in the observations.

Page 26, Figure 5: only correlation scores are given. The Nash-Sutcliffe (Nash and Sutcliffe, 1970) score could be used and commented in the text to quantify the quality of the simulation.

Authors: There are two important issues related to streamflow in these simulations. They are discussed in section 3.2 in the paper, but perhaps they need further clarification. The first one is related to the precipitation forcing data. From figure 5 it is obvious that there is a large amount of missing water in the model results. Only basin 5 (Guadalquivir) shows less streamflow in the observations than in the model, but this is because in this strongly regulated basin, water is heavily used for irrigation. While there can be some errors due to evaporation biases, we have evidence from local independent observation networks that this missing water is more related to the precipitation forcing (please, see the discussion about the same bias in the IB02 dataset in Rios-Entenza and Miguez-Macho, 2014). In the mountains, especially in the north, the IB02 dataset does not properly capture orographic enhancement, since it was obtained using simple interpolation algorithms.

The second problem is due to model parameterizations and is also commented in the paper. Surface runoff from excess saturation in thin soil or in subgrid near saturated areas is unrepresented. Due to unresolving hillslope hydrologic gradients at the 2.5km resolution, the connection between rivers and groundwater in cells where the mean water table is deep does not produce a good result either.

Since both forcing and model problems affect mostly mountain areas where terrain is complex, we are confident that the main conclusions in our work about groundwater and soil moisture are sound. However, we cannot say the same about riverflow, since the contribution from the mountainous areas to their total water budget is very important.

We have now calculated other skill scores for both experiments, as suggested by the reviewers. In the FD simulation, the lack of surface runoff is compensated by the fact that infiltration is readily incorporated into the rivers. Because precipitation amounts are biased low, winter peaks may look better in this FD simulation and some skill scores are better than in the WT simulation, but this does not mean that the result is physically correct.

| Station - Basin | Basin Catchment Area (km²) | E WT | r WT | $r_{mm}$ WT | E FD | r FD | $r_{mm}$ FD |
|---|---|---|---|---|---|---|---|
| Foz de Mouro - MIÑO | 15.407 | -0.13 | 0.89 | 0.98 | 0.55 | 0.93 | 0.95 |
| Puentepino - DUERO | 63.160 | -0.52 | 0.73 | 0.96 | 0.18 | 0.72 | 0.69 |
| Almourol - TAJO | 67.482 | 0.28 | 0.91 | 0.93 | 0.80 | 0.91 | 0.89 |
| Pulo do Lobo - GUADIANA | 61.885 | 0.07 | 0.65 | 0.71 | 0.21 | 0.54 | 0.66 |
| Cantillana - GUADALQUIVIR | 44.871 | 0.44 | 0.76 | 0.66 | 0.15 | 0.56 | 0.67 |
| Tortosa - EBRO | 84.230 | 0.36 | 0.74 | 0.93 | 0.55 | 0.82 | 0.87 |

For all the aforementioned reasons, we purposely wanted to limit our discussion about streamflow in the paper and just show the WT results. The only point that we wanted to make with the FD simulation is that in the Mediterranean climate of the Iberian Peninsula, summer stream flow is sustained by groundwater and, without it, in a simulation with a free drain approach, rivers dry out. We are confident that this result holds true, despite all the problems in the forcing and model parameterizations. We show the Ebro basin to illustrate this point because it is the one showing less streamflow total annual underestimation and annual cycle better matching observations in the control WT run, especially in winter. It so happens that it is also the largest basin the Peninsula, so the example is significant.

Page 29, Figure 7: add units for the maps and the y-axis of the bars. The (a), (b) and (c) letters need to be added to the titles.

Authors: Ok. Done in revised manuscript.

Page 30, Figure 8: the authors should think to show the seasonality of ET in Figure 8, or elsewhere, as said earlier.

Authors: Replied earlier. Thanks.

Page 34, Figure 12: add a title.

Authors: Ok. Done in revised manuscript.

Page 37, Figure 15: suppress this Figure and add the FD simulation to Figure 5. #

Authors: Responded above

Technical corrections

Page 1, line 2: "a key role" not "an key role"

Page 8, line 19: "simulated time series" instead of "simulated series"
Page 13, line 1: "for the large" instead of "for the the large" References

Authors: All technical corrections have been included in the revised manuscript. Thanks.

---

## Author Comment (AC2) · 8 Jul 2019

**AUTHOR'S RESPONSE TO RC2:**

Manuscript hess-2018-626 by Martinez-de la Torre & Miguez-Macho: "**Groundwater influence on soil moisture memory and land–atmosphere interactions in the Iberian Peninsula**"

The LEAFHYDRO model was among the first ones to couple a physically-based 2D groundwater (GW) flow model with a land surface model (LEAF2). Transient applications so far focused on the USA (2007 papers), the Amazon (2012 papers), while the groundwater model has been coupled to Noah over South-America (Martinez et al 2016), and over New-Zealand (Westerhoff et al., 2018, not cited). The present paper reports the application of the LEAFHYDRO over the Iberian Peninsula (IB), with novel insights regarding the propagation of precipitation anomalies to water table depth (WTD) and soil moisture anomalies, at a pluri-annual timescale. This effect is all the more pronounced as the climate is more arid, as are the more "classical" impacts of GW on ET, soil moisture, and river discharge at seasonal or shorter timescales.

Authors: Thanks. We have included the Westerhoff (2018) reference in Section 2.4, when we describe the 2D groundwater model (uncoupled to LSMs). This work was finally published while finishing up our manuscript initial writing.

This makes this paper very commendable for HESS, but on the other hand, the paper lacks (i) good quality figures, (ii) sufficient information on the model and forcing datasets, (iii) solid quantification of the reported impacts and model validation, (iv) a real discussion of the results, including the limitations of the approach. Other problems include a structure that is not always very logic, and a tendency to overstatement. Apart from a few spelling errors, the language is very clear. **Eventually, I advise to substantially revise the paper before publication in HESS**. None of the suggested revisions is very complicated, but many of them are advisable, as detailed below.

Authors: Thanks for this complete assessment. We acknowledge the issues pointed out by the reviewer and have introduced substantial editions and changes to the manuscript to address them, making the paper stronger in our view. We discuss such changes in response to the reviewer's specific comments below.

1. **Figures:**
   (a) Most IB maps are too small, and it's almost impossible to see something. Please remove Southern France, Northern Africa and oceans, and magnify the remaining IB. When possible, please use the same color scale for comparable variables (e.g. equilibrium and non- equilibrium WTD). It would also be very informative to add the mean and std of each map over the IB.

Authors: Thanks. In the revised manuscript, we have cropped the figures including parts of Africa and France to make the results over the Iberian Peninsula more visible. The non-equilibrium color scale was chosen in order to contrast with the mean top-2m soil moisture plot in the same map. The means of the variables represented in each plot are included in the bars below on the anomaly figures (Figs. 10 and 11 in the original submission).

   (b) Fig. 3 is too small as well, and the color code of the points locating the points with observed does not seem well adapted, since a point can meet several conditions: for instance, what is the color of a point where bias is less than 2 m (red), and correlation with observed time series is more than 0.5 (green), which must be possible? There also seems to be some black points, in which case their meaning should be explained. But maybe they are purple… It would also be useful to report the classification used for Fig. 3 on the various panels of Fig 4 (insert for each point the bias, the correlation coefficient, and the wtd slope).

Authors: Thanks. In the revised manuscript, we have cropped figures 3 (now 4) and made it larger. The colour of the dots is easier to identify now, and we have also explained in the caption the meaning of red dots (no validation criteria matched) and the hierarchy of the criteria (in the case pointed out by the reviewer the dots appear as red, this happens in most red dots on the next figure showing time series and it is also explained now in the text when describing the figure). New caption: "*Shallow water table zones (light blue shades) and Iberian Peninsula wtd observation stations (dots). Red dots are locations where observed and simulated wtd differences are within 2 m; green dots are stations with correlation over 0.5 between observed and simulated wtd series; purple dots are stations with steep wtd slope (>=0.035 m month-1), well captured by the model; orange dots are cells containing more than one observation station; black dots are cells where none of the above criteria is met by the model. Over cells where more than one validation criteria is reached the point adopts the colour of the first criterium met (in the order presented here); for instance, cells with mean wtd differences lower than 2 m and also correlations above 0.5, are shown as red on the map*"

    (c) For Fig3, Fig 5, Fig 15 (and potentially Fig4), it must be clarified if the reported correlation coefficient is calculated on the full time series (120 monthly values), or on the mean seasonal cycle (12 monthly mean values).

Authors: The correlations used in the wtd validation (Fig. 3, now 4) were calculated with the full time series available, where the time scale varied amongst the stations (see Fig. 4, now 5). The correlations reported in the river flow time series are calculated on the mean seasonal cycle, and that was the reason to show them inside the seasonal plots. We have clarified both scales in the revised manuscript and figure captions. Thanks

    (d) Fig. 6: R seems negative if downward, which seems odd for the flux which recharges the GW.

Authors: Yes, that is true. We acknowledge the groundwater recharge is often referred to as the positive flux into the groundwater reservoir. In this work, we have followed the model criteria for signs in fluxes, so that upward is positive (like evapotranspiration from the surface) and downward is negative (like the water flux through the soil layers and then into the groundwater). We have changed the name of the flux to "net recharge" in the revised manuscript in order to clarify this point at different instances. The first time the net recharged is referred to in the manuscript in Section 2.1: "*The water flux through the water table or net recharge R is the sum of gravitational downward groundwater recharge and capillary flux, and depending on soil wetness and atmospheric demand, it can be downwards, causing the water table to rise, or upwards, causing the water table to deepen*"

    (e) Fig. 7c: why not show a real mean seasonal cycle, with 12 monthly mean values, instead of 4 seasonal mean values? And couldn't you plot the seasonal cycle of the shallow WTD as well?

Authors: We think that the plot focusing on the 4 seasonal means is stronger, illustrating the greater groundwater influence in water-scarce seasons. Also, the presentation as 4 seasons follows the net recharge figures (Fig. 6, 7 now), connecting with the seasonal variability in the net recharge and making the point clearer for the reader.

    (f) Fig. 8: the color scale is not clear, we cannot distinguish the values that are not zero. Besides, could you add a scatter plot of summer ET difference against summer WTD, to show if there is a kind a threshold WTD inducing a marked ET difference?

Authors: We have cropped and made the figure larger in the revised manuscript, making the green scale more distinguishable for the reader.

(g) Fig. 9 is very noisy: could you add the difference between center and left panels?

Authors: We have noticed a mistake in the caption, the left and centre panels descriptions were switched. We have corrected it in the revised manuscript. Of course we could add the difference plot suggested by the reviewer, but would not this make the figure noisier? The idea is that the plot on the right highlights with high values those areas of difference between the centre and left plots.

(h) Fig. 11: please add the lon/lat of the mapped area, either on the maps, or on the caption.

Authors: Yes, done. Thank you.

(i) Consider merging top panel of Fig 5 and Fig12; same for Fig 5 and Fig 15.

Authors: That was our initial approach in the first draft, but we believe after consideration that having the 2 figures makes it easier for the reader, as we present Fig 15 (now 16) when we study the main basins, while the plot on top of Fig. 5 (now 6) is about rivers and gauging stations.

By the way, why not show the FD simulations at all stations? And correct the statement that Ebro at Tortosa is where the model exhibits the best scores (L32-33 p 14): based on correlation coefficient, this station is only the third best for simulation WT based on Fig 5; besides, the ms discusses two models, so clarification is needed. Finally, the correlation coefficient is far from enough to support a performance analysis, and I strongly recommend that other classical criteria are documented (bias, important since ET changes between the two simulations; RMSE; Nash and/or KGE, which are classical skill criteria in river hydrology).

Authors: There are two important issues related to streamflow in these simulations. They are discussed in section 3.2 in the paper, but perhaps they need further clarification. The first one is related to the precipitation forcing data. From figure 5 it is obvious that there is a large amount of missing water in the model results. Only basin 5 (Guadalquivir) shows less streamflow in the observations than in the model, but this is because in this strongly regulated basin, water is heavily used for irrigation. While there can be some errors due to evaporation biases, we have evidence from local independent observation networks that this missing water is more related to the precipitation forcing (please, see the discussion about the same bias in the IB02 dataset in Rios-Entenza and Miguez-Macho, 2014). In the mountains, especially in the north, the IB02 dataset does not properly capture orographic enhancement, since it was obtained using simple interpolation algorithms.

The second problem is due to model parameterizations and is also commented in the paper. Surface runoff from excess saturation in thin soil or in subgrid near saturated areas is unrepresented. Due to unresolving hillslope hydrologic gradients at the 2.5km resolution, the connection between rivers and groundwater in cells where the mean water table is deep does not produce a good result either.

Since both forcing and model problems affect mostly mountain areas where terrain is complex, we are confident that the main conclusions in our work about groundwater and soil moisture are sound. However, we cannot say the same about riverflow, since the contribution from the mountainous areas to their total water budget is very important.

We have now calculated other skill scores for both experiments, as suggested by the reviewer. In the FD simulation, the lack of surface runoff is compensated by the fact that infiltration is readily incorporated into the rivers. Because precipitation amounts are biased low, winter peaks may look better in this FD simulation and some skill scores are better than in the WT simulation, but this does not mean that the result is physically correct.

| Station - Basin | Basin Catchment Area (km$^2$) | E WT | r WT | $r_{mm}$ WT | E FD | r FD | $r_{mm}$ FD |
|---|---|---|---|---|---|---|---|
| Foz de Mouro - MIÑO | 15.407 | -0.13 | 0.89 | 0.98 | 0.55 | 0.93 | 0.95 |
| Puentepino - DUERO | 63.160 | -0.52 | 0.73 | 0.96 | 0.18 | 0.72 | 0.69 |
| Almourol - TAJO | 67.482 | 0.28 | 0.91 | 0.93 | 0.80 | 0.91 | 0.89 |
| Pulo do Lobo - GUADIANA | 61.885 | 0.07 | 0.65 | 0.71 | 0.21 | 0.54 | 0.66 |
| Cantillana - GUADALQUIVIR | 44.871 | 0.44 | 0.76 | 0.66 | 0.15 | 0.56 | 0.67 |
| Tortosa - EBRO | 84.230 | 0.36 | 0.74 | 0.93 | 0.55 | 0.82 | 0.87 |

For all the aforementioned reasons, we purposely wanted to limit our discussion about streamflow in the paper and just show the WT results. The only point that we wanted to make with the FD simulation is that in the Mediterranean climate of the Iberian Peninsula, summer stream flow is sustained by groundwater and, without it, in a simulation with a free drain approach, rivers dry out. We are confident that this result holds true, despite all the problems in the forcing and model parameterizations. We show the Ebro basin to illustrate this point because it is the one showing less streamflow total annual underestimation and annual cycle better matching observations in the control WT run, especially in winter. It so happens that it is also the largest basin the Peninsula, so the example is significant.

(j) Fig. 13: a full paragraph is devoted to analyzing the seasonal variations of the different variables (L21-31 p 13), but we cannot see them. Please add the mean seasonal cycle next to the 10-yr time series to support this discussion. It would also help to magnify the scale of SM differences. The caption says the precipitation anomalies are calculated over the entire basins, while the other anomalies are calculated in the fraction with shallow WT (ca 1/3): why not calculate them over the same domain, to avoid any doubts. Finally, the text says the WTD of Ebro and Segura recover after the pluri-annual central drought, but it is not discernible in the panels.

Authors: We chose to plot only shallow wtd regions within the basins to highlight the effects of the interaction with the water on soil moisture and ET; however, seasonal precipitation is computed over the whole basin because lateral groundwater flow and river infiltration redistribute infiltration horizontally, making precipitation over all cells in the basin potentially relevant for the results over shallow water table cells. The recovery over the last 3 years is, we believe, clear in most all basins that suffered the central drought; however, the reviewer is right in that in the Ebro and Segura basins only a change in tendency from deepening to stabilizing or slightly rising is discernible. We have clarified this in the revised version.

**2. Methods:**

The LEAFHYDRO model has already been published, but a paper needs to be self-consistent, and more info is needed on the parts that are relevant to the conclusions.

The recharge calculation, in particular, is far from being clear, at least to me, although I have looked for more information in Miguez-Macho et al. 2007, Part 2. This should be clarified in the article, and the following questions might help the authors:

(a) The calculation is different depending if the WTD is larger or not than the soil depth, but I couldn't understand scenario b with larger WTD. In this case, how are the water content of points B and C estimated? It's written the one of point C "is determined by mass balance from the fluxes above and below" (L5-6 p 5) but these fluxes also need to be estimated, and there seem to be too many unknowns: please clarify the system, including flux equations, boundary conditions, or any assumption regarding water content profiles, etc.

(b) In both cases, the water content of the unsaturated zone and WTD must be coupled, so what is the effective sequence of calculations over time? I struggle with "R is the amount of water from or to the unsaturated portion of layer 1 necessary to cause the rise or fall of the water table from its former position", knowing that WTD is updated based on equation 1 which depends on R.

Authors: [(a) and (b)] Yes, we agree that, since we initially tried to avoid a methodology section too long and filled with equations already published in Miguez-Macho et al (2007), the methodology section is not as explanatory as it should. The reviewer is right to point out that some clarifications are needed. We have changed Section 2.1 and we believe that the issues raised by this and other reviewers about the model formulation and steps have been addressed in the revised manuscript.

(c) Since R is calculated differently if the WTD is larger or smaller than 4m, can we see a discontinuity of net recharge values at 4m (plotting R as a function of WTD)?

Authors: Possibly, but the model formulation is designed precisely to avoid any discontinuity in water table depth or recharge. When the water table goes below 4m, calculations are identical to when it was in the layer above, and only as it goes below 4.5m they start to differ, but they do it very gradually. No discontinuity is observed in water table depth as it goes deeper (or shallower) than 4 or 4.5 m, which is a good indication that there isn't one in recharge either.

(d) This flux R is defined as the result of downward gravitational flux and capillary flux, which can be either up or downward. The resulting flux is called recharge in the results, but "flux through the water table in section 2.1 (L22 p 14): I invite the authors to harmonize throughout the ms, and use recharge, but as mentioned in my comment 1d, this "net" recharge, which can be positive or negative, should positive when down, to match the meaning of GW recharge.

Authors: We have followed the reviewer's suggestion and called the flux "net recharge" throughout the text, presenting it initially in Section 2.1 as "*The water flux through the water table or net recharge R*". We would prefer though to keep the signs as they are, positive upwards and negative downwards, as these are the signs in the model calculations too, where upward fluxes like ET are positive and downward fluxes like infiltration are negative.

(e) As an interested reader, I would also appreciate some explanations regarding the links between R and evapotranspiration, which must be tightly coupled as well: how is transpiration described? How is rooting depth described? What is the vegetation description at the surface: PFTs, mosaic approach, constant or varying over time, which input datasets?

Authors: The model parameterizes the calculation of transpiration and evaporation from canopy interception using PFTs and the vegetation data described in Section 2.2. We have added a paragraph at the end of Section 2.1 pointing out this: "*When there is vegetation on the surface, the water and heat exchanges between vegetation and the surrounding canopy air parameterization is based on Avissar et al. (1985). This methodology uses PFTs (Plant Functional Types) constant through the simulation period, assigning a type to each cell that will determine parameters like the root depth, the minimal stomatal conductance (that will be increased by atmospheric factors) and the LAI (Leaf Area Index), that will affect the calculation of canopy resistance, transpiration and evaporation from the canopy surface. The transpiration is taken from the moistest level in the root zone.*"

The persistence induced by the GW component must somehow be related to its long residence time (as written p12, L27):

(f) Is there a way to quantify it, at least at first order? How does persistence link with the transmissivity of the GW system (it would be useful to give information on it, how is Ks estimated, based on soil texture? which effective thickness?) and the GW-river flux (Qr), for which some quantitative parameter values would also be useful.

Authors: As in the response to (a) and (b), after this and other reviewer's comments, more information has been added in Section 2.1 about the transmissivity and Qr flux and how they are calculated. We believe this has made the paper more consistent.

River flow scheme:

(g) It is said that river width is taken form HydroSHEDS, but this variable does not belong to the standard dataset (https://hydrosheds.org/pages/availability). Please be more specific.

Authors: A complete description of how the river parameters have been calculated (including the new Fig. 2) has been added to Section 2.2 in the revised document.

The simulations are forced by an atmospheric reanalysis, ERA-Interim, without any bias correction except for precipitation:

(h) The reported horizontal resolution is about 0.7° x 1°, but the authors should check L30-31 p5, since I don't see why the resolution would be fixed in latitude and varying in longitude, it's usually the opposite which is done if seeking for constant grid-cell areas, but on the other hand, a factor of two over IB seem excessive.

Authors: Thanks for pointed this out. The resolution of the driving ERA-Interim data is now reported in Section 2.3, from Berrisford et al. (2011), as: "*ERA-Interim is presented in a reduced Gaussian grid with approximately uniform 79 km spacing for surface grid cells*". And the actual LEAFHYDRO model resolution, to which driving data are interpolated, is reported in Section 2.5 as: "*The simulation domain is a Lambert-Conformal grid centered in the Iberian Peninsula with a spatial resolution of 2.5 km*."

(i) Precipitation is bias-corrected and downscaled to the 0.2° resolution. At 40°N (inside IB), the area of a 0.2°x0.2° grid-cell is a bit less than $20^2$ km², thus includes 64 LEAFHYDRO grid-cells ($2.5^2$ km², cf. resolution introduced L6 p7, when presenting the simulations). This resolution mismatch should be discussed, as it can have an impact on validation performances.

Authors: Yes, of course. In fact, as mentioned earlier, we believe that this is the main reason for the underestimation of winter streamflow in our simulations. We discuss it in Section 3.2: "*There is a clear underestimation of the winter river flow by the model. Two factors contribute to this bias. First, a lack of precipitation in the forcing data, since the IB02 analysis dataset original resolution (0.2º) is coarser than our model simulations and the station density (7 km in Spain and 11.7 km in Portugal) is not sufficient to capture precipitation peaks due to orographic enhancement over the mountains, which is very pronounced in the northern cordilleras.*"

(j) Better meteorological forcing data sets probably exist in Spain, as the SAFRAN dataset of Quintana-Segui et al. 2017, containing all the variables required to force a LSM at the 5km resolution and 1-hourly time step, for 1979-2014. Else, WFDEI (Weedon et al., 2014) is a ready-to-use forcing data set, with bias-correction and downscaling to the 0.5° resolution, based of ERA-Interim. The submitted paper should include a justification for choosing ERA- Interim compared to other products, especially given that Gonzalo Miguez-Macho, co-author of the submitted paper, is also co-author of Quintana-Segui et al. 2017.

Authors: Yes, we are aware of such datasets. The decision of using ERA-Interim an IB02 was adopted at the time of conceptualization and set up for this study, when the other (newer) datasets were not available. Both mentioned datasets, however, and also the newly developed MSWEP dataset (Beck et al., 2017), have been more recently used with LEAFHYDRO simulations in a study focused on droughts: *The Utility of Land-Surface Model Simulations to Provide Drought Information in a Water Management Context Using Global and Local Forcing Datasets* (Quintana-Segui et al., 2019)

(k) I couldn't find the time step of LEAFHYDRO, and it is required for a modelling paper. If the model time step is shorter than the one of the forcing dataset, the downscaling should be mentioned.

Authors: The time resolution for resolving heat and water fluxes in the soil and at the land surface was 60 s. The time step for groundwater-streams exchange, groundwater mass balance and water table adjustment in the WT run is 900 s. We have included this information in Section 2.5.

The meteorological driving data are linearly temporally interpolated to the model time steps.

Initial WTD: section 2.4 is not crystal clear for me, and some rewriting is advisable. In particular, the

order of what is done is hard to follow, and the reasons to do what is done are not justified:

- (l) There seems to be three successive initial WTD estimates at three different resolutions (1°; 9 arc-sec; the 2.5-km resolution of the simulations) but I don't understand at all what relates to the last two resolutions in the explanations of L12-16 p6. Can you please clarify?
- (m) Why using recharge from a model without GW (Mosaic LSM) at 1°? Why not relying instead on the FD version of LEAFHYDRO model? At L10 p 6, is Qsr surface runoff?
- (n) If topography is very important for the WTD patterns (L12 p 6), why using a higher resolution for the initial WTD and not for the transient simulation? Is it a problem of a computing power?
- (o) The differences between the initial WTD (EWTD from Fig2) with the mean WTD over the 10 years (Fig 7b) should be discussed.

Authors: Yes, we agree that Section 2.4, in the original version, would gain with some clarifications and we have rewritten it. Hopefully the reviewer and the readers will find it now clearer.

In response to particular comments:

(l) the 1 degree resolution corresponds only to the initial recharge dataset from the Mosaic LSM that is used to feed the 2D groundwater model (Fan et al., 2007). This 2D steady-state model runs at 9 arc-sec and produces initial conditions for wtd at this resolution, resolving hillslope gradients.

(m) We needed an initial guess for climatological recharge to feed the 2D groundwater model. We had tried other datasets and results using Mosaic recharge, despite being very coarse, gave the best skill scores in validation with point observations. Running LEAFHYDRO with FD would mean an extra step that we deemed unnecessary.

Yes, Qsr is surface runoff.

(n) Of course the 2.5km resolution for the LEAFHYDRO simulations is a compromise needed for lack of computational resources. This is explained in Section 2.5 (now renamed "Simulations set-up")

(o) The differences come mostly from the colorscale and resolution difference in the plots.

The way to obtain the power spectra of Fig. 14 is not at all explained but a few words wouldn't hurt.

- (p) Shouldn't the compared curves have the same integral if they are calculated from time series of the same length?

Authors: We used the intrinsic function "spectrum" from the MathWorks software, which computes the power spectrum of a given time series, and it is simply based on performing the Fourier transform. We have added a comment about this in the manuscript.

3. **Quantification of results**
   - (a) The difference maps are interesting since they reveal clear sensitivity patterns related to WTD. Yet, an important part of the results is about water budgets, and means of the differences over IB would be interesting. This can be achieved either on the maps, or in a summary Table.

Authors: Yes, we agree. This was the reason to add the colour bars below each plot in figures 9 and 10 (now 10 and 11), representing the mean anomaly value for the peninsula or the zoomed area.

- (b) An important question is about the significance of the reported changes in front of variability (seasonal and inter-annual), which can be assessed using inference tests. With simulations of only 10 years, non-parametric tests are probably advisable, and another solution would be to extend the simulation period, with additional advantages for persistence and long-term memory analysis.

Authors: Unfortunately, the simulation period could not be extended due to computational limitations.

    (c) The validation of the models should involve more quantitative criteria. In particular, Fig 5 shows that WT strongly underestimates observed river flow (written p9 L7). Since ET is higher in WT than FD, one would expect that the river flow bias is smaller with FD (less ET with the same precip means more runoff and river flow): is it what is found? By how much? It doesn't seem true for the Ebro based on Fig. 15, which is weird.

Authors: The presence of the groundwater reservoir in the WT simulation buffers out variations in climate. Even with the same ET in both WT and FD run, in dry periods there can be more baseflow input in the WT simulation, coming from groundwater. The opposite can also be true. In a wet year, there can be less baseflow in the WT run if the groundwater reservoir levels are low from a previous drought. The mean flow for a given period should be about the same if there is no trend in groundwater levels. But this is not guaranteed either, because there can be some extra store or depletion of water in the layer between the water table and the bottom of the soil column at 4m, which is not existent in the FD run.

In Fig. 15, there is a declining trend in the water table, albeit small. In shallow water table areas, the lowering water table might be partially sustaining ET; however, where the water table is deeper, the lowering groundwater store is sustaining streamflow. This trend explains why there is more water in the annual mean total streamflow in the Ebro in the WT run, and is common to the other Mediterranean basins, more affected by the drought. We will now discuss this issue, related to the relatively short period of simulation, in the revised manuscript. The point that we wanted to make with the figure is still true, though, as it is apparent from the comparison with FD run that groundwater sustains streamflow not only through summers, but also longer dry periods.

    (d) Eventually, what is the best simulation if we try to combine several performance criteria (correlation and bias, and also RMSE and Nash efficiency, or KGE which directly combines these scores, cf. Gupta et al., 2009)?

Authors: In terms of simulating river flow, we obtain better metrics (Nash-Sutcliffe efficiency) with FD, but in terms of simulating surface flow exchanges, we have shown how a more realistic soil moisture in the presence of the groundwater influence will result in different surface-atmosphere coupling effects, which is ultimately the issue we are focusing on. As mentioned earlier, streamflow results are more complex to analyze, and a better score with bad forcing does not mean that the simulation is more physically realistic.

(e) P8 L27-28 claims that "the model's performance is reasonably good at shallow water table depth points, but significantly worse where the water table is deeper": I don't think it is supported by any figure or result.

Authors: With this statement we refer to the improvement in the wtd validation reported when we consider only shallow water table points as compared with all points, deep and shallow are considered. We have rewritten the paragraph where we provide this wtd validation results and the point should be clearer in the revised manuscript.

(f) P10 L7-8: can you prove/justify/quantify how "small" is the long-term upward flux in flat areas?

Authors: The flat areas we refer to present values between 0 and 150 mm/yr, or between 0 and 0.5 mm/day in the case of the seasonal plots, in both cases, closer to 0, than to the upper value of the range. Our point was to separate them from the river valley areas we mention in the following sentences, where faster lateral drainage due to the steepness of the terrain result in much higher upward flux values.

(g) P12, L3: "precisely where the correlation between soil moisture and precipitation are reduced": this is not obvious from Fig. 9, and additional diagnostics would be interesting to prove this conclusion.

Authors: This sentence refers to the "missing plot" of differences between right and center maps in the figure that the reviewer mentioned in the concern (1g). We think that once the error in the caption has been corrected the intended exercise of focusing on differences between both maps is easier for the reader.

4. **Structure and writing**
   (a) In absence of land-atmosphere coupling, the title is not well supported for the "land-atmosphere interactions", and should be modified.

Authors: We did debate about this wording during the writing process. We have now changed the word "interactions" for "fluxes" in the title in the revised manuscript.

   (b) The introduction is long and messy, and would benefit from serious reshaping. The discussion on the need for realistic water table simulations (L34 p2 to L5 p3) is not well articulated with the rest, and is actually contestable. Besides, it raises questions since the WTD and river flow simulated by LEAFHYDRO (section 3.1) are not particularly realistic, although not very bad either. The paragraph at L13-20 p3 is very general and seems odd when the introduction starts to present the specificities of the presented work (starting at "Our work, p3, L5). The last part of the introduction (p2 L21 to P4 L8) reviews Spanish hydrology, and finishes on irrigation. Eventually, the specific research questions of the paper are not clearly stated by the end of the introduction.

Authors: We have reshaped the introduction as we agree with this and other reviewers in that the paper would benefit from it. Now the structure is clearer and simpler. It still starts from general statements on groundwater interactions, soil moisture memory and observational evidences of groundwater-soil coupling. Then we review other modelling efforts. Then we introduce particularities about the Iberian Peninsula and literature on it. Finally, we introduce the research questions (that we did introduce in the first paragraph in the original submission) and the particularity of LEAFHYDRO calculating lateral drainage that makes it a candidate to tackle challenges presented earlier.

   (c) The paper frequently refers to a "bimodal" memory of soil moisture induced by GW persistence, but this term "bimodal" is not very clear: why isn't it just normal memory? I urge

the authors to define what they really mean.

Authors: The author is right in that the term memory should suffice, but we introduced the term "bimodal" to insist on the capability of the fluxes to go two-ways and the soil moisture to not only remember dry conditions but also wet conditions providing water to the system during dry years following wet ones.

(d) Consider gathering the validation of river flow and sensitivity of river flow to WT vs FD.

Authors: Responded above.

(e) Consider presenting Fig. 10 before Fig. 9, which makes a nice introduction to Fig. 10, and justifies why time correlation is analyzed on time series of annual means, while many papers in the literature consider time lags of months. The paper may insist on memory at pluri- annual timescale, which is quite novel in the literature surface-GW interactions.

Authors: Yes, this is a good point about changing the order. We originally considered that we should send the message of the water table affecting the annual correlations and then presenting the annual maps in Figs. 10 and 11 (now 11 and 12) for the reader to better appreciate these effects. Of course we could change them if the reviewer insists. Thanks for the tip, we have included "at pluri-annual timescale" in the title of Section 4.2.

(f) The conclusion is mostly a summary of what was just presented in sections 3 and 4. The summary part should be strongly shortened, to better highlight the main findings instead of again comparing % changes. Another advantage would be to leave space for a real discussion of the results, which is cruelly lacking.

(g) In particular, the results and the conclusion they support are likely dependent on the model, its assumptions, and the forcing datasets (meteorology, soils, vegetation). This must be said and leads to compare the conclusions of the paper to the literature.

(h) For instance, the underestimation of riverflow (Figs 5 and 15) means that ET is too strong in the model(s) or precipitation too weak: can't it create a bias in the sensitivity of ET to WTD? This should be discussed, in relationship with the quality of meteorological and vegetation forcing datasets, or the fact that irrigation is not taken into account (cf. p9 L31-32).

(i) The perspectives could also be developed… Are "coupled land hydrology-climate models" (p16, L29) the only to move forward?

Authors: We have divided Section 5 into "Discussion", where we have discussed the results maintaining our structure of groundwater influences on three parts of the land surface hydrology system and have added comparisons with external work cited in the introduction, and a shorter "Summary and conclusions" section at the end. Please see the revised version of the manuscript.

The attribution of the improvements to our WT run, and therefore suggesting that the ET enhancement is right during the dry season is based mostly in our wtd validation.

Therefore a validation of ET fluxes with suitable data is, of course, another perspective for future work. We just were not able to do it for this paper for lack of data and did not think it was necessary to mention in the text.

We believe, as has been pointed out in the text, that the driving precipitation is biased low.

**References (for those not cited in the discussion paper):**

Gupta, Kling, Yimaz, Martinez: Decomposition of the Mean Squared Error and NSE Performance Criteria: Implications for Improving Hydrological Modelling; Journal of Hydrology, 377(1), 80-91, doi:10.1016/j.jhydrol.2009.08.003, 2009.

Quintana-Seguí, P., Turco, M., Herrera, S., & Miguez-Macho, G.: Validation of a new SAFRAN-based gridded precipitation product for Spain and comparisons to Spain02 and ERA-Interim, Hydrology

and Earth System Sciences, 21(4), 2187–2201. doi: 10.5194/HESS-21-2187-2017, 2017.

Weedon, G.P., Balsamo, G., Bellouin, N., Gomes, S., Best, M.J. and Viterbo, P., 2014. The WFDEI meteorological forcing data set: WATCH Forcing Data methodology applied to ERA-Interim reanalysis data. *Water Resources Research*, 50, doi:10.1002/2014WR015638.

Westerhoff, R., White, P., and Miguez-Macho, G.: Application of an improved global-scale groundwater model for water table estimation across New Zealand, Hydrol. Earth Syst. Sci., 22, 6449-6472, https://doi.org/10.5194/hess-22-6449-2018, 2018.

---

## Author Comment (AC3) · 9 Jul 2019

**AUTHOR'S RESPONSE TO RC3:**

Manuscript hess-2018-626 by Martinez-de la Torre & Miguez-Macho: "**Groundwater influence on soil moisture memory and land–atmosphere interactions in the Iberian Peninsula**"

This paper discusses the role the water table plays in the terrestrial water cycle through the provision of vertical fluxes it provides for crops to evapo-transpire. The authors apply a Land Surface Model LEAFHYDRO that also simulates the dynamics of water table. They present results that show the difference between the simulated soil moisture values with and without the inclusion of the water table. I think this paper addresses relevant scientific questions within the scope of HESS, it represents interesting tools and ideas; however, the presented methodology and data fall short from supporting the reached conclusions.

It is clear that significant work has been undertaken to produce the results; however, I think because the authors are dealing with many processes including land surface, unsaturated zone, and saturated processes, the paper as it stands lacks a lot of information that are necessary to convince the reader with the applied methodology and possibly the repeatability of the experiment. In addition, there are concerns related to the structure of the paper where introduction, results, and discussions are all mixed together.

Authors: Thanks for the reviewer's complete assessment. We understand the issues pointed out by the reviewer and have introduced substantial editions and changes to the manuscript to address them. We discuss such changes in response to the reviewer's specific comments below.

My points below make these comments clear:

At the beginning of the introduction, the authors state that "groundwater exchanges with the land surface occur via vertical fluxes through the water table surface, and horizontal water redistribution via gravity driven lateral flow". The authors must be specific regarding the type of the lateral flows. Are these flows in the saturated zones only? Are they in the main aquifers or perched aquifers? Or do they also include what is called through flows, i.e. lateral movement of infiltrated due to the existence of low permeability materials above the water table?

Authors: In the introduction and the rest of the paper, we mean lateral flow *within the saturated zone,* as explained in the methodology section. We have added this to the text pointed out by the reviewer. The model does not represent lateral transport in the unsaturated zone.

The scale the authors are dealing with is a national scale. It is expected that many types of hydro-geological conditions will be met at this scale. It is not expected that they will deal with all possible hydro-geological settings, however, the paper must clearly state the selected hydro-geological condition the model is applied to. A diagram showing a conceptual model of this hydrogeological setting is needed. All the results to be presented and discussed has to be put always within the context of this conceptual model.

Authors: Yes, agreed, thank you. The original submission did not go into enough detail about the methodologies of the model. After this and the rest of reviewer´s comments, we have edited substantially the methodology section 2.1, adding information about how the model represents the hydro-geological conditions using the conductivity parameters. Please see the revised manuscript.

The introduction must be more focused. The paper states the aim of the paper in the first paragraph of the paper. The introduction then tries to explain the reasons for undertaking the work afterwards. I think the argument should be built the other way round. In addition the introduction includes description of the methodology applied (Page 3 Lines 5 to 12) and site description (paragraph starting from Line 20 on Page 3). I have difficulties with some of the definitions and terminology used. For example, on Line 34 Page 3, the authors write "reflecting the importance of groundwater memory". Why do they need to call it memory? It is the groundwater storage that reduces the impact of extreme weather events. The use of

positive and negative recharge is also confusing (although clearly defined) and not intuitive.

Authors: We fully agree and have edited the introduction section, stating the research questions and the particularities of our approach (Page 3 L5-12 of the original submission) at the end. The discussion about the Iberian Peninsula and its hydrological characteristics has been slightly modified, but we still think it should be part of the Introduction as it focuses the reader on the problems that the paper has to deal with.

Yes, the reviewer is right, we have rephrased "reflecting the importance of groundwater memory", it now reads "reflecting the importance of groundwater influence on surface hydrology".

About the use of positive and negative recharge, we acknowledge that the groundwater recharge is often referred to as the positive flux into the groundwater reservoir. In this work, we have followed the model signs for fluxes, as in upward is positive (like evapotranspiration from the surface) and downward is negative (like the water flux through the soil layers and then into the groundwater). We have changed the name of the flux to "net recharge" in the revised manuscript in order to clarify this point at different instances. This is now clarified the first time the net recharged is referred to in the manuscript in Section 2.1: "*The water flux through the water table or net recharge R is the sum of gravitational downward groundwater recharge and capillary flux, and depending on soil wetness and atmospheric demand, it can be downwards, causing the water table to rise, or upwards, causing the water table to deepen*"

Section 2 must be split into two sections one describing the study area including the information that are presented in the "Introduction", in addition to the conceptual model. The other section must be dedicated to the Methodology, which must include a lot more information than what is already presented. For example:
Equation 1 shows the temporal variations of groundwater storage as a response to recharge. What about the soil moisture temporal variations?

Authors: Information has been added in Section 2.1 about the flux calculations within the unsaturated zone, following the Richards' equation.

How does the model calculate evapotranspiration? Does it calculate runoff? Does it account for overland routing? Is overland water added to the groundwater flows emerging in the rivers to calculate total flows at the gauging station?

Authors: Information has been added at the end of Section 2.1 about the ET methodology.

Details on the river routing scheme and a sketch on the river parameters calculation have been added in Section 2.2.

The model does not calculate overland routing, but it does calculates surface runoff as infiltration excess, which is added to groundwater baseflow in the cell to calculate streamflow. Further details are part of the original model LEAF, described in the reference given as Walko et al., 2000.
How is the capillary flux calculated? Is it dependent on the position of the water table? (It is clear it is but at least it must be described in the methodology)

Authors: Details on this have been added to Section 2.1.
How capillary forces are presented in the model? When a water table exists, the water is available to evapo-transpire wherever the water table depth is?

Authors: Yes, regardless of the water table position, the vegetation has access to the soil water within the root zone depth. Of course if the water table is there, this means higher water availability for the plants.

It is not clear how the high resolution steady state simulation results are used in the low resolution time variant results (This is explained later, but what is mentioned in Section 2 is not enough to clarify this approach.

Authors: Yes, agreed. We have realized that Section 2.4 was not completely clear in the original submission. In the revised manuscript we have rewritten Section 2.4 to make it more explanatory.
It is stated that the shallow water table slows down drainage. If the soil is not fully saturated and

the water does not pond on the surface, how the shallow water slows down drainage?

Authors: If soil moisture increases with depth when approaching the water table, as it is usually the case, capillary fluxes are upward. The always downward gravitational flux may dominate the net flux, but the latter is certainly smaller than when there is no groundwater. In the FD run, the net flux at the bottom of the soil columns is just the gravitational flux, with no upward capillary flux to counteract it, at least partially, thus drainage is faster. Furthermore, when the water table is within the resolved layers, drainage at 4m is zero, and if the water table reaches the surface, infiltration ceases altogether. In the FD run, drainage at 4m is always occurring when the bottom layer is above field capacity.

It must be explained here that rivers could be influent and effluent

Authors: It was very briefly explained in the original submission, refering to "gaining" and "losing" streams. After the reviewer suggestion, we have added the following information in Section 2.1:

"*This flux can occur as groundwater discharge (subsurface runoff) into gaining streams when the water table is above the river, sustaining stream baseflow, or as river infiltration into the groundwater reservoir in losing streams when the water table is below river bed. For gaining streams, LEAFHYDRO approach combines the physically based parameters of Darcy's law into a parameter called river conductance, commonly used in groundwater modeling literature, like the MODFLOW model (Harbaugh et al., 2000). Even though the river conductance is physically based and observable, detailed data on river geometry and bed sediments are lacking for the region studied, hence it needs to be parametrized. Such parametrization consists in a representation of the river conductance that includes two contributions; an equilibrium part, and a dynamic part that depends on the water table deviation from equilibrium at the time. Further details on this dynamic river conductance parametrization and discussion on its choice are found in Miguez-Macho et al. (2007). For losing streams, the distance of flow or river bed thickness in Eq. 10 is the same as the water table minus riverbed elevation difference (third parenthesis in Eq. 10, only with negative sign provided that wth < z_{r}), and hence these factors cancel out one another, leaving the flux calculation to be given by*
*(new Eq. 11).*

*Therefore, the losing stream flux Q_{r} in the model is not dependant on the water table position, once the latter is below riverbed, but on the groundwater-rivers hydraulic connection.*"

Are the groundwater flows also driven using Darcy's law or is it based on hydraulic gradient only? What is the calibration procedure used to find the spatially distributed hydraulic conduct values?

Authors: Further explained in the revised version of the manuscript (Section 2.1)

Section 2.2 provides information about the source of data but no information about the data are provided. For example information about the spatial distribution of landuse is important to understand the amount of water extracted by evapo-transpiration from the soil store. Nothing is mentioned about the hydrogeological data used in the model such as the values of the hydraulic conductivity and storage coefficient of the aquifer, river bed conductance values, etc.

Authors: Thanks. We believe that with the inclusion of Equations 1 and 2 and the last paragraph about ET and PFTs in Section 2.1 , the model approach is clearer now. We have edited also the first paragraph in Section 2.2 as follows:

"*The 11 soil textural classes used in LEAFHYDRO, necessary to derive soil parameters in Eq. 2 controlling the vertical water fluxes, are defined by the United States Department of Agriculture (USDA) from fractions of silt, clay and sand. The data for top (0-0.30 m depth) and bottom (0.30-4 m depth) soil layers comes originally from the Food and Agricultural Organization of the United Nations (FAO) world database (http://fao.org/soils-portal/soil-survey). Other processes in the model, such as evapotranspiration, need parameters dependent on the vegetation type (PFTs) at the land surface. For vegetation type we usethe COordination of INformation on the Environment (CORINE) Land Cover Project database (EEA, 1994)*"

Details about lateral groundwater flow calculations and aquifer properties are also included as follows:

"*Lateral groundwater flow $Q_n$ is determined by the slope of the water table surface, applying Darcy's law the water flux from the $n^{th}$ neighbour into a model cell is given by*

*$Q_n = cT(wtd_n-wtd)/l$*

*where c (m) is the flow cross-section connecting the cells, $T$ ($m^2\ s^{-1}$) is the flow transmissivity between the cells, wtd and 30 $wtd_n$ (m) are the water table depths for the centre cell and the $n^{th}$ neighbour cell, respectively, and l (m) is the distance between cells. T is calculated as an integration of the lateral hydraulic conductivity at saturation, for which the model uses observedvalues of the anisotropy ratio relating vertical and lateral conductivities (Fan et al., 2007), and assumes exponential decay of the vertical hydraulic conductivity at saturation $KV_f$ with depth, as*

*$KV_f = K0exp(-z'/f)$*

*where $K_0$ ($m\ s^{-1}$) is the known value at 1.5 m deep, $z'$ (m) is the depth below 1.5 m and f (m) is the e-folding depth, 5 calculated as a function of terrain slope β as f = 75/(1 + 150β), where f is limited to 4 m when β ≥ 0.118.* "

In Section 2.4, can you state please which groundwater model is used with the Mosaic LSM recharge model to calculate the initial EWTD? On Lines 10 to 18 (Page 6) it is unclear which model has the high resolution and which one has the low resolution. A diagram that shows the steps followed in methodology will be helpful. Text from Line 18 onward in this section are results. Why are they included in this section?

Authors: Section 2.4 has been edited as pointed out before. Please see the revised manuscript. Even though we agree that our EWTD is a result, we decided to include it at this point in the methodology section, since it is used as an initial condition for the main experiment.

In Section 3 the authors dip into discussing the validation of a model while no infor- mation about the hydraulic parameters used in the model are provided. These include parameters controlling overland, subsurface, and unsaturated flows as well as soil and landuse data. They claim that the temporal variabilities are reproduced. However, with the lack of the parameter values and the definition of the context (assumptions and conceptual model) within which the model is built, this conclusion is easily challenged.

Authors: Thanks. We again refer to the revised and more detailed new Section 2 that now includes discussions about all the required parameters in calculations.

In Section 4.1 (Lines 25 to 30 on Page 9), the authors define positive and negative recharge in an unintuitive way since in groundwater, recharge is referred to as inflow to the groundwater reservoir and the opposite is a discharge from the water store and that could be in any direction (like the upward capillary fluxes). The sentences on Lines 10 to 14 on Page 10 are not very well formulated and together with the comment above, it is difficult to understand the point the authors are trying to make. On Line 15, the argument "this cycle is more pronounced the shallower the water table" is not very strong since Figures 6c to f all show seasonal variations across the whole peninsula.

Authors: Yes, we have responded to this concern about the signs of the recharge flux above. The point we try to make in the referred lines is to differentiate between large areas of low positive flux and river valleys with high positive flux. We have slightly edited the sentences and we believe the point is clearer now:

"*However, in river valleys where steep slopes in the water table head drive strong local lateral groundwater flow convergence, groundwater-fed ET can exceed precipitation by large amounts, resulting in higher values for the positive recharge. This is apparent n Fig. 7a along the main river valleys crisscrossing the dry Mediterranean areas of the Iberian Peninsula.*"

We agree with the reviewer in that the point we made in Page 6 line 15 (original submission) is not sufficiently supported by the figure. We have deleted the sentence. The point about seasonality and the influence from shallow water tables is made in the following sentences. Thanks

In Section 4.3: can you please state how annual anomalies are calculated? Is it a difference from a long term average value or the difference from an average calculated on the day the anomaly is determined?

Authors: Anomalies are differences between the given year values and annual means in the simulations. It has been clarified in the revised manuscript.

Line 28 Page 11: are anomalies in precipitation and anomalies in soil moisture corre- lated or are the anomalies in soil moisture correlated with precipitation. Please clarify

Authors: The correlations are calculated between anomalies. In this case, anomalies in precipitation and anomalies in soil moisture. It has been clarified in the revised manuscript.

Section 4.4 Line 21: "water table depth (red lines)" are observed or simulated? If simulated is it from the model with water table or with free drainage?

Authors: It is the water table depth simulated in the WT run. The FD run does not simulate any water table. It has been clarified in the revised manuscript.

Figure 9: Please correct the caption for the left figure which should be related to the free drainage (FD)

Authors: Corrected. Thanks for spotting this.

In Figure 11, I expect the soil moisture anomalies calculated from the simulation with a water table to be lower in absolute value than those calculated from the simulation with free drainage. This appears to hold true for all hydrological years except Years 8 and 9 (Compare row 3 to row 2). Why?

Authors: The soil moisture anomalies when the water table is considered do not necessarily have to be smaller than in the FD simulation. In areas where the water table is shallow, while this is the case, it is true that variations are buffered. However, if a shallow water table deepens as a result of a prolonged drought, so that the connection with the top soil is lost, soil moisture anomalies are going to be larger than in a FD simulation. Soil moisture values in both runs would be similar, but the anomaly is going be larger in the WT run where the soil is typically wetter due to the presence of a shallow water table. This is what happens in years 8 and 9, after the drought. In the FD run, soil moisture anomalies rapidly follow those in climate. In the WT run, however, the water table has not fully recovered, and soils are still much more anomalously dry.

Finally, I think the paper has to include a Discussion section where the analysis of the results has to be aligned with the assumptions listed in the conceptual model together with the hydraulic characteristics of the studied domain and the landuse controlling the amount of evapotranspiration from the soil zone. While the amount of work that has been taken and presented must be recognised and appreciated, I think the addition of a discussion section and rewriting the conclusion section to address the main findings concisely will greatly improve the presentation of this work.

Authors: Yes, agreed. We have re-structured the paper, including a Discussion section after the results and a shorter conclusion section at the end. Please check the revised manuscript.

---

## Author Comment (AC4) · 24 Jul 2019

I have added a combined pdf with the 3 responses to reviews as it was required in the "final response", but I encourage to use the responses posted here as author comments since they are easier to handle. We have produced the manuscript using online latex, therefore we do not have a tracked version in pdf comparing the original and final versions. The changed introduced are very clearly stated though, with quotes of the revised texts, in the author responses to reviews.

Thanks

---

## Author Response (AR2)

**RESPONSE TO MINOR REVISIONS, REFEREE #3**

The paper as it stands is clear and I think ready or publication. However, I found a couple of very minor issues the authors need to address before final publication. These are as follows:
- The first line on Page 5 discusses the signs of the fluxes based on their directions, which is not consistent with the negative sign given to the river/aquifer flux in Equation 3. Please revise.

Authors: Thanks for the comment. The reviewer is right, we have rephrased the erroneous sentence in the revised manuscript as follows: "Fluxes R and $Q_n$ in Equation 3 are assumed to be positive when going into the groundwater reservoir and negative when going out of it, whereas $Q_r$ is positive when going into the river and negative when going from the river into the groundwater reservoir".

- For Equation 6, the authors need to define h1 and h2 on Figure 1. Also line 16 just after Equation 6 mentions "the head of layer height". This is not clear and needs revision.

Authors: Thanks for this point. We have now identified h1 and h2 in Figure 1 in the revised manuscript, which looks as follows:

[Figure]

[Figure]

We have made the figure larger in the manuscript so that all the variables can be

easily read.

We have also revised the confusing text changing "head of layer height" by "depth of the top of layer". The revised manuscript refers to hx as "the depth of the top of layer x ". In addition, we have defined z1 and z2 after they are introduced in eq (4) as the depths of midlayer 1 and 2, respectively. We believe this is clear now, thanks.

-End of Page 6 and start of Page 7, the authors discuss the calculation of Transmissivity T. This paragraph is not clear because the vertical hydraulic conductivity is shown (Equation 13) while we are discussing the lateral movement of water and the lateral transmissivity. Also, if T is equal to zero at 4 meters, what is the transmissivity value that controls the lateral flow in this case? How does the WT goes deeper than 8 meters, if only PE is taken from a soil zone that is 8 m deep in this case?

Authors: We need to know the lateral conductivity $K_L$ variation with depth in order to calculate the transmissivity T as an integration of $K_L$ along the vertical axis. To do so, we derive it from vertical conductivity $K_V$ using the anisotropy ratio relating lateral to vertical conductivity $\alpha=K_L/K_V$, which adopts values dependent on the clay content of the soil as detailed in Fan et al. (2007) (section 3.3 and Table 2). Then, for vertical conductivity, we assume an exponential decay from the known value at a depth of 1.5m (eq 13). The e-folding depth of this exponential decay depends on terrain slope (thinner soils in steep terrain and thicker soils in flatter areas), with a lower limit of 4m at slopes bigger than 0.118 and maximum value of 75m in flat terrain. Therefore, T is not zero at 4m because K is not zero either; the 4 m value is the lower bound for f, the e-folding depth for K. We agree that this might not be clear enough in the referred paragraph, and we have edited it in the new revised manuscript as follows:

"T is calculated as a vertical integration down from the water table depth of the lateral hydraulic conductivity at saturation $K_L$ (Fan et al., 2007), which is derived from vertical conductivity $K_V$ using the anisotropy ratio parameter $\alpha$ relating both parameters as $\alpha=K_L/K_V$. We apply values of $\alpha$ dependent on the clay content of the soil and within the range of observations in nature, as detailed in Fan et al. (2007). For vertical conductivity, we assume an exponential decay with depth, as

Eq 13

where K0 (m s−1) is the known value at 1.5 m deep, z' (m) is the depth below 1.5 m and f (m) is the e-folding depth. The e-folding depth f is calculated as a function of terrain slope $\beta$ as f = 75/(1 + 150$\beta$), with a lower limit of 4m where $\beta$ ≥ 0.118. Further details …."

- Line 16 Page 13 " water table is deeper 3" , the sentence is missing

Authors: Yes, this is right. That sentence was intended to refer to Fig. 3 with the Equilibrium Water Table Depth, we have corrected it in the revised manuscript as: "Where the water table is deeper (Fig. 3)". Thanks for pointing this out.

- Line 19 Page 13 ""when the surface water is minimal", please revise the sentence regarding the words "water balance"

Authors: We have edited the sentence in the revised manuscript, as "when the balance between precipitation and evapotranspiration is at its seasonal minimum (P-ET=-0.80 mm day$^{-1}$)".

- Section 4.2, please clarify and add if correct that the volumetric water content "eta" (dimensionless) is used to describe the soil moisture content in this section and in the figures

Authors: That is correct. We have clarified that the soil moisture is referred in terms of volumetric water content in both the text in Section 4.2 and the Fig. 8 caption.

[revised manuscript text omitted]